# Single cell analysis in head and neck cancer reveals potential immune evasion mechanisms during early metastasis

Hong Sheng Quah [1,2,14], Elaine Yiqun Cao[3,14], Lisda Suteja[1,4], Constance H. Li[1,2], Hui Sun Leong[1], Fui Teen Chong[1], Shilpi Gupta[5], Camille Arcinas [1,2], John F. Ouyang [3], Vivian Ang[5], Teja Celhar [6], Yunqian Zhao[6], Hui Chen Tay[6], Jerry Chan[7], Takeshi Takahashi[8], Daniel S. W. Tan [1,2,9], Subhra K. Biswas[5], Owen J. L. Rackham [3,10,11] & N. Gopalakrishna Iyer [1,2,12,13] ✉

Profiling tumors at single-cell resolution provides an opportunity to understand complexities underpinning lymph-node metastases in head and neck squamous-cell carcinoma. Single-cell RNAseq (scRNAseq) analysis of cancer-cell trajectories identifies a subpopulation of pre-metastatic cells, driven by actionable pathways including AXL and AURK. Blocking these two proteins blunts tumor invasion in patient-derived cultures. Furthermore, scRNAseq analyses of tumor-infiltrating CD8 + T-lymphocytes show two distinct trajectories to T-cell dysfunction, corroborated by their clonal architecture based on single-cell T-cell receptor sequencing. By determining key modulators of these trajectories, followed by validation using external datasets and functional experiments, we uncover a role for SOX4 in mediating T-cell exhaustion. Finally, interactome analyses between pre-metastatic tumor cells and CD8 + T-lymphocytes uncover a putative role for the Midkine pathway in immune-modulation and this is confirmed by scRNAseq of tumors from humanized mice. Aside from specific findings, this study demonstrates the importance of tumor heterogeneity analyses in identifying key vulnerabilities during early metastasis.

In most solid tumors, development of lymph-node metastasis portends poor outcomes, pre-dating distant metastasis[1–3]. In head and neck squamous-cell cancers (HNSCC), these patients are treated with curative intent by surgery and radiation therapy with the prime objective of eradicating existing and future diseases by depleting clones with a metastatic potential[4,5]. Metastasis is a continuum of phenotypes ranging from pre-metastatic features (local invasion, increased motility), circulating tumor cells/emboli, microscopic lymph-node deposits, gross nodal involvement and adjacent soft-tissue invasion, oligo-metastasis and finally, distant metastasis[6]. Most

[1]Cancer Therapeutics Research Laboratory, National Cancer Centre, Singapore, Singapore. [2]Academic Clinical Program in Oncology, Duke-NUS Medical School, Singapore, Singapore. [3]Program in Cardiovascular and Metabolic Disorders, Duke-NUS Medical School, Singapore, Singapore. [4]Cancer and Stem Cell Biology Program, Duke-NUS Medical School, Singapore, Singapore. [5]Singapore Immunology Network (SIgN), Agency for Science Technology and Research (A*STAR), Singapore, Singapore. [6]HuNIT platform, Singapore Immunology Network (SIgN), Agency for Science, Technology and Research (A*STAR), Singapore, Singapore. [7]Department of Reproductive Medicine, KK Women's and Children's Hospital, Singapore, Singapore. [8]Laboratory Animal Research Department, Central Institute for Experimental Animals (CIEA), Kawasaki, Japan. [9]Division of Medical Oncology, National Cancer Centre, Singapore, Singapore. [10]School of Biological Sciences, University of Southampton, Southampton, United Kingdom. [11]The Alan Turing Institute, The British Library, London, United Kingdom. [12]Department of Head and Neck Surgery, National Cancer Centre, Singapore, Singapore. [13]Division of Medical Sciences, National Cancer Centre, Singapore, Singapore. [14]These authors contributed equally: Hong Sheng Quah, Elaine Yiqun Cao. ✉e-mail: gmsngi@nus.edu.sg

studies focus on the terminal event, highlighting the role of definitive epithelial–mesenchymal transition (EMT); however, bulk analyses in HNSCC suggests that EMT does not appear to be a pre-requisite for lymph-node dissemination[7–11]. Recent studies have also highlighted that EMT itself exists as a spectrum, and tumor cells exhibit a significant amount of plasticity which may account for the range of clinical manifestations observed[12, 13]. One likely explanation could be insufficient resolution of traditional methods in identifying infrequent sub-clones with true metastatic potential that would be targetable for anti-metastatic therapy. Single-cell analyses offer ways to resolve both issues: identification of rare clones with true metastatic potential and identifying pathways and vulnerabilities that can be exploited in the clinical setting to prevent further dissemination of these.

The role of the immune system during the metastatic cascade is gaining clinical relevance with current advancements in checkpoint blockade therapies[14]. This is especially pertinent in the context of lymph-node metastasis, as lymph nodes are believed to be the main organ for T-cell priming, expansion and trafficking[15]. Understanding the mechanisms by which tumors evade immune-based killing within lymph nodes is critical to target early metastases[16–19]. Again, this can be addressed by single-cell analyses by defining the immune landscape, and in-depth dissection of interactions involved during immune evasion at the primary and nodal sites.

Here, we profile primary and early (nodal) metastatic HNSCC tumors using single-cell RNAseq (scRNAseq) and TCRseq (scTCRseq), and analyze this data by reconstructing evolutionary trajectories focusing on cell types that are known to transit across the two subsites: cancer cells and CD8+ T cells. The objectives are to identify pre-metastatic tumor subpopulations and targetable vulnerabilities, and to determine the evolutionary trajectory of tumor-targeting T cells as well as dissect pathways tumors employ to evade immune destruction during nodal dissemination.

## Results

### Single-cell transcriptional states of primary and lymph-node metastasis in HNSCC

To delineate 'whole-tumor' single-cell landscapes in primary tumors and lymph-node metastases, we developed a protocol to rapidly process freshly resected tissue for single-cell RNA sequencing (scRNAseq) and establishing primary cultures (Fig. 1a)[20, 21]. Tumors were harvested from fourteen treatment-naïve patients with locally advanced, HPV-negative HNSCC from primary and cervical lymph nodes (Supplementary Data 1, 2). Seven pairs were processed for scRNAseq and single-cell T-cell receptor sequencing (scTCRseq), while primary cultures were successfully established for seven.

scRNAseq data for fresh tumors describes 53,459 cells (3553–11,308 per patient) and 23,148 genes (details on quality controls steps in "Methods" and Supplementary Fig. 1a–c). Using Seurat v3.0, the data was normalized, pooled, and clustered (Fig. 1b, c and Supplementary Fig. 1d). Canonical markers were used to broadly annotate these populations into epithelial (*KRT7, KRT17*), salivary (*STATH*), fibroblasts (*COL1A2*), endothelial (*PECAM*) and immune *(PTPRC)* cells (Fig. 1d and Supplementary Fig. 1e). Fibroblasts were further subdivided into cancer-associated fibroblasts (CAFs; *MMP2*) and myofibroblasts (*ACTA2*), while immune cells were organized into T-(*CD3E, NKG7*), NK- (*NKG7, XCL2*), B- (*CD79A*), plasma- (*IGHG1*), mast- (*TPSAB1*), conventional (*LAMP3*) and plasmacytoid (*LILR4*) dendritic cells, as well as macrophages/monocytes (*CD163*). These were well-distributed across samples from all patients, apart from salivary cells, which were only observed in one patient, likely due to harvest of adjacent parotid gland tissue (HN263). However, there were differences in composition between primary and metastatic sites (Fig. 1d), with higher proportions of CAFs and TAMs in the primary tumor, versus more B-cells, plasma cells and dendritic cells at the metastatic sites, typical of a lymph-node. These were similar to cellular composition

proportions derived from bulk data from TCGA (Supplementary Fig. 1f). Inferred copy number variant analyses on the epithelial population showed that copy number alterations were evident in >95% of cells, which confirmed that this subpopulation predominantly comprised cancer cells (Fig. 1e and Supplementary Fig. 1g). Copy number alterations (CNAs) were further analyzed using the CopyKat algorithm[22], and identified those frequently observed in HNSCC[23], including gains across chromosomes 7 and 8q and loss of 3p and 5q (Supplementary Fig. 1h). Significant overlap of CNAs was also noted between the primary and metastatic sites in each patient (Supplementary Fig. 1i).

### Tumor cells demonstrate varying degree of epithelial–mesenchymal transition during metastasis

We next focused on tumor cells (total of 6115 cells and 17,784 genes; 1427 unique genes per cell) by extracting only the epithelial population with copy number alterations. Using Seurat 3.0, we pooled and re-analysed this subset, visualized as distinct clusters for each individual patient, with varying degree of overlap across cells from primary and nodal sites (Fig. 2a and Supplementary 2a). Tumor-cell data can be accessed and interrogated as an interactive web application via the following Shiny app (http://hnc.ddnetbio.com/). One of the major objectives here was to identify pre-nodal cells, which are cancer cells within that primary tumor that have the capacity to metastasize to the lymph nodes, and hence we hypothesize should have similar gene signatures to cancer cells within the lymph-node. Visualizing the UMAP, it was evident that tumors from patients HN242 and HN257 show significant overlap in tumor cells derived from both sites. Although patients HN251 (cluster 10 vs. 11 nodal), HN279 (clusters 4,3,9 vs. 5 nodal) and HN272 (clusters 0 vs. 6 nodal) show distinct sub-clusters where nodal tumor cells appear to predominate, these were not sufficiently robust to support the identification a distinct pre-nodal subpopulation. Similar findings can be seen using PCA and TSNE (Supplementary Fig. 2a). In general, comparing EMT gene markers in primary *vs* nodal metastases populations, nodal tumor cells had higher EMT scores compared to the corresponding primary in all patients except HN257 (Fig. 2b).

Therefore, in order to identify the pre-nodal metastases subpopulation in primary tumors, we built trajectories using Monocle 2.0 to identify primary cells that were the 'nearest neighbor' to nodal epithelial cells. The trajectories were labeled based on the origin and direction based on the ground truth of site (i.e., primary tumor presumed to pre-date nodal disease)[24, 25], incorporating EMT-scores, and CytoTRACE (see Methods). CytoTRACE is a tool to determine degrees of differentiation, assuming de-differentiation co-occurs with the metastatic phenotype[26, 27]. The same strategy was also used to re-analyze the dataset generated by Puram et al. as an external validation (with 2076 epithelial cells available for analysis)[12]. The assumption for this approach is that evidence of the pathway leading from pre-nodal to nodal metastases would be evident, despite ongoing evolution in the primary tumor. This approach identified three different patterns (Fig. 2c, d and Supplementary Fig. 2c–u). The first, seen in patients HN251, HN242 and HN279, pseudo-time ordering demonstrated an ordered, progressive, stepwise transition from primary to nodal disease, with little further evolution in the primary tumor. Nodal tumor cells largely dominate the end of the trajectory with higher CytoTRACE scores. Major pathways over-represented across pseudotime include epithelial de-differentiation, oxidative phosphorylation and EMT (Fig. 2e). A similar pattern is seen in the Puram dataset for tumors p26 and p28 (Supplementary Fig. 2l, m). The second pattern shows a trajectory where the pathway to pre-nodal cells continues to be evident, however, the primary tumor also shows continual evolution. This is seen in HN272, as well as p25 in the Puram dataset

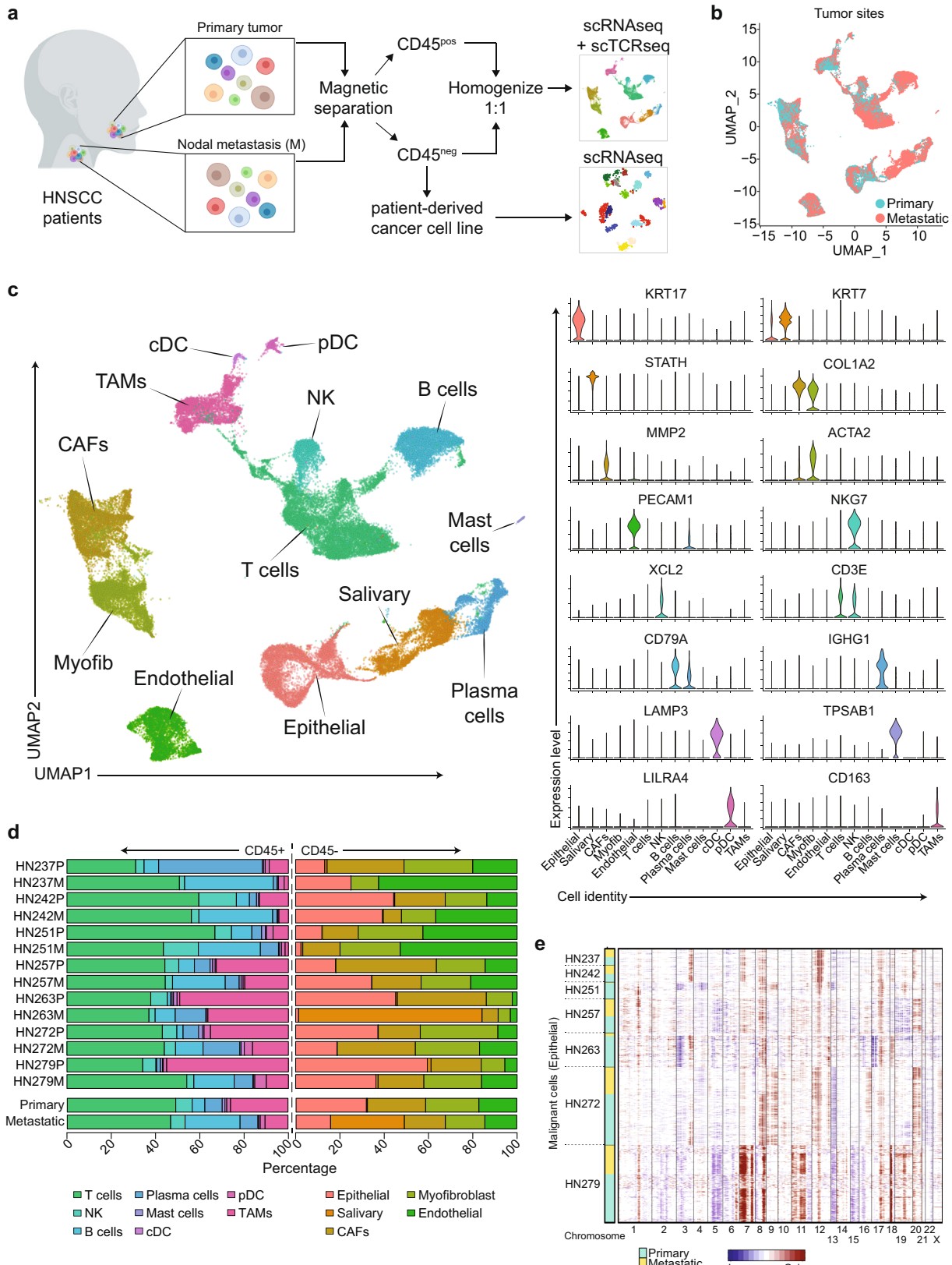

(Supplementary Fig. 2c, n). In both these patterns, we were able to further identify actionable genes associated with the trajectory from primary to pre-nodal cells using the GeneSwitches algorithm[28], which identified *AXL*, Aurora kinase, *TYMS* and *STAT2* at potentially critical genes in this process (Fig. 2f and Supplementary Fig. 2d–f). Analysis of the validation dataset similarly

identified *AXL* (p25, p26, P28), *STAT2* (p25, p26) and *AURKB* (p26, p28) (Supplementary Fig. 2o).

The third pattern seen in patient HN257, as well as p5 and p20 was more complicated as the primary tumor had higher EMT scores than nodal tumor cells, and tumor trajectories were haphazard with no directionality, and no evidence of an evolutionarily "earlier" time point

**Fig. 1 | Tumor samples for single-cell RNAseq. a** Workflow of sample acquisition, processing, and analyses for single-cell transcriptome and TCR clonality of tumors (and patient-derived cultures) from primary and metastatic lymph nodes of HNSCC patients. Diagram was created with BioRender.com. **b** Uniform manifold approximation and projection (UMAP) of scRNAseq from all 53,459 cells separated by primary tumors and metastatic lymph nodes from 7 patients. **c** UMAP of scRNAseq data of all cells from 7 patients with clusters denoted by colors and labeled according to inferred cell types. Violin plots show the expression of selected genes used to define the inferred cell types. **d** Distribution of different cell types (color), categorized by CD45+ or CD45−, for each patient sample (upper) and comparing primary and metastatic samples (lower) as indicated on the y-axis. **e** Chromosomal gains and losses prediction for malignant epithelial cells by inferCNV using non-malignant cells from respective samples as controls. Cyan indicates primary malignant epithelial; yellow indicates lymph-node malignant epithelial; sample identities on the y-axis, chromosome numbers on the x-axis.

in the primary tumor (Supplementary Fig. 2p–u). In HN257, Cyto-TRACE showed a distinct de-differentiated sub-population in the primary tumor that had high EMT scores and expression of *SNAI2* (Fig. 2g and Supplementary Fig. 2p–r). We hypothesized that this was an aggressive, rapidly evolving tumor subpopulation. In this subpopulation, differential expression analyses identified a panel of 132 upregulated involved in oxidative phosphorylation and tumor metabolism, and 45 downregulated genes involved in immune evasion (Fig. 2h and Supplementary Data 3). Based on these gene sets, tumors in TCGA with the same signature (based on RNAseq data) had significantly poorer outcomes (Fig. 2i and Supplementary Fig. 2s). Therefore, we postulate that in these tumors, distinct subpopulations in the primary tumor showed a more aggressive phenotype, that likely evolved further after nodal dissemination had occurred.

## Identifying vulnerabilities to target pre-metastatic tumor cells

We then proceeded to test whether specific targets identified in this manner present an opportunity for therapeutic intervention. scRNAseq using the C1 platform was performed on patient-derived cultures (PDCs) from primary and nodal metastatic sites (*n* = 7 pairs). The data was processed using Seurat 3.0 and PAGODA (pathway and geneset overdispersion analysis) (Fig. 3a and Supplementary Fig. 3a, b). We derived scRNAseq data for a total of 1317 cells and 55,216 genes. Similar to above, tumor-cell clusters were based on individual patients. However, PDCs demonstrated distinct separation between primary and metastatic cells, with EMT as one of the major differentiating principal component pathways (Fig. 3b and Supplementary Fig. 3a, b). Here, pre-nodal cells in HN137, HN159 and HN220 were identified as small primary subpopulations that clustered with metastatic cells.

Differential expression analyses for these pre-nodal populations identified *AXL* (in HN137) and *AURKB* (in HN159 and HN220) as putative actionable targets (Fig. 3c and Supplementary Data 4–6). Expression of these genes was validated using immunohistochemistry or immunofluorescence staining in both PDCs and respective tumor tissue (Fig. 3d and Supplementary Fig. 3c, d), and this was recapitulated on flow cytometry for AXL (HN137) and AURKB (HN159 and HN220) (Supplementary Fig. 3e, f, j, k). In HN137, expression of protein and transcript AXL was detected in a majority of metastatic cells compared with only a small sub-population of primary cells. For HN159, but unexpectedly not for HN220, AURKB protein expression was lower in metastatic cells when compared to primary cells. We focused on AXL and AURKB because both have specific inhibitors: BGB324 targeting cells with high AXL expression, and barasertib (pan-AURK inhibitor) targeting cells with limiting AURKA/AURKB levels. There were no differences in clonogenicity between primary and metastatic cultures from patient HN137 treated with BGB324, nor HN159 and HN220 treated with barasertib (Supplementary Fig. 3g–i). In contrast, all three metastatic lines HN137, HN159 and HN220 (treated with their respective drugs) demonstrated lower cell migration/invasion compared to untreated cultures, measured by scratch and Boyden chamber invasion assays (Fig. 3e–g). AXL-inhibition significantly reduced invasive potential of both primary and metastatic cells of HN137 (Fig. 3e) while AURK-inhibition significantly reduced the invasive potential of only metastatic cells of HN159 and HN220 (Fig. 3f, g). As AXL is a surface

membrane protein, primary cells were sorted into AXL low-, medium- and high-expressing cells. As predicted, BGB324 specifically inhibited invasion only in the AXL-high primary subpopulation compared to AXL-low cells (Fig. 3h, i, Supplementary Fig. 3j, k). We further analyzed bulk-RNAseq data for primary HNSCC tumors in the TCGA dataset to determine the correlation between AXL or AURKB with EMT (as a proxy for metastatic potential). The results demonstrated that AXL expression is significantly correlated with increasing EMT score, even if this analysis was limited to patients with no nodal metastasis (Supplementary Fig. 3l, m). Conversely, our analyses suggests that AURKB had the opposite trend, even in N0 tumors, although the association is less robust. These, data indicate AXL and AURKB play major roles in invasion and provide an opportunity for specific anti-metastatic therapy.

## Evolution of CD8+ T cells derived from analysis of primary tumor and lymph-node metastasis

CD3+ T cells form one of the major subpopulations sequenced at both primary and nodal sites. Data from 10,168 cells (covering 13,729 genes) were pooled, analyzed using Seurat, and visualized as ten distinct T-cell clusters (Fig. 4a). The identity of each cluster was delineated based on differential gene expression of known T-cell markers (Fig. 4b and Supplementary Fig. 4a, b). Some were distinct for CD4+ cells (Tregs and Tfh) and CD8 + cells (Pre-dysfunctional, Dysfunctional, Proliferative), while others comprise both CD4+ and CD8+ lineages (Naïve-like and Transitional). Majority of naïve-like cells were derived from nodal tissue while the remaining clusters appear to have equal representation from the primary and nodal metastatic sites (Fig. 4b and Supplementary Fig. 4c).

CD8+ T cells (total of 3387 cells, 11,847 genes) were extracted from this pooled T-cell dataset and re-analyzed after regression for cell cycle-driven artefacts to identify lineage-based clusters. CD8+ T-cell data can be accessed and interrogated as an interactive web application using the following Shiny app (http://hnc.ddnetbio.com/). Six distinct clusters were labeled as naïve, transitional, tissue-resident memory, pre-dysfunctional, proliferative and late dysfunctional based on canonical markers (Fig. 4c, d and Supplementary Fig. 4d, e). Using Slingshot, we performed trajectory analyses on the CD8+ T cells using the *CXCL13*-high, *LAYN*-high exhausted/senescent population as the end-point[29], and this identified two convergent trajectories (Fig. 4e). Expression plots across Trajectory 1 showed a progressive loss of naïve markers, gradual gain of dysfunctional (and senescent) markers and an intervening proliferative 'burst', that likely reflects expanding clones of tumor-targeting CD8+ cells (Fig. 4f). Specifically, this lineage suggests a scenario where naïve CD8+ T cells from lymph nodes or circulation were trafficking into the primary tumor with loss of circulating markers *KLF2*, *SELL* and *CCR7*, gain of tissue-resident marker *CD103/ITGAE*, progressive decline in the expression of naïve genes *TCF7*, *IL7R*, *CCR7*, and gradual gain of dysfunctional markers (*TIM3*, *CTLA4*, *TIGIT*, *CXCL13*, *LAYN*) with an intermediary proliferative burst with high levels of *MKI67*, *TOP2A*, *TYMS* (Fig. 4b, e, f). This is also reflected by progressive increase from *GZMK* to *GZMB*, *PRF1*, and *IFNG* in pre-dysfunctional to dysfunctional cells. In contrast, the trajectory of tissue-resident memory (TRM) to dysfunctional cells (Trajectory 2) shows fewer

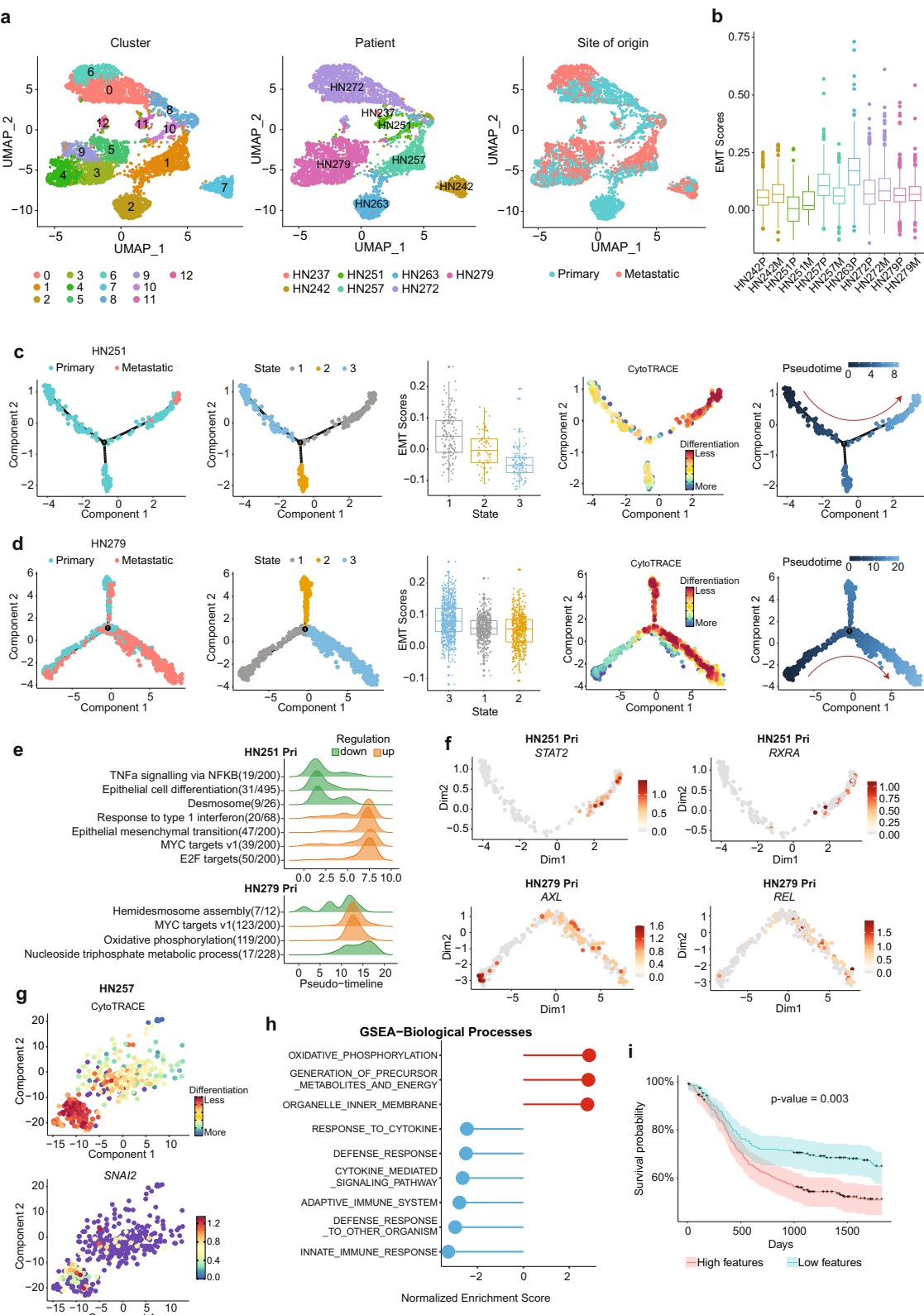

genes being activated as the expression level of many of the tissue-resident (*ITGAE*), dysfunctional (*CTLA4*) and granzymes (*GZMs*) genes were already upregulated (Fig. 4b). The Geneswitches algorithm was applied to Trajectory 1 (naïve-to-dysfunction) to predict key gene expression changes across pseudotime and identify factors that could account for these (Fig. 4g)[28]. Our results indicate the major nodes appear to be an early loss of

*KLF2*, intermediate increase in *NKG7* and late increase in *SOX4*, *DUSP4* and *RBPJ* (Fig. 4g, h).

## Modulating genes driving tumor-targeting cells dysfunction/exhaustion

Based on the data above, expression of *SOX4*, *DUSP4* and *RBPJ* appears to coincide with the transition between dysfunction and exhaustion,

**Fig. 2 | scRNAseq analysis of malignant epithelial cells and identification of pre-metastatic sub-population. a** UMAP of 6115 malignant epithelial cells only, clustered by Seurat clusters (left), patients (middle), and tissue origin (primary/metastatic) (right). **b** Turkey boxplot showing epithelial–mesenchymal transition (EMT) scores across patients and tissue origin (primary versus metastasis). $N = 6115$ cells. **c, d** Monocle plots demonstrating the derivation of pre-metastatic populations in HN251 ($n = 304$ cells) and HN279 ($n = 1764$ cells), (**c, d**) respectively, based on (from left to right) tissue origin, monocle clusters, EMT scores, CytoTRACE scores to derive trajectory. **e** Gene ontology pathways that are significantly altered across pseudotime derived in (**c, d**). **f** Potentially actionable genes identified to be increased in pre-metastatic population in primary tumors cells of HN251 (upper,

$n = 228$ cells) and HN279 (lower, $n = 659$ cells). **g** t-SNE plot of tumor cells in HN257 showing a highly aggressive sub-population in the primary tumor with high CytoTRACE scores and expression of *SNAI2*. $N = 403$ cells. **h** Geneset enrichment analysis (GSEA) showing normalized enrichment scores, and (**i**) Kaplan-Meier plot of TCGA data showing survival probability (%) over days in patients with high (red) versus low (blue) scores or features based on genes expressed by the specific subpopulation in (**g**). Shaded area shows 95% confidence interval of the centered median survival time and *p* value as indicated based on log-rank test. Boxplots in (**b, c, d**) centered at the median with hinges at 1st and 3rd quartiles and whiskers drawn from hinges to the lowest and highest points within 1.5 interquartile range.

but whether these genes modulate either the genes involved in dysfunction or progression to exhaustion remains untested. We attempted to validate these findings in two separate datasets. Re-analysis of data from Puram et al. (scRNAseq from 542 CD8+ T cells) showed that expression levels of SOX4 and RBPJ were higher in dysfunctional CD8 cell populations, while DUSP4 expression was more generalized (Fig. 5a and Supplementary Fig. 5a–c)[12]. The second scRNAseq dataset comprised T cells obtained from cutaneous squamous-cell carcinoma patients before and after treatment with PD1-blockade (Supplementary Fig. 5d)[30]. Although cSCC is a completely different entity than mucosal HNSCC, the cSCC scRNAseq data lends support to a more general concept of these specific genes in tumor-targeting CD8+ cells, and the effect of immune checkpoint blockade on the expression of these transcription factors. Here, all three genes showed higher expression in the exhausted CD8 subpopulation in this dataset (Fig. 5b and Supplementary Fig. 5e). However, only levels of *SOX4* and *DUSP4* were reduced after PD1-blockade, where there is expected reactivation of tumor-targeting clones and reduction in the exhaustion phenotype (Fig. 5c). Combining these results, *SOX4* appears to be the most likely gene associated during the transition from pre-dysfunction to dysfunction/exhaustion, which is supported by a number of recent publications[31,32]. To test whether *SOX4* as well as *DUSP4* played a role in the expression of genes involved in T-cell dysfunction or modulation of the exhaustion phenotype itself, we performed RNAi-based knockdown on activated PBMCs. Cells were transfected with Accell pooled siRNA against *SOX4*, *DUSP4* or non-targeting siRNA as controls, activated with anti-CD3/CD28 microbeads to induce the upregulation of selected T cell exhaustion-related markers and harvested for flow cytometry. Remarkably, *SOX4* knockdown resulted in a reduction in senescent CD57+, and dysfunctional PD1+ and CD39+ populations (Fig. 5d and Supplementary Fig. 5f, g). Given these results, we proceeded to knockdown *SOX4* using the same system, in a panel of expanded tumor-infiltrating lymphocytes (TILs) cultures derived from HNSCC primary tumors from four patients. In these TILs, siRNA against *SOX4* consistently reduced the frequency of PD1 + CD8 + cells, with no effect on the CD57+ population (Supplementary Fig. 5h, i). This was more evident when we focused our analyses on the putative tumor-targeting CD8+CD39+ subpopulations[33,34], where *SOX4* knockdown resulted in a reduction in the PD1+CD39+CD8+ TILs subpopulation (Fig. 5e and Supplementary Fig. 5j). Taken together, these data provide functional validation for the presented approach to CD8+ T-cell trajectory mapping and implicate SOX4 as a potential modulator regulating the expression of T-cell dysfunction/exhaustion markers.

## Establishing clonal architecture in CD8+ T cells using single-cell T-cell receptor sequencing

Clonal identifiers obtained by TCR analysis allows for elucidation of CDR3 sequences as well as providing a unique dataset to infer the lineage structure of T cells. Specifically, our current dataset can be used to model clonal selection and amplification across the CD8 + T-cell subpopulations and trajectories. We recovered productive TCR-alpha and TCR-beta sequences from 1461 and 1948 cells, respectively, and identified 1,590 unique TCR sequences (Supplementary

Data 7–13). Consensus sequence analysis using the GLIPH algorithm also showed minimal sharing across patients (only 5 of the 126 clusters with sharing across patients) (Supplementary Data 14). Clonal expansion was seen in 17.39% of CD8+ cells, and clone size ranged from 2 to 60 cells per clone (Fig. 5f and Supplementary Fig. 5k, l). Clonal overlap between the two different sites for each tumor (primary and lymph-node) was demonstrated in six out of seven patients (Supplementary Fig. 5m–u). There was a progressive increase in clonality across the dysfunctional gradient, with evidence of single naïve or TRM-derived clones subsequently primed and activated to give rise to multiple dysfunctional clones that span these trajectories (Fig. 5g, h and Supplementary Fig. 5m–u). We then proceeded to analyze CD8+ cells with shared clonotypes in greater detail, specifically to identify those that are "antigen-encountered" as defined as those specifically expressing genes of T cell activation (*GZMB*, *GZMA*, *PRF1*, *IFNG*, *TNFA*, *CD69* and/or *TNFRSF9*) (Supplementary Fig. 5v). This data suggests the presence of "antigen-encountered" CD8+ subpopulations with shared clonotypes at both sites. Generally, the number of "activation genes" is higher in the primary tumors than nodal metastases in six out of seven patients, reflecting higher tumor burden and an "earlier encounter" as expected.

There appeared to be patient-specific biases for one trajectory over the other. For example, there are CD8+ T-cell clones in patient HN272 that followed a naïve-dysfunction trajectory (Trajectory 1), with numerical expansion of lymph-node-derived naïve clonotypes, migrating to the primary site and captured there along a dysfunctional gradient (pre-dysfunctional, proliferative and then late-dysfunction) (Fig. 5g and Supplementary Fig. 5m, n, t). This supports a clonal replacement model where circulation is one of the major sources of tumor-targeting dysfunctional cells, which in this case is the regional lymphatics draining nodal tissue[30]. In contrast, we also observe clonal revival in patient HN263 and selected CD8+ T-cell clones in patient HN272. In these tumors, the dysfunctional gradient appears to comprise tissue-resident memory (TRM) cells derived from the primary tumor, which re-activated and amplified in numbers into putative tumor-targeting clonotypes (Fig. 5g and Supplementary Fig. 5m, n, s, t). This is consistent with a model of ongoing differentiation and proliferation of dysfunctional T cells at the tumor site itself [35]. It is likely that both mechanisms contribute to the dysfunction gradient, sometimes even within the same patient. For example, lineage tracing of CD8+ T cells in HN257 and HN272 demonstrates extensive trafficking and interplay between the primary site and lymph-node: there is evidence of lymph-node-derived naïve cells being activated, expanded (detected in at least two cells) and migrated into the primary site as expected, but also surprisingly TRM cells re-activating and expanding in numbers, and subsequently migrating to the lymph-node (Fig. 5g, h and Supplementary 5m, n, r, t). This scTCR data adds intriguing complexity to concepts of clonal expansion and lineage structure in a treatment naïve setting.

## Pre-nodal cells and immune microenvironment

Our analyses identified a pre-nodal sub-population in primary tumors with intrinsic properties of invasion and migration. However, metastasis also requires acquisition of an immune evasion phenotype. To

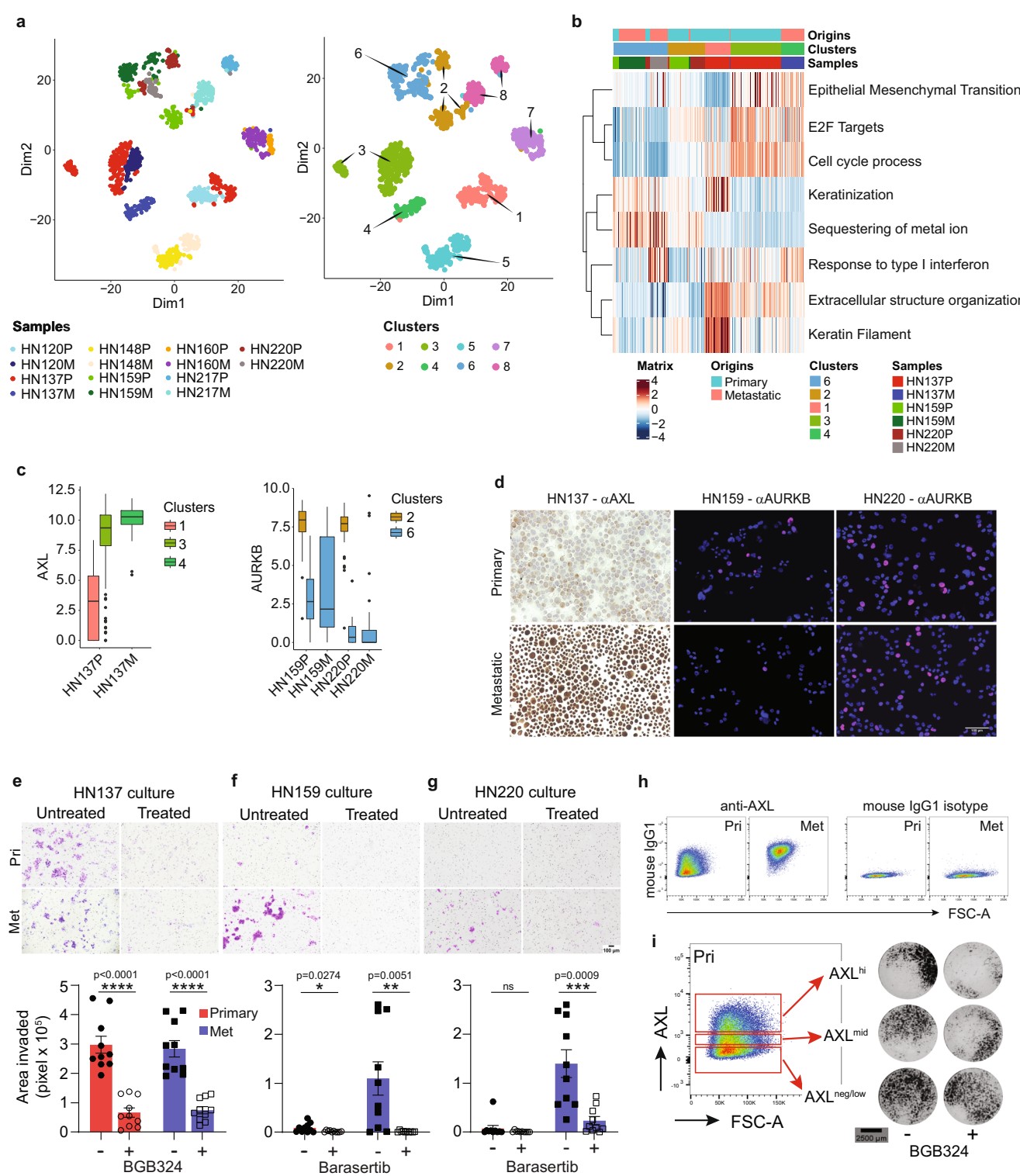

test whether the pre-nodal cells identified above demonstrated specific immune-modulatory phenotypes, we subjected three tumors (from our study) and two tumors (from the Puram dataset) each with a minimum RNAseq dataset to interactome analyses using Cellchat. To do this, we divided primary tumor cells into two subpopulations (primary and pre-nodal) and analyzed the interactions of these two tumor subpopulations with CD8+, CD4+ and T-reg lymphocytes and TAMs. For HN251, HN272 and HN279, the analysis showed similar trends in primary to pre-nodal malignant cells, with increasing interactions between the pre-nodal subpopulation and T-lymphocytes, specifically

with CD8+ cells (Fig. 6a). The analyses implicated a number of pathways that were differentially modulated by primary versus pre-nodal populations on T-lymphocytes (Supplementary Fig. 6a–c). In particular, the interaction between Midkine (*MDK*, secreted by tumor cells) and a number of MDK-receptors (*ITGA4, ITGA6, ITGB1, NCL, LRP1*) on CD8+ T cells appears to be a recurrent immunosuppressive pathway seen across all three patients (Fig. 6b). Applying the same approach to the external dataset also implicated the MDK pathway as being differentially activated by the pre-nodal population in one (p17) out of two tumors tested (Fig. 6b and Supplementary Fig. 6d, e).

**Fig. 3 | Functional analysis of actionable genes enriched in pre-metastatic population in patient-derived cultures (PDCs). a** Dimension reduction plots based on PAGODA for PDCs derived from matched primary and metastatic lymph nodes (nodal metastatic; M). Clusters are denoted by patient identity and site of origin (left), and Seurat clusters (right). *N* = 1317 cells. **b** Heatmap of differentially expressed pathways (rows) across samples and tumor origin (columns), showing selected Hallmark and Gene Ontology (GO) gene sets. Bars on the top of the heat map indicate the site of sample origins, clusters and patient samples corresponding to those of (**a**). **c** Boxplot showing the gene expression level of *AXL* (left, *n* = 353 cells) and *AURKB* (right, *n* = 318 cells) of malignant cells from primary and metastatic PDCs for the indicated patients. Boxplots centered at the median with hinges at 1st and 3rd quartiles and whiskers drawn from hinges to the lowest and highest points within 1.5 interquartile range; colors and cluster numbers of the bars correspond to (**a**). **d** Representative micrographs of immunocytochemistry of AXL in HN137 (*n* = 4) and AURKB in HN159 (*n* = 14) and HN220 (*n* = 12) of primary and metastatic PDCs from at least 2 independent experiment. Scale bar indicates 100 μm. **e–g** Representative micrographs from Boyden chamber assays of invaded cells (purple) (top), and quantification of invaded cells (bottom) from primary (*n* = 10 each; red) and metastatic (*n* = 10 each; blue) cell cultures treated with/without BGB324 or barasertib. At least 3 independent experiments were performed for each culture. Data are represented as mean ± SEM. **p ≤ 0.01; ***p ≤ 0.001; ****p ≤ 0.0001 indicate significant difference using unpaired two-tailed *t* test compared with untreated at corresponding site of origin, not corrected for multiple comparison. Scale bar indicates 100 μm. Source data are provided as a Source Data file. **h** Flow cytometry dot plots representing anti-AXL (left) and mouse IgG1 isotype control (right) staining of primary and metastatic PDCs of HN137. **i** Gating used for identification and isolation of AXL$^{hi}$, AXL$^{mid}$ and AXL$^{neg/low}$ from HN137 primary PDC by FACS sorting (left). Micrographs representing isolated AXL-based subpopulations treated with/without BGB324 and their respective invasive potential in Boyden chamber assays (right) from 2 independent experiment. Scale bar indicates 2500 μm.

Recent published data suggest that MDK-driven modulation is important for immune evasion in melanomas with activation of NFKB signaling cascade, which in turn can promote dysfunction in CD8+ T cells and negatively affect their anti-tumor cytotoxicity[36]. To test this in vitro, we treated cancer-cell suspension from tumors of three patients (HN372, HN377, HN380) with or without MDK inhibitor (Fig. 6c and Supplementary Fig. 6f–h). MDK inhibition reduced the frequency of non-T cells (CD45-CD3-) in all three tumors, and increased the percentage of CD8+ T cells in HN372, and to a lesser extent in HN377 and HN380. The frequency of cancer-cell (panCK+), as well as proliferating cancer cells (ki67+), was also reduced after MDK-inhibition in HN372 and HN380 (Fig. 6c and Supplementary Fig. 6f–h). The data suggests that in some tumors, MDK signaling suppress CD8-mediated anti-tumor activity and blockade of the signaling reverses this effect. Next, to investigate whether MDK-driven immune-suppression dampens the effect of immune checkpoint blockade (ICB) therapy, we developed a humanized mouse model engrafted with pre-nodal cells from the tumor culture of patient HN279 and treated the mice with PD1-blockade. We observed a slight reduction in tumor size at about 1 week after the start of the anti-PD1 treatment, and terminated the experiment one day after the final dose to examine the early events of anti-PD1 response within tumor by single-cell RNAseq (Supplementary Fig. 6i, j). As expected, the majority of HN279 cancer cells expressed *MDK* (Fig. 6d and Supplementary Fig. 6j–l), together with a number of genes associated with the pre-nodal phenotype (eg *SNAI2*, *AXL*, *STAT2*) that were unaffected by ICB (Fig. 6e and Supplementary Fig. 6m). In contrast, expression of *AURKB* and *TOP2A* (cell cycle genes) in cancer cells was significantly downregulated after pembrolizumab treatment (Fig. 6e), indicating a reduction in cancer-cell proliferation. In turn, the suppression of cell cycle in malignant cells upon anti-PD1 treatment resulted in a reduction of epithelial (EPCAM+) cells in the tumors, supporting the tumor size shrinkage observed on the anti-PD1 treated mice (Supplementary Fig. 6n).

Analyses of the CD8+ T-cell fraction revealed naïve, TRM, transitional, proliferative and dysfunctional/exhausted subpopulations, with an additional cytotoxic population (likely bystander) (Fig. 6f and Supplementary Fig. 6o). CD8+ cells from mice treated with pembrolizumab showed reduction in naïve, dysfunctional and memory with concomitant increase in proliferative, cytotoxic/bystander, tissue-resident subpopulations compared to untreated mice (Fig. 6g). These changes suggest a re-invigoration and reactivation of dysfunctional and memory, respectively, into tumor-targeting cells[35]. Remarkably, analyses of MDK receptor-expressing CD8 cells (*ITGA4*, *ITGB1*, NCL) showed the opposite trend, with an increase in dysfunctional and reduction in the proliferative (tumor-targeting) populations (Fig. 6h and Supplementary Fig. 6p). These findings suggest MDK-signaling promotes immune-suppression that abrogates re-invigoration by PD1-blockade. Indeed, these changes were also associated with *NFKB1* activation which is significantly higher in the dysfunctional CD8 population after pembrolizumab treatment (Fig. 6i). Moreover, plotting the expression levels of several MDK-receptors (*ITGA4*, *ITGB1*, NCL) with *NFKB1* show a good correlation in gene expression in CD8+ T cells where the RNA could be quantified (Fig. 6j). Taken together, these results could implicate MDK-signaling as a pathway through which pre-nodal cells evade CD8-mediated immune-editing.

## Discussion

Currently available algorithms analyzing single-cell data have the ability to construct evolutionary trajectories, which are especially powerful in studying specific events in space (eg. relationships between different tumor sites, primary vs lymph-node metastasis) and time (eg. pre- and post- treatment analysis)[12, 30]. Here, we applied these to explore early lymph-node metastasis across tumor and immune sub-compartments within the tumor. Analysis of tumor cells shows that nodal metastasis is an early event, where canonical epithelial-to-mesenchymal transition is less apparent than postulated. Our findings support previous studies that suggest EMT is not an all-or-none phenomenon, but instead occurs at graded levels[37, 38]. This contrasts with in vitro systems (including our own) where cultured tumor cells from lymph nodes display more canonical features of EMT[39]. Despite the overlap between tumor cells derived from primary and nodal sites, trajectory mapping could define evolutionary pathways at individual tumor levels, although this process requires a combination of trajectory algorithms, scoring for aggressiveness (based on EMT and stemness) and knowledge of the ground truth. Conversely, the complexity of these analyses highlights one of the limitations of these conclusions, which is examining a small number of tumors. An alternative hypothesis is that there is no coherent pattern and lymph-node metastasis is merely a passive drainage process that does not mark a biologically (and thus temporally) distinct phase of disease, but instead a marker of an aggressive primary tumor. Differentiating these requires a well-controlled system including animal models that could capture this evolutionary trait dynamically, and this would be an important extension to validate our findings. Nevertheless, we have expanded the results of previous studies in the identification of a pre-nodal or metastatic population[12], and importantly identified actionable drivers that could be targeted for anti-metastatic therapy, in this case AXL and AURK. While these have previously been identified and linked to metastases, our data provides compelling evidence that the contribution of these genes is limited to certain tumors and specific sub-populations within those tumors. Theoretically therefore, targeting AXL would not have the potential to prevent dissemination, but presumably reduces tumor heterogeneity by targeting the specific clones[40]. The role of aurora kinases is less clear. Rather than impacting the metastatic process, it is possible that this vulnerability reflects a

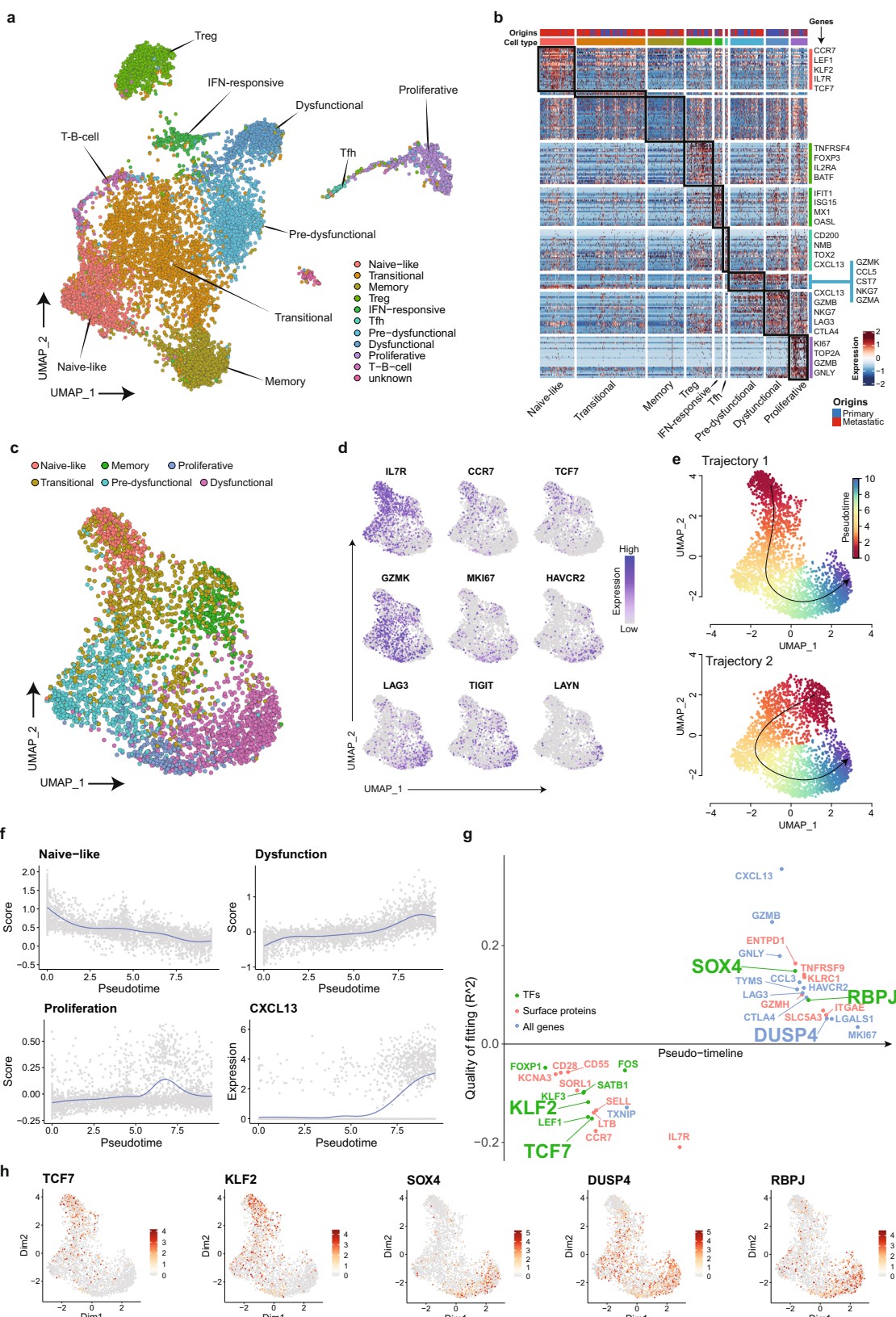

generalized reduction in cell cycling that occurs during EMT with a concomitant sensitivity to all cell cycle inhibitors. We recently demonstrated the same phenomenon during drug resistance: reduction in cell proliferation, limited AURK expression and sensitivity to inhibitors of AURK and other cell cycle targets[41]. Nevertheless, the ability to profile tumors and identify vulnerabilities in metastasis-

inducing clones is an attractive notion, with increasing interest in low-dose, long term anti-metastatic therapy.

Alignment of CD8+ T lymphocyte populations is driven by existing knowledge on T-cell developmental states. The fact that we could pool data across different patients increased the number of cells available and in itself was a form of validation. The alignment was

**Fig. 4 | scRNAseq analysis of tumor-infiltrating T cells and establishing a trajectory for tumor-targeting CD8+ lymphocytes. a** UMAP of tumor-infiltrating T cells from primary and metastatic tumors with clusters denoted by colors and labeled with inferred cell identities. N = 10,168 cells. **b** Heatmap of differentially expressed genes (rows) between cells classified into inferred T-cell subsets. Bars on the top of the heatmap indicate the site of origin and cell type corresponding to those of (**a**) with selected genes indicated. **c** UMAP of all 3387 CD8 T cells from primary and metastatic tumors. Clusters are denoted by colors and labeled with inferred cell identities based on (**d**) expression of selected genes used for CD8 T-cell subset annotation for. **e** Slingshot analysis of CD8 T cells showing two potential trajectories giving rise to tumor-targeting CD8+ cells: Trajectory 1 (top,

n = 2904 cells)−from naïve to dysfunctional and Trajectory 2 (bottom, n = 2863 cells)- memory to dysfunctional. **f** Graphs showing the estimate scores of curated genes related to naïve-like (*IL7R, TXNIP, SELL, CCR7, TCF7*), proliferative (*MKI67, HMGB2, TYMS*), dysfunctional (*GZMB, GNYL, CTLA4, LAYN, LAG3, TIGIT*) populations, and expression of *CXCL13* during the development of CD8 T-cell along the naïve-proliferation-dysfunction axis in Trajectory 1. N = 2904 cells. **g** Geneswitches output showing ordering of the top switching genes along the naïve to dysfunctional (Trajectory 1) CD8 T-cell axis using. Key genes are highlighted with enlarged font size. **h** UMAP projections of expression levels for genes highlighted in (**g**). N = 2904 cells.

further supported by single-cell VDJ sequencing, which reinforced trajectories from naïve or memory populations, towards clonally expanded, dysfunctional and potentially tumor-targeting CD8+ subpopulations. These supported both clonal replacement and revival models in HNSCC, where tumor-targeting T cells could be derived from both adjacent lymph nodes and tissue-resident CD8+ T cells. Remarkably though, our data also suggests that these putative tumor-targeting T cells were able to traffic in a bidirectional manner, although one of the limitations in this study is that these data neither precludes passive drainage of T cells across lymphatic channels nor collective migration of T cells along with tumor cells from one site to the other. Nevertheless, this trajectory could be used to identify modulators of T-cell dysfunction by studying gene expression changes along pseudotime, and was used to identify SOX4 as driver of dysfunction in CD8 cells. Several recent studies have implicated SOX4 in T-cell exhaustion in the context of pan-cancer single-cell analyses or associated with CAR-T cell dysfunction and exhaustion, lending support to our data[31, 32]. However it remains unclear as to whether SOX4 merely functions as transcription factor for a number of dysfunctional genes or actually regulates the exhaustion program, and this will require further investigation.

Interactome analyses performed to identify signaling networks within CD8+ T cells during early metastasis converged onto the MDK pathway. Remarkably, in a humanized mouse model, MDK signaling was associated with a reduced ability to reinvigorate exhausted T cells. This is supported by a recent publication which identified that the MDK pathway could abrogate immune reactivation by ICB therapy in melanoma, and this could be reversed using MDK-specific inhibitors[36]. In a similar context, MDK-inhibition could be explored in the prevention and treatment of tumor metastasis in HNSCC and add synergy to PD1-blockade which is the current standard of care in metastatic HNSCC.

Recent studies have further defined exhausted/dysfunctional CD8 T cells into four subsets, involving transcription factor TCF1/7 and surface receptor LY108 (SLAMF6 in human). In a stepwise development, tissue-resident LY108+CD69+(Tex^prog1) can interconvert with circulatory LY108+CD69− (Tex^prog2), while the latter can undergo a reversible proliferation-driven transition to circulatory LY108-CD69-effector-like (Tex^int), and eventually Tex^int can undergo an irreversible transition to tissue-resident LY108−CD69+(Tex^term) CD8 T cells[42, 43]. These exhausted subsets scatter across different anatomical sites, similar to the TCF7+CD8 T cells in our data although these predominate in the lymph-node, suggesting a large pool of TCF1/7+ progenitors. This reported interconvertibility of resident and circulating T cells likely accounts for the intertumoral clonotype sharing detected across naïve-like, memory, pre-dysfunctional and dysfunctional states, and even supports the potential for bidirectional movement of T cells between the primary tumor subsite and regional lymph-node (Fig. 5f). This supports a model of bifurcation among these phenotypes[30, 35], in which levels of inflammation and/or suppression may differ across anatomical locations within tumor, contributing to CD8 T cell fate decision. Nevertheless, T cell dysfunction is associated with the change of functionality rather than inactivity, as dysfunctional TILs still retain

cytotoxic ability, and their anti-tumor reactivity can be re-invigorated under appropriate conditions[35, 44–48].

There were a number of deficiencies in our study, not least of which the challenge to collect, process and analyze a larger cohort of paired tissue. This can be overcome by validation in external datasets, although we acknowledge that there are limited number of HNSCC tumor datasets that have been published to date. We also focused the analyses in this study to epithelial and T cells, while our data identified a range of different cell types in the primary tumor and lymph-node sites. One important cell type identified here was the cancer-associated fibroblasts (CAFs). These cells are considered to promote an immunosuppressive tumor microenvironment (TME) by regulating function and recruitment of immune cells, and transition to metastatic disease[49, 50]. CAFs can negatively influence TME through production of chemokines and cytokines to drive angiogenesis and cancer growth (eg. VEGF, TGFβ)[51–53], suppress CD8 T cell anti-tumor responses directly (eg. IL6, CXCL12, PDL2) and/or the recruitment of suppressive immune cells into the proximity[50, 54–56]. CAFs also contribute to the dysregulation of metabolic exchange between cells[57], as well as mechanical remodeling of extracellular matrix to facilitate tumor-cell migration and invasion[58, 59]. Although single-cell RNAseq can identify and study the transcriptome of CAFs, it lacks spatial resolution to show in situ CAFs interplay with other cell types (eg cancer cells and immune cells) which can be useful for the discovery of reliable biomarkers for therapeutic targets and risk-stratification of patients for treatment and prognosis.

In conclusion, we applied single-cell genomics, and specifically focused on trajectory and interactome analyses to uncover pathways and mechanisms that mediate early nodal metastasis in HNSCC. The data presented here shows that early metastasis is a much more nuanced process than previously presumed. Collectively these indicate the discovery potential of single-cell studies and existing computational tools, when applied to specific clinical contexts and questions. Future studies will focus on more specific tumor subpopulations including CD8+ cells and the impact of treatment on tumor recurrence and metastasis.

## Methods
### Ethics approval
This research complies with all relevant ethical regulations. The study of patient tumor samples was approved by SingHealth Centralized Institutional Review Board (CIRB: 2014/2093, 2018/2512 and 2016/2757) and each patient's written consent. Human cord blood samples were collected at the KK Women's and Children's Hospital, Singapore, and were performed in compliance with Institutional Review Board (CIRB Ref: 2013/778/D and 2019/2443/D) and each patient's parent(s) written consent for donation of cord blood for the generation of humanized mouse models. Umbilical cord blood (UCB) samples were collected from uncomplicated pregnancies at term and patients were recruited from planned Cesarean deliveries without any prejudice. All animal experiments were approved by the Institutional Animal Care and Use Committee of the Biological resource center (BRC), A*STAR, Singapore (IACUC numbers 161192, 191496).

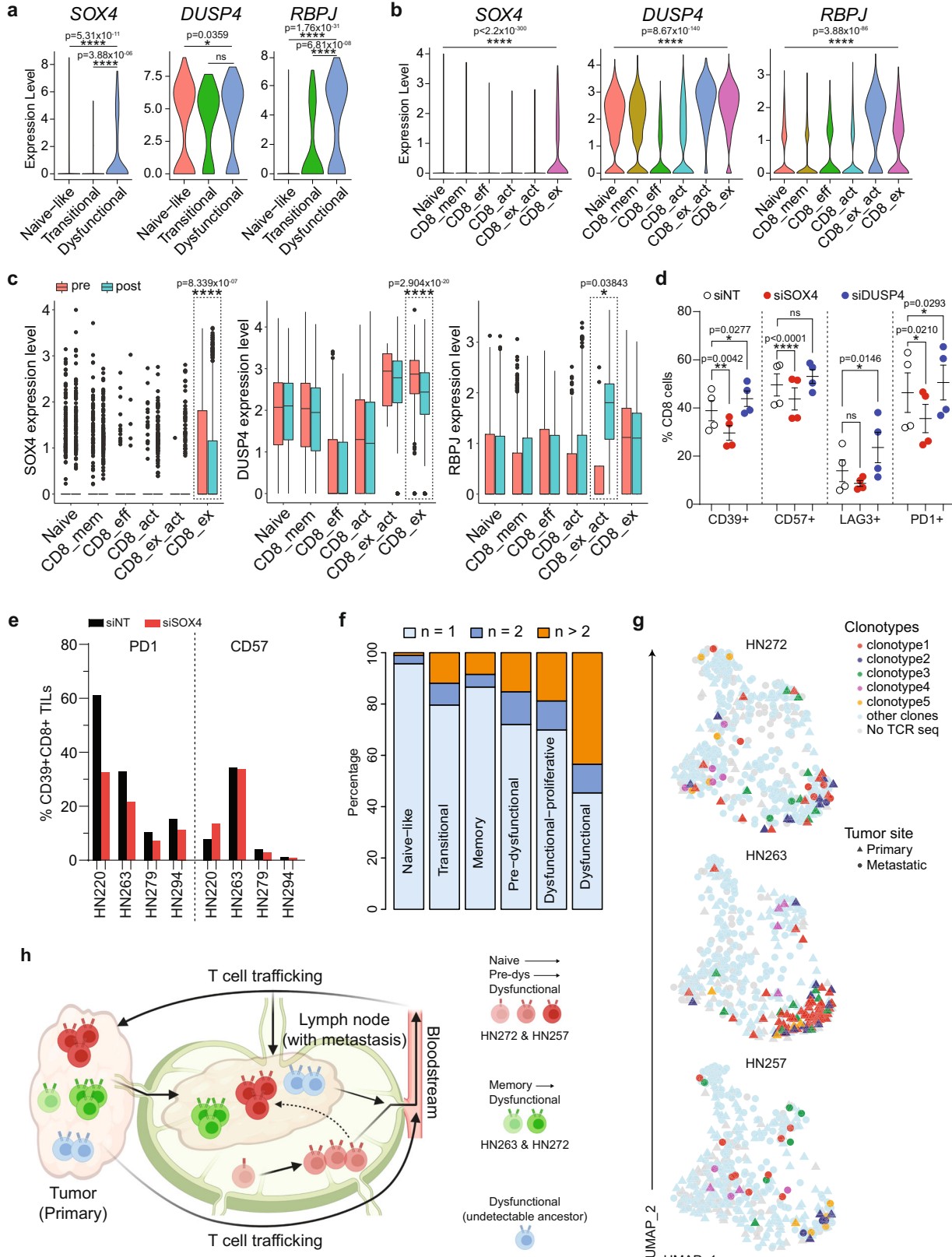

## Patient sample collection

For tumors, all patients were confirmed histologically to be HNSCC and suitable for surgical resection (with no prior cancer treatment). Patients included males and females, aged 21–85, as the information on sex and gender is not relevant in our study. Only primary lesion larger than a T2, with sufficient tissue for study without compromising pathological exam were included. Details of clinical and pathologic features are provided as Supplementary Data 1 and 2. Fresh tumors were collected from the primary site and metastatic draining lymph-node, and these arrived in the laboratory within 30 min upon resection in the operating theater.

**Fig. 5 | Functional analysis of genes involved in CD8 dysfunction and T-cell receptor sequencing analysis.** Violin plots showing expression of *SOX4*, *DUSP4* and *RBPJ* in CD8 T-cell subpopulations derived from published cohorts of scRNA-seq meta-dataset from (**a**) HNSCC (*n* = 542 cells from 11 patients) and (**b**) skin squamous-cell cancer (*n* = 17,561 cells from 11 patients)[12, 30]. P values are calculated by one-sided unpaired Wilcoxon test. **c** Boxplots showing expression of *SOX4*, *DUSP4* and *RBPJ* in CD8 T-cell subpopulations from (**b**), grouped by pre- (*n* = 6986 cells) and post-pembrolizumab (*n* = 10,575 cells) treatment. Boxplot represents median ± upper/lower quartile; whiskers represent 1.5 interquartile range; p values are calculated by two-sided unpaired Wilcoxon test, where *$p ≤ 0.05$; ****$p ≤ 0.0001$, ns not significant. Boxplots indicate quartiles with median at middle, and the whiskers drawn at the lowest and highest points within 1.5 interquartile range of the lower and upper quartiles, respectively. **b**, **c** X-axis labels: CD8_mem = CD8 memory; CD8_eff = CD8 effector; CD8_act = CD8 activated; CD8_ex_act = CD8 exhausted/activated; CD8_ex = CD8 exhausted. **d** Bar graph showing percentage of CD8 T cells expressing CD39, CD57, LAG3 or PD1 from PBMCs that were activated and cultured with siNT, siSOX4 or siDUSP4 for 5 days (*n* = 4). Black lines and error bars represent mean ± SEM. *$p ≤ 0.05$; **$p ≤ 0.01$; ****$p ≤ 0.0001$ indicate significant difference by paired two-tailed t-test compared with siNT of respective markers, ns not significant. **e** Frequency of CD39+CD8+ TILs expressing PD1 or CD57 that were activated and cultured with siNT or siSOX4 for 5 days (*n* = 4). **d**, **e** Source data are provided as a Source Data file. **f** Barplots of the percentage of TCR clone(s) detected once (*n* = 1399 cells), twice (*n* = 182 cells) or more than two times (*n* = 405 cells) across the CD8 T-cell subpopulations of all patients with HNSCC subjected to scRNAseq. **g** UMAP projection of CD8 T cells from HN272 (n = 377 cells), HN263 (*n* = 340 cells) and HN257 (*n* = 347 cells) colored by selected TCR clonotypes. **h** Schematic diagram summarizing the development and trafficking of CD8 T-cell clones between primary tumor, lymph-node and metastasis, and bloodstream of HN272, HN263 and HN257 based on the clonotype data from (**f**). Diagram was created with BioRender.com.

UCB were collected into EDTA tubes in a sterile field at birth. Sample was kept on ice, transported to the laboratory and processed within 2 h. CD34+ hematopoietic stem cells (HSC) were enriched using the CD34 MicroBead Kit UltraPure (Miltenyi Biotech, Bergisch Gladbach, Germany), and each enriched sample was passed through a second column, according to the manufacturer protocol. The average purity of the enriched CD34+ HSC was 95% as examined by flow cytometry. The enriched CD34+ HSC was immediately cryopreserved using serum-free cell freezing medium Bambanker (GC Lymphotec, Japan) in liquid nitrogen for storage and thawed just before use.

## Tumor sample dissociation

Tumors were minced, placed into C Tubes and digested using a human tumor dissociation kit and gentleMACS™ Octo system (all from Miltenyi Biotech, Bergisch Gladbach, Germany) as described in manufacturer's protocol. After digestion, single-cell suspensions were passed through 70 μm cell strainers (Sigma-Aldrich, St. Louis, MO), washed with sterile PBS and pelleted by centrifugation at $300 × g × 5$ min. Cells were resuspended in appropriate medium and cell count was performed using The Countess™ Automated Cell Counter (Invitrogen, Carlsbad, CA) with trypan blue for dead cell exclusion.

## Cell enrichment by magnetic separation

For patient tumors, up to $1 × 10^7$ of tumor cells were magnetically labeled with anti-CD45 microbeads (Miltenyi Biotec, Bergisch Gladbach, Germany) for 20 min at 4 °C in MACS buffer (0.5% BSA and 2 mM EDTA in PBS). Following incubation, cells were washed once with MACS buffer, filtered through 40 μm filter mesh (Sigma-Aldrich, St. Louis, MO), pelleted and resuspended in 500 μL of MACS buffer before loading onto the MS column and magnetic stand for cell separation as described in the manufacturer protocol. After the separation, cells were washed, pelleted and resuspended in appropriate medium for downstream experiments. The numbers of enriched CD45+ and CD45− cells were adjusted to 1:1 ratio before proceeding to droplet-based single-cell capturing.

For humanized mouse tumors, up to $1 × 10^7$ of tumor cells were magnetically labeled with anti-CD8 microbeads (Miltenyi Biotec, Bergisch Gladbach, Germany), incubated, washed and loaded onto the MS column as described above. After the separation, the CD8+ fraction was washed, pelleted and resuspended in appropriate medium for downstream experiments. The CD8− fraction was enriched for tumor cells using a Tumor Cell Isolation Kit, human (Miltenyi Biotec, Bergisch Gladbach, Germany). The numbers of enriched CD8+ and tumor cells were adjusted to 1:1 ratio before proceeding to droplet-based single-cell capturing.

## Patient-derived cell cultures

Unsorted or CD45- sorted tumor suspensions derived from digested tumor samples (see above) were seeded onto wells of Corning CellBIND plates (Corning, New York City, NY) during first two weeks. All patient-derived tumor-cell cultures were passaged until a tumor majority was observed, and well characterized previously[20, 21]. Cells were maintained in complete RPMI (C/RPMI) containing 10% FBS, 1% pen-strep, 1% antimycotic and a humidified incubator at 37 °C with 5% $CO_2$. Cell culture identity was authenticated by comparing the STR profile (Indexx BioResearch), mutational and/or expression profile of each cell line to its original tumor. All cultures were tested and confirmed to be free of mycoplasma using an EZ-PCR Mycoplasma Detection Kit (Biological Industries, Kibbutz Beit Haemek, Israel) at the time of experiments.

## Xenograft tumor model in humanized mice and anti-PD1 treatment

NOG-EXL (hGM-CSF/hIL-3 NOG) female mice (*n* = 8), pre-engrafted with human CD34+ hematopoietic stem cells enriched from cord blood of two de-identified donors (KKH88/KKH92) in the KK Women's and Children's Hospital, Singapore (see above), were procured from CIEA-SIgN. As the information on sex and gender was not relevant to our study, only female mice were available and used for the experiment. At 16 weeks post-engraftment and when human chimerism in blood was approximately 40% or more, mice were injected subcutaneously at both flanks with 5 million cells of primary tumor cultures derived from patient HN279. Eight days after tumor inoculation, tumor growth kinetics was closely monitored by measuring tumor size at every alternate day using calipers, and mice were shaved for clearer visualization of tumors when necessary. At Day 16 of tumor implantation, mice were randomly divided into two groups of 8 mice each: Control group and anti-PD1 treated group. Pembrolizumab (Keytruda; Merck, Kenilworth, NJ), a monoclonal antibody against human PD1, was administered intraperitoneally (i.p.) at 12.5 mg/kg on day 28, 30, 32 and 34 to mice in the treated group, while those in the control group were administered with phosphate buffered saline (PBS). Mice were monitored for signs of treatment-related toxicity twice weekly and tumor size was not allowed to exceed 1.5 cm in diameter. One day after the last antibody administration, mice were euthanized by $CO_2$ inhalation followed by cervical dislocation, and tumors were harvested for dissociation and preparation for single-cell capture and single-cell RNAseq as described in other sections of Methods. Mice were housed in a 12 light/12 dark cycle at approximately 18–23 °C with 40-60% humidity.

## Histology

Approximately 30,000-40,000 cells were deposited onto glass slides using Shandon Cytofunnels and a Cytospin 4 Cytocentrifuge instrument (both from Thermofisher, Waltham, MA). Subsequently, these cells were fixed in 4% paraformaldehyde and blocked with 5% normal goat serum prior to immunostaining with primary antibodies against

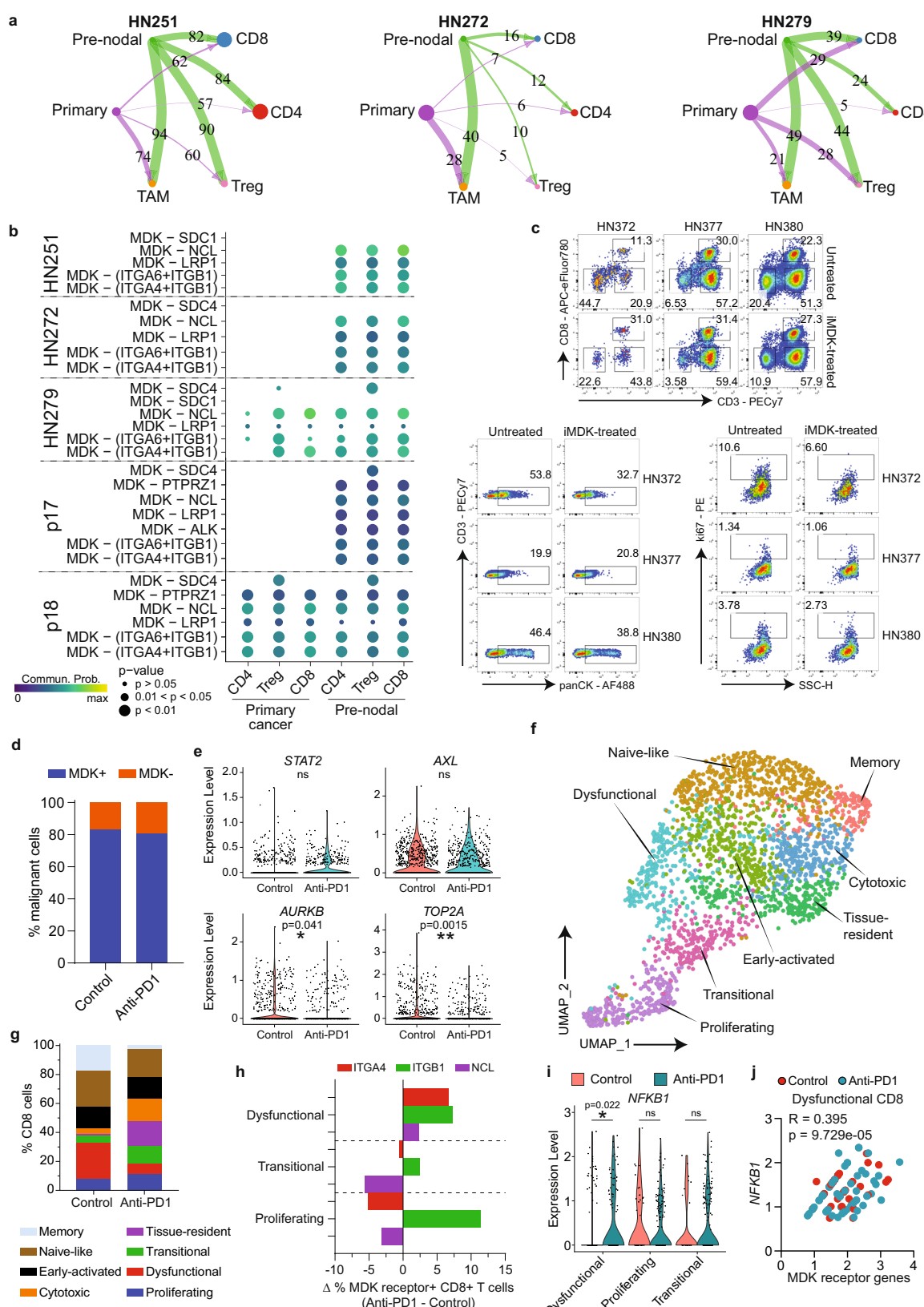

AXL (clone C89E7; CST, Danvers, MA; 1:200 dilution) or AURKB (clone RM278; Invitrogen, Carlsbad, CA; 1:200 dilution) at 4°C overnight. For AXL staining, slides were washed and stained with Dako REAL™ EnVision™ Detection System (Dako, Santa Clara, CA). Slides were then counterstained with hematoxylin, dehydrated and mounted with coverslip using DPX mountant (Sigma-Aldrich, St. Louis, MO). For

AURKB staining, slides were washed with PBS and incubated with AF488-conjugated goat anti-rabbit secondary antibody (#A11008; Invitrogen; 1:200 dilution) at room temperature for 1 h. After washing with PBS, the cells were then stained with DAPI for 5 mins, washed and mounted with coverslip using a DAKO mounting reagent. Immuno-histochemistry images were take using a Nikon Eclipse 80i microscope

**Fig. 6 | Determining interactions between pre-metastatic malignant cells and CD8+ T-lymphocytes. a** Hierarchical plot from Cellchat analyses showing ligand-receptor interactions between tumor subpopulations (primary/pre-nodal) with T-cell subsets and TAMs. Circle sizes are proportional to number of cells in each cell group, and edge width represents communication probability with putative ligand-receptor pairs as indicated. **b** Dot plots showing significant MDK ligand-receptor pairs contributing to signaling. Dot color and size represent the calculated communication probability, and *p* values determined from one-sided permutation test. **c** Flow cytometry of cancer cells isolated from HN372, HN377 and HN380 tumors treated with/without MDK-inhibitors (iMDK). (Top) Percentage of live CD3−CD8−, CD3+CD8− and CD3+CD8+ Tcells gated on CD45+ cells, (bottom left) panCK+CD3-cancer cells gated on CD45- cells and (bottom right) ki67+ cancer cells gated on CD45-panCK+ cells. Numbers within each plot indicate percentage. PatientID and treatment conditions (untreated/iMDK-treated) are indicated on top or adjacent to the plots. **d** Frequency of MDK + (blue; *n* = 969 cells) and MDK- (orange; *n* = 210 cells) malignant cells in control or anti-PD1-treated mice (*n* = 4 mice each). **e** Expression level of selected genes involved in tumor-cell proliferation from control or anti-PD1-treated mice (*n* = 4 mice each, 1179 cells). **f** UMAP of tumor-infiltrating CD8 T cells extracted from humanized NOG-EXL mice treated with/without anti-PD1 (*n* = 4 mice each; 2342 cells) (see Supplementary Fig. 6i). Clusters denoted by colors labeled with inferred cell identities. **g** Distribution of CD8 T-cell subpopulations in control vs anti-PD1-treated mice (*n* = 4 each). **h** Delta (Δ) percentage of CD8 T cells expressing the specific MDK-receptors *ITGA4*, *ITGB1* or *NCL* in dysfunctional, transitional and proliferating subpopulations, comparing untreated versus anti-PD1-treated mice (*n* = 4 each). Delta percentage is determined by the percentage of MDK receptor+ CD8+ T cells from anti-PD1-treated mice minus control mice (*n* = 4 each). **i** *NFKB1* expression in the three CD8 subpopulations in controls and anti-PD1-treated mice (*n* = 4 mice each, 760 cells). **j** Scatterplot showing the correlation of expression between *NFKB1* with the following MDK receptor(s): *ITGA4*, *ITGB1* and/or *NCL* in dysfunctional CD8 T cells subpopulations. Each dot represents one dysfunctional CD8 T-cell from control (red; *n* = 35 cells) or anti-PD1-treated (blue; *n* = 58 cells) mice. R and two-sided p values determined using Pearson correlation statistical analysis. **p* ≤ 0.05; ***p* ≤ 0.01 indicate significant difference, ns not significant. **d, i** Data analyzed by unpaired two-tailed t test (**d, g, h, j**) Source data provided as a Source Data file.

---

with DS-Ri2 camera (Nikon, Tokyo, Japan) at 20x optical magnification. Immunofluorescence images were obtained using a Vectra 3 pathology imaging system microscope and quantified using inForm software (version 2.4.2; both by PerkinElmer, Inc., Waltham, MA).

### Invasion assay
25,000 single cells were resuspended in serum-free RPMI with or without 0.25 μM of bemcentinib (BGB324) or 0.25 μM of barasertib (both from Selleck Chem, Houston, TX). Next, cells were seeded on top of a 8 μm filter membrane within a 24-well transwell insert (Corning, New York City, NY). C/RPMI was added to the bottom of wells of 24-well Falcon TC Companion Plate (Corning, New York City, NY). After 72hrs, inserts were removed, washed with PBS and the upper surface of the membrane was scrubbed twice using cotton tipped swab to remove non-invading cells. To quantify invaded cells, the exterior bottom of each insert was fixed and stained with 25% methanol and 0.5% crystal violet solution for 5 mins. After washing of the inserts with excess distilled water and air-drying, the membranes were carefully removed using scalpel blades and mounted onto glass slides with coverslips. Slides were visualized and digital images recorded using a Nikon Eclipse 80i microscope with DS-Ri2 camera (Nikon, Tokyo, Japan) at 20x optical magnification. Cell invasion area was determined by quantifying the area with crystal violet staining using the ImageJ software.

### In vitro tumor-infiltrating lymphocytes expansion
Tumor-infiltrating lymphocytes (TILs) were initiated in 24-well plates (Corning, New York City, NY) with each well containing one tumor fragment (1-8mm³) in 1 ml of TIL-CM and 6000 IU/ml of recombinant human interleukin-2 (IL-2; Proleukin, Clinigen Healthcare Ltd, Staffordshire, England)[44, 60]. TIL-CM comprises RPMI 1640 with GlutaMAX supplemented with 2 mM HEPES, 10ug/ml gentamicin and 1% Antibiotic-Antimycotic (all from Thermofisher, Waltham, MA) and 10% human AB serum (Sigma-Aldrich, St. Louis, MO). After 5 days of initiation, half the media was removed and refreshed with TIL-CM and IL-2. Subsequently, half the media was exchanged every 2–4 days. After 14-18 days, 1 to 2 million TILs were transferred to T75 flasks containing irradiated (50 Gy) allogenic PBMC feeders at a ratio 1 to 200 for the Rapid Expansion Protocol (REP). Cells were cultured with a mixture of TIL-CM and AIM V (50/50) media, 6000 IU/ml of IL-2 and 30 ng/ml of anti-CD3 antibody. After 5 days, half the media was exchanged with fresh 50/50 media containing 6000 IU/ml of IL-2. Subsequently, half the media was replaced and refreshed with AIM V supplemented with 5% AB serum and 6000 IU/ml of IL-2 every 2–4 days. After 14-18 days in REP, these TILs (termed REP-TILs) were then used for further experiments.

### Small-interfering RNA knockdown of SOX4 and DUSP4
Peripheral blood mononuclear cells (PBMCs) from healthy donors or REP-TILs were cultured at a density of 0.2–1 × 10⁶ cells/ml in 24-well plate (Corning, New York City, NY), containing TexMACS Medium and T Cell TransAct (both from Miltenyi Biotech, Bergisch Gladbach, Germany) at 1:200 dilution. A final concentration of 1 μM of Accell pooled small-interfering RNA (siRNA) targeting human SOX4 (Gene ID 6659) or DUSP4 (Gene ID 1846), or non-targeting siRNA (all from Dharmacon, Lafayette, CO) was added into respective wells. After 5 days of incubation, cells were harvested, stained with fluorochrome conjugated antibodies and analysed by flow cytometry. To access knockdown efficiency, after 24 h of treatment with siRNAs, cells were harvested for RNA extraction using a Qiagen RNeasy Mini kit with on-column DNA removal (Qiagen, Valencia, CA) and reversed transcribed using a SuperScript II Reverse Transcriptase kit (Invitrogen, Carlsbad, CA), as described in the manufacturer protocol. Real-time PCR was performed using iTaq Universal SYBR Green Supermix and a CFX96 Touch Real-Time PCR machine (both from Bio-Rad Laboratories, Hercules, CA), according to the manufacturer protocol. Primers for SOX4-Fwd: 5'-GGT CTC TAG TTC TTG CAC GCT C-3'; SOX4-Rev: 5'-CGG AAT CGG CAC TAA GGA G-3'.

### Midkine inhibition assay
Single-cell suspension from dissociated tumors were seeded at a density of 0.3–0.5 × 10⁶ cells/ml containing C/RPMI in a 24-well plate. Small molecule 3-[2-(4-fluoro-benzyl)imidazo[2,1-b][1,3] (MDKi; Calbiochem, San Diego, CA) were added into wells at a final concentration of 100 nM. After 5 days, suspension cells were first harvested, and adherent cells were trypsinized and harvested for antibody staining and flow cytometry analysis as per below.

### Flow cytometry
For AXL surface staining, trypsinized cells were stained with fluorochrome conjugated antibody recognizing AXL (#108724; R&D systems; 1:25 dilution) or with isotype IgG1 antibody (MOPC-21; BD Biosciences, Franklin Lakes, NJ; 1:200 dilution). For intracellular AURKB staining, trypsinized cells were fixed and permeabilized with a Foxp3/Transcription Factor Staining Buffer Set (eBioscience, San Diego, CA) according to the manufacturer protocol. After fixation, cells were stained with primary antibody recognizing AURKB (clone RM278; Invitrogen, Waltham, MA; 1:200 dilution) or rabbit IgG1 isotype antibody (DA1E; R&D systems, Minneapolis, MN; 1:500 dilution), and subsequently with goat anti-rabbit IgG secondary antibody conjugated to Alex Fluor 647 (#A32733; Invitrogen; 1:200 dilution). For Midkine inhibitor assay, harvested cells (as above) were surface stained with fluorochrome conjugated antibodies (1:50 dilution each) recognizing

CD45 (HI30) from Biolegend, San Diego, CA; CD8 (HIT8a) from eBioscience; CD3 (OKT3) from Biolegend or BD Biosciences. Then, cells were fixed and permeabilized and intracellularly stained for antibodies recognizing ki67 (B56; BD Biosciences; 1:5 dilution) and pan-CK (AE1/AE3; eBioscience; 1:100 dilution). For siRNA knockdown experiment, harvested cells were stained with fluorochrome conjugated antibodies (1:50 dilution each) recognizing CD57 (HNK-1), LAG3 (11C3C65), CD39 (A1) and CD4 (OKT4) all from Biolegend; PD1 (J105) and CD8 (SK1) from eBioscience; and CD4 (SK3) from BD Biosciences. These cells were stained for 30 min on ice in the dark with 2% BSA in PBS. Live and dead cells were distinguished using a Fixable Viability Dye eFluor 506 (eBioscience; 1:200 dilution). For EPCAM detection, tumors harvested from humanized mice were dissociated into single-cell suspensions as described above, and stained with antibodies recognizing human CD45 (HI30) and mouse CD45 (30-F11) both from eBioscience and at 1:50 dilution each; and human EPCAM (HEA-125; Miltenyi Biotec, Bergisch Gladbach, Germany; 1:50 dilution), and dead cells were excluded using DAPI (#4220801; Biolegend). Cells were acquired using a BD FACSCanto II or LSRFortressa instrument and analysed using FlowJo v10.5.3 software (both from BD Biosciences). Cell sorting was performed using a BD FACSAria III instrument (BD Biosciences).

### Generation of single-cell gene expression libraries by microfluidic-based technology
Patient-derived cell lines were trypsinized into single-cell suspensions, loaded and captured using medium-sized (10-17um) Fluidigm Integrated Fluidic Circuit (IFC) and a Fluidigm C1 instrument (Fluidigm, South San Francisco, CA), as described in the manufacturer's protocol. Each well in the IFC was visualized and cataloged for single-cell capture, prior to reloading into the C1 instrument for lysis, reverse transcription and cDNA synthesis. The cDNA product was harvested from the IFC, barcoded for individual cell identity, pooled and cleaned for next generation sequencing. Libraries were sequenced by an Illumina Hiseq 4000 (Illumina, San Diego, CA) with 151-bp single-ended or pair-ended reads.

### Generation of single-cell gene expression and TCR libraries by droplet-based technology
The 5′ gene expression (GEX) and TCR single-cell RNA libraries were prepared using the 10x Chromium Single-Cell V(D)J Reagent Kits (10x Genomics, Pleasanton, CA), as described in the manufacturer's protocol. Briefly, freshly dissociated patient tumor cells were sorted into CD45+ and CD45- fractions, mixed at a 1:1 ratio, washed and resuspended with PBS 0.04% BSA to a final concentration of 500-1200 cells/μl with cell viability of more than 85%. For humanized mice, dissociated tumor cells were mixed at 1:1 ratio with enriched CD8+ and tumor cells. Subsequently, cells were loaded into the Single-Cell A Chip for gel bead-in-emulsion (GEM) generation and barcoding, targeting for a cell recovery of 4000-7000 cells per sample. Next, reverse transcription occurred within the GEM, and the GEM was broken, and cDNA purified using Dynabeads MyOne SILANE (Thermofisher, Waltham, MA). cDNA was amplified to construct a 5′ library which was used to build the GEX and TCR libraries. For GEX library construction, 2−50 ng of amplified cDNA was fragmented, end repaired and size-selected using SPRIselect Reagent (Beckman Coulter, Brea, California), and sample index PCR was performed. For TCR library construction, 2 μl of amplified cDNA was first enriched for V(D)J sequence using a human T Cell V(D)J Enrichment Kit. Subsequently, 2−50 ng of the enriched transcripts was fragmented, end repaired, sample indexed and size-selected using SPRIselect Reagent (Beckman Coulter, Brea, California). Libraries were sequenced using an Illumina Hiseq 4000 (Illumina, San Diego, CA) with 151-bp pair-ended reads.

### Data processing of single-cell RNA-seq libraries and clustering
The scRNA-seq reads were aligned to the GRCh38 reference genome and quantified using cellranger count version 2.2.0 (10x Genomics, Pleasanton, CA). Downstream analyses were performed using Seurat version 3.1.5[61]. For human samples, cells with (i) greater than 20% mitochondrial RNA content, (ii) less than 200 genes detected, or (iii) greater than 40,000 UMI and 6000 genes detected were excluded from analysis. For NOG.EXL mouse samples, cells with (i) greater than 10% mitochondrial RNA content, (ii) less than 100 genes detected, or (iii) greater than 7000 genes detected were excluded from analysis.

For clustering of all cell types, Seurat alignment across patients was applied. The Seurat object with all the cells was first split by patient ID, then for each patient's object, raw UMI counts were lognormalized and variable genes were called independently based on average expression > 0.1 and average dispersion >1. Sets of anchors across patient's objects were identified using parameters CCA *dims = 1:30* and number of neighbors *k.filter = 200*, and followed by the integration. Scaled z-scores for each gene were then calculated using the *ScaleData* function and regressed against the number of UMIs per cell and mitochondrial RNA content. Scaled data were used input into a principal component analysis (PCA) on the basis of variable genes. Clusters were identified using shared nearest neighbor (SNN)-based clustering on the basis of the top 83 significant principal components determined by *JackStraw* function, with *resolution = 0.6* which is determined by *clustree* output. The same principal components were used to generate the UMAP projections, which were generated with a minimum distance of 0.3 and 30 neighbors. Differentially expressed (DE) genes for each of the identity clusters were generated using *FindAllMarkers* with *min.pct = 0.25* and *logfc.threshold = 0.25*. Cell types were annotated using the resulted DE genes together with the expression of known marker genes.

For malignant-cell clustering, we isolated subsets of cells from the complete dataset that were identified as malignant cells based on broad clustering. Cells were then reclustered using Seurat without patient alignment, since tumor cells tend to be patient specific. Raw UMI counts were normalized using regularized negative binomial regression via function *SCTransform*, where cellular sequencing depth is utilized as a covariate in a generalized linear model. The number of UMIs per cell and mitochondrial RNA content were regressed out in a second non-regularized linear regression. Malignant-cell clusters were identified using SNN-based clustering based on the first 15 principal components with *resolution = 0.5*. To assign the epithelial–mesenchymal transition score, we used the *AddModuleScore* function based on genes annotated with the GSEA MSigDB geneset "HALLMARK_EPITHELIAL_MESENCHYMAL_TRANSITION".

For T-cell clustering, we first isolated subsets of cells from the complete dataset that were identified as T cells based on broad clustering. We removed specific immunoglobulin genes, mitochondrial genes, genes linked with poorly supported transcriptional models (annotated with the prefix "RP-") and ribosomal proteins related genes (annotated with the prefix "RPL-" and "RPS-"). Ribosomal RNAs were removed to allow accurate and efficient downstream analyses. Cells were then reclustered using Seurat alignment across patients similar as the previous analysis. Sets of anchors across patient's objects were identified using parameters CCA *dims = 1:30* and number of neighbors *k.filter = 70*. T cell clusters were identified using SNN-based clustering based on the first 30 principal components with *resolution = 0.8*. For UMAP visualization, we used the same principal components, a minimum distance of 0.05 and 30 neighbors. DE genes for each of the identity clusters were generated using *FindAllMarkers* with *min.pct = 0.2* and *logfc.threshold = 0.2*. Based on the resulted DE genes and the expression of known marker genes, some clusters identified as B-cells were excluded from the downstream analysis.

For CD8+ T-cell clustering, CD8+ T cells were first extracted from the T-cell clustering based on the following two criteria: (1) in Pre-

dysfunctional, Dysfunctional and Proliferative clusters, and with zero CD4 expression, (2) in Naïve-like, Memory and Transitional clusters, with zero CD4 and positive CD8 (either CD8A or CD8B) expression. We assigned G1, G2/M and S cell cycle phases to CD8+ T cells using the *CellCycleScoring* function. In order to remove such cell cycle-driven artefacts, cells were reclustered using Seurat alignment across patients and cell cycle phases. Some cell groups, such as G2/M cells of patient HN237, with less than 50 cells were removed before alignment. Sets of anchors across cell groups were identified using parameters CCA *dims = 1:20* and number of neighbors *k.filter = 50*, and followed by the integration. Clusters were identified using SNN-based clustering based on the first 60 principal components with *resolution = 0.8*. For UMAP visualization, we used the same principal components, a minimum distance of 0.05 and 30 neighbors.

### Data processing of single-cell TCR-seq libraries

TCR reads were mapped to vdj_GRCh38_alts_ensembl-3.1.0-3.1.0 reference genome and quantified using cellranger count version 3.1.0 (10x Genomics, Pleasanton, CA). In total, 60% of annotated CD8+ T cells were assigned a TCR and only 3.2% of cells not annotated as T cells were assigned a TCR. Overall, productive TCR-alpha and TCR-beta sequences were recovered from 1461 and 1948 CD8+ T cells. 3.3% of CD8+ T cells with TCR reads were assigned only TRA sequences and 27.5% of T cells with TCR reads were assigned only TRB sequences. 1,590 unique TCR sequences, and clonotype sizes ranged from 1 cell to 60 cells were identified. Clonotypes were defined either as expanded (i.e., detected in at least two cells) or unique (i.e., detected in no more than one cell).

### Inferring copy number alterations from scRNA-Seq data

InferCNV (v1.2.2) (https://github.com/broadinstitute/inferCNV) was used to identify somatic large-scale chromosomal copy number alterations using single-cell gene expression data. InferCNV was applied individually to each patient with immune cells, fibroblasts and endothelial cells as the reference "normal" group. Gains or deletions of large segments of chromosomes were observed in the epithelial cell population of each patient.

### Comparison of cellular components by tumor site

TCGA RNA-seq and clinical annotation data was downloaded from the GDC Data Portal [https://portal.gdc.cancer.gov/]. The RNA-seq profiles were deconvolved using xCell [https://portal.gdc.cancer.gov/] to estimate immune and stromal cell components. Tumor site data was extracted from clinical annotation and sites with fewer than 10 tumors were removed from analysis. Associations between cell component and site were assessed by one-way ANOVA tests.

### Labeling of tumor cells using InferCNV and CopyKAT information

CopyKAT v1.0.8 was used to infer copy number profiles and assign with or without copy number alterations (CNAs) labels to each cell. Preprocessed single-cell RNA-seq counts were given as input with default parameters. Some cell profiles failed to meet the CopyKAT's internal quality thresholds and were filtered out. CopyKAT predictions are also conservative, as some epithelials it predicted to be normal had CNAs characteristic of HNSCC.

### Identification of pre-metastatic subpopulation features in primary tumors

For the trajectory analysis in Fig. 2C, branches and states were detected using the *monocle* v2.16.0R package. Genes that define the trajectory were selected using the *differentialGeneTest* function and with q-values <0.05. Dimensional reduction was performed using the *DDRTree* method and states were identified using the *orderCells* function. EMT score of each cell calculated previously was plotted as boxplot with group of states. *CytoTRACE* v0.3.2R package was applied to predict the differentiation score and plotted on the monocle trajectories. We hypothesized that de-differentiation co-occurs with the metastatic phenotype. Hence, the monocle state with low EMT score and more differentiated score inferred by CytoTRACE was selected as the root. This state was then indicated in the monocle function *orderCells* to generate a pseudo-time. Based on the derived CytoTRACE differential score, primary cells of each patient were divided into two groups with high score (top 25% less differentiated) and low score. The cell group of high score were considered as pre-metastatic subpopulation in the primary tumor.

In order to determine the significant genes that regulate the trajectory from primary to pre-metastatic cells, GeneSwitches v0.1.0R package was applied[28]. GeneSwitches is a statistical framework based on logistic regression to find the set of genes that switch during the transition. Cells from the primary tumor site were first extracted from the specific trajectories. Then the corresponding single-cell log-transformed gene expression and monocle pseudo-time were input into GeneSwitches. Function *binarize_exp* with fixed cutoff 0.2 was used to binarize the gene expression into on or off states. For each gene, *find_switch_logistic_fastglm* function calculated a switching time and associated confident level. Top 70 genes of high confident levels, including surface proteins and transcription factors, were plotted using function *plot_timeline_ggplot* to visualize the switching orders and quality (Supplementary Fig. 2A, B). For gene ontologies analysis, genes of confident level above 0.01 were input into function *find_switch_pathway* with default parameters.

### Identification of aggressive sub-clone in HN257 primary sample and mapping to TCGA

Cells from the primary site of patient HN257 were extracted and applied to *CytoTRACE* as described above. Based on the derived differential score, cells were divided into two groups with high score (top 30% less differentiated) and low score. The cell group with high score was considered as an aggressive subpopulation in the primary tumor. DE genes between these two groups were determined using Seurat function *FindMarkers* with default parameters. We identified a panel of 132 genes that were over- and 45 genes that were under-expressed in the aggressive subpopulation with adjusted p value <0.05 and absolute fold change (nature log) > 0.3. Geneset enrichment analysis was then conducted on this panel of genes using GSEA 4.1.0 app package downloaded from GSEA website (http://www.gsea-msigdb.org/gsea/index.jsp).

To apply this panel of genes to TCGA, we collected the HNSCC transcriptome profiling gene expression files with FPKM values and related clinical parameters from TCGA portal (https://cancergenome.nih.gov). We focused on data from primary tumors and 5-years survival information which resulted in a total of 477 patients. The panel of 132 upregulated feature genes were extracted from the gene expression matrix of TCGA samples. Then, based on the mean threshold of these feature genes, 477 TCGA samples were divided into two sub-classes (high features class and low features class). For survival analysis, we use R package *survival* to generate the Kaplan-Meier curves for these two classes of patients, showing that tumors that were similar to the aggressive HN257 sub-clone (high features class) had significantly poorer outcomes.

### Data processing of microfluidic-based single-cell RNA-seq libraries and clustering

The raw reads in FASTQ files were aligned to the human genome (hg19 assembly) using STAR v2.6.0, followed by quantification of gene expression using RSEM v1.3.0 to generate TPM values. Cells with (i) less than 100,000 raw counts, (ii) less than 10% unique exonic mapping rate, or (iii) less than 1000 expressed genes (TPM > 1) were excluded from analysis.

For gene ontologies analysis and clustering, we applied pathway and geneset overdispersion analysis (PAGODA) using scde v1.99.2R package. The algorithm allows identification of the most over-dispersed gene sets among the input ones including MSigDB hallmark[62] and C5 gene ontology geneset collections. PAGODA calculated weighted first and second principal component magnitudes for each geneset (or aspect) and evaluate the statistical significance of the observed overdispersion compared to the background expectation. For clustering, cells were grouped based on a weighted correlation of genes that drive the significant aspects. The same correlation from the clustering were used to visualize cells in two dimensions using t-SNE plots. In order to reduce redundancy, gene sets that are driven by the same sets of genes or showing similar expression patterns were integrated into aspects using a distance threshold of 0.5. Top significant merged aspects were then plotted in heatmap.

We observed that primary cells of patient HN137 were separated into two clusters, and one of them shared the similar expression pattern of aspects with metastatic cells (Fig. 3b). In addition, we also observed that a small primary subpopulation clustered together with metastatic cells for patient HN159 and HN220. Hence, we applied nonparametric differential expression for single-cells (NODES, https://www.biorxiv.org/content/biorxiv/early/2016/04/22/049734.full.pdf) to identify the gene features for the pre-metastatic subpopulation in these patients.

### Trajectory analysis of CD8+ T cells
We performed trajectory analyses of the CD8+ T cells using Slingshot v1.4.0R package. Naïve-like and memory population were selected as two starting points to infer the trajectories. We applied GeneSwitches to the trajectory from naïve-like to dysfunctional cluster (trajectory 1) as described above.

The naïve-like score was generated using the *AddModuleScore* function based on the naïve marker IL7R and its top 30 co-expressed genes (highest pearson correlation). Similarly, the dysfunction score was based on the dysfunctional marker TIGIT and its top 30 co-expressed genes. Lastly, the proliferation score was determined by the average of S and G2M score calculated previously.

### Validation using published scRNAseq dataset
*Validation of AXL and AURK expression*. We extracted cells that were annotated as cancer cells for each patient in Puram et al.[12]. Five patients were obtained cells from both primary site and lymph-node. Our approach to identify the pre-nodal subpopulation features in primary tumors were applied to these five patients. Due to the limitation of low number of cells, we were not able to apply GeneSwitches to identify regulatory genes along the trajectories. However, several actionable genes identified in our paper also appear to be implicated in this dataset: AXL (p25, p26, P28), STAT2 (p25, p26) and AURK (p26, p28).

### Validation of SOX4 expression
We extracted cells that were annotated as T cells in Puram et al.[12] and clustered using the Seurat R package. Raw counts were normalized using regularized negative binomial regression via function SCTransform. The number of genes per cell were regressed out in a second non-regularized linear regression. T cell clusters were identified using SNN-based clustering based on the first 11 principal components (except the third PC which was contributed by RP-genes) with resolution = 0.8. In total, we obtained 7 clusters. Based on the DE and marker genes, three clusters were annotated as CD8 T cells and the rest clusters were CD4 T cells. Since we focus on CD8 T-cell analysis, cells in three CD8 T-cell clusters were extracted and re-applied the above clustering analysis with the following changes: SNN-based clustering was based on the first 5 principal components (except the third PC which was contributed by RP- genes) with

resolution = 0.5. As results, three clusters were annotated as naive-like, transitional and dysfunctional cells. Expression of SOX4 and RBPJ was higher in CD8 dysfunctional populations compared to other CD8 T cells (Wilcoxon rank sum test).

### Statistical analysis
Statistical analysis was performed using GraphPad Prism software (GraphPad Software, Inc., San Diego, CA), or otherwise indicated in the figure legends or other Methods sections.

### Reporting summary
Further information on research design is available in the Nature Portfolio Reporting Summary linked to this article.

## Data availability
The raw human and mouse 10x scRNAseq raw data generated in this study and the corresponding processed Seurat objects with all cells have been deposited in the GEO under accession code GSE188737 and GSE225170 respectively. The Fluidigm C1 scRNA-seq raw data and processed gene expression matrix have been deposited in the GEO under accession code GSE225331. The scRNAseq count data and original cell annotations for the published study of HNSCC[12] and cutaneous squamous-cell carcinoma[30] were downloaded from the GEO under accession code GSE103322 and GSE123813. The HNSCC transcriptome profiling gene expression files with FPKM values and related clinical parameters were obtained from TCGA portal (https://cancergenome.nih.gov). 10x scRNAseq data for tumor cells and CD8+ T cells can be accessed and interrogated as an interactive web application via the following Shiny app (http://hnc.ddnetbio.com/). The remaining data are available within the Article, Supplementary Information. Source data are provided with this paper.

## Code availability
All code used to analyse the dataset is openly available at https://doi.org/10.5281/zenodo.7692887[63]. All software and algorithms used in this study are publicly available and are listed in the Methods section.

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

## Acknowledgements

We would like to thank all patients and families who contributed to this project. Additional technical support and access for 10x experiments were obtained from the Laboratory of Cell Therapy and Cancer Vaccine, National Cancer Center Singapore (under Dr Han-Chong Toh), C1 experiments from the Laboratory of Cancer Epigenome (under Professor Bin-Tean Teh and Mr Cedric Ng). Humanized mice were produced by the CIEA-KKH-SIgN HuNIT platform supported by the Industry Alignment Fund – Pre-positioning Program (IAF-PP) titled, "Humanized NOG Mice for ImmunoTherapy (HuNIT): Development and large-scale production of HLA-matched humanized mice and their application to translational medicine (H16/01/a0/002)". The computational work for this article was partially performed on resources of the National Supercomputing Center, Singapore (https://www.nscc.sg). This project was funded through the following grants awarded to the respective investigators, for which we are truly grateful: Khoo Postdoctoral Fellowship Award (Duke-NUS-KPFA/2018/0024) to QHS, National Medical Research Council (Singapore) Clinician Scientist Awards (NMRC/CSA/001/2016, MOH-000325-00) to NGI, the Peter Fu Head and Neck Cancer Program (under the Oncology Academic Clinical Program, National Cancer Center Singapore) to NGI, core funding by Singapore Immunology Network (A*STAR) to SKB and a Singapore National Research Foundation grant (NRF-CRP20-2017-0002) to OR and JO.

## Author contributions

H.S.Q., E.Y.C., and N.G.I. designed the experiments and analyzed the data. N.G.I. surgically retrieved the tumor tissues from patients. H.S.Q. and L.S. conducted the experiments with assistance from H.S.L, F.T.C and C.A. E.Y.C performed the single-cell RNAseq analysis for the human samples and H.S.Q. performed the single-cell RNAseq analysis for the humanized mouse samples with expert assistance from C.H.L. C.H.L. performed the CopyKAT analysis for single-cell RNAseq and TCGA bulk RNAseq analyses. S.G and V.A. established the HNSCC cancer humanized mouse model and performed the pembrolizumab treatment. H.S.Q. and S.G. performed the flow cytometry analysis. T.C., Y.Z., H.C.T., J.C. and T.T. provided the mouse model. C.H.L., J.F.O., D.SW.T, S.K.B, O.JL.R. contributed to data interpretation. H.S.Q, E.Y.C. and N.G.I. wrote the paper. H.S.Q, E.Y.C., C.H.L. and N.G.I. revised the paper. All authors contributed to the paper preparation.

## Competing interests

NGI has/had a consulting or advisory role in PairX Therapeutics and Invitrocue PLC, and received honoraria from Kalbe Biotech and Agilent, all of which are outside this submitted work. DSWT received honoraria from Bristol-Myers Squibb, Takeda Pharmaceuticals, Novartis, Roche, and Pfizer; and has consulting or advisory role in Novartis, Merck, Loxo Oncology, AstraZeneca, Roche, and Pfizer. DSWT also received research funding from Novartis (Inst), GlaxoSmithKline (Inst), and AstraZeneca (Inst), outside this submitted work. None of the remaining authors have any other conflicts to report.
