## [Peer Review File · Nature Communications]

Single cell analysis in head and neck cancer reveals potential immune evasion mechanisms during early metastasisReviewers' Comments:

Reviewer #1:

Remarks to the Author:

I reviewed "Single cell analysis of early metastasis identifies targetable tumor subpopulation and 2 mechanisms of immune evasion in squamous cell cancers" by Quah et al initially with interest. Unfortunately, I found the methodology troubling, the conclusions poorly supported, and generally numerous assumptions being made to fit an explanation to the data rather than using the data to come up with robust observations. I am most troubled by the use of just a few tumors for most analyses, despite capturing data from 14 tumors, as well as the back and forth between which tumors are used (almost entirely distinct) from one section of the paper to another. The biological validations at a surface level seem interesting, but are missing important controls and do not rigorously support the broader conclusions outlined. These limitations severely dampen my enthusiasm. At the end, I was left with no really clear idea of what this paper is convincingly demonstrating or trying to convey.

MAJOR

1. Median number of genes detected is very very low. Most studies we would expect ~2000 genes per cell. This raises concerns about the quality of the data obtained here.
2. "95% of tumors had aneuploidy" – this is very surprising and unexpected. Most head and neck tumors have gain or loss of an arm, but not an entire chromosome and this finding is inconsistent with head and neck TCGA, again raising concerns about data analysis. I think the authors mean 95% of epithelial cells have CNAs, in which case that makes more sense to me.
3. I do not really appreciate or see the site specific clustering mentioned for HN251 and HN279 – on the UMAP these are hardly distinct clusters and a much more rigorous approach would be needed to show these clusters are robust. For example, do the authors see a distinct cluster by NMF or a similar approach? And does that cluster appear to be primarily from the mets vs pri? What are the top differentially expressed genes if so?
4. There is a fundamental problem with using trajectories of any sort with the experimental paradigm as presented. Primary tumors do not represent an evolutionarily "earlier" time point because they are continuing to evolve at the same time as the LN metastasis forms (patients have surgery as a single time point where both are collected). Thus the statement "primary tumor presumed to pre-date nodal disease" is fundamentally flawed. That is why time-based trajectories with a single chronologic time point are not really meant to be used in the fashion utilized here. This is emphasized by the unclear separation of primary and metastatic states seen by the authors. If the authors really saw clear differences in mets vs primary with EMT or other signatures, then that's perhaps interesting but their data suggest these differences are minor at best. The authors use an orthogonal dataset to help support their points, but even here only 3/5 tumors suggest "more EMT" while the other 2/5 showed primary to have more EMT and these differences are rather underwhelming overall. Hardly, the type of robust orthogonal validation one would hope for.
5. In general there is a lot that is being assumed.
 - For example, it is unclear what the basis for the assumption that a primary tumor population with EMT evolved after nodal metastasis. It could also be that the authors' hypothesis is simply wrong, but they don't acknowledge this possibility. In some sense the authors have put themselves in a bind – on the one hand they argue the primary is representative of any early time point in tumor biology yet on the other hand when the data don't fit their model, they say it must be because the tumor evolved. This inconsistency is very tough to evaluate the results and emphasizes the points above about issues with pseudotime (or making comments about tumor evolution) in a static biological specimen.
 - I would raise the same concerns about the T-cell analysis. There are many poorly executed papers that use this approach and I have yet to see a finding here that is robust and biologically validated in cancer. The same issue of making broad assumptions extends to these analyses where the authors presume there is a "burst" of CD8+ tumor specific T-cells without doing the clonal analysis to support that.
6. Generating tumor derived cultures is challenging in my experience for HNSCC and I reviewed the methods outlined here which seem rather simple. I would therefore think it is important to confirm

that these tumor derived cultures are in fact malignant cells with WES to confirm CNAs are present and match the inferred CNAs by Copykat or something similar. Otherwise, many of the signatures could represent merely fibroblasts being present in one sample vs another, or alternatively genetic drift in the derived cultures.

7. Finding one or two genes of interest such as AXL and AURK misses the real story and biology. Single genes being identified and studied by scRNA-seq is notoriously poor as an approach – the authors should be looking for more robust signatures. I would expect other members of these pathways to be modulated but I do not see that data shown or presented. As a small aside, I would note that AXL has been shown to play a role in metastasis in HNSCC previously and I am unclear what the added value is of the findings shown here that goes beyond what's been done.

8. AXL is specifically higher in metastatic cells in HN137 according to the authors yet AXL inhibitors have an effect on inhibition in both pri and met. This goes against the authors conclusions and findings.

9. Sox4 finding suffers from many of the same issues as above. A single gene is not proof of a coordinated state or program. There is very little here to suggest that Sox4 is the critical driver of these states. Data from cSCC which is used is not relevant given the distinct biology of these tumors (for example PD1 response in cSCC is 47% while for mucosal it is 14%). Clearly these tumors are different, especially as it relates to the immune system and immune exhaustion. Along these lines, the FACS analysis with Sox4 KD is not all it seems to me. PD1 might be lower in these cells but the differences are not that strong while other markers of exhaustion such as LAG3 are not changed. Is there actually a coordinated difference in exhaustion beyond a single gene? Have the authors confirmed these cells are functionally exhausted? Also on a more of a gut check level, there are thousands of studies on T-cell exhaustion and many on Sox4, yet no other study has seen such a finding. This seems more likely to reflect an issue with the authors' approach than an unexpected finding.

10. "There was a progressive increase in clonality across the dysfunctional gradient, with evidence of single naïve or TRM-derived clones subsequently expanding to give rise to multiple dysfunctional clones that span these trajectories." I see a cartoon but don't see data supporting this statement. These trajectories also add very little in my mind in terms of our understanding of tumor biology more fundamentally as they are all over the place from sample to sample and not much is seen consistently across the dataset (e.g. Fig 5F).

11. Although the authors analyze 14 tumors, they spend the majority of their efforts analyzing just 2-3 tumors for any given portion of their study. They also jump around between different tumors from section to section (e.g. tumor analyzed for EMT in mets vs pri vs T-cells). This raises questions about just how generalizable these findings are and how the observations being made fit into the broader dataset of the authors' own cohort.

12. I am not sure what the pre-nodal subpopulation is defined as and this label seems highly arbitrary/dubious. Also, why only analyze 3 tumors out of 14 and 2 out of 5 from the prior published dataset? The explanation of "minimum RNA-seq" does not make sense given that the prior dataset had thousands of genes detected per cell. I recommend the authors perform the Cellchat analysis on all tumors from both datasets to show their findings are robust. I would also note that here again the authors use tumors that are entirely different than those used for other analyses (both from their dataset as well as the other study).

13. Although differences in AURKB and TOP2A might be different after pembro (6E), the magnitude of difference is very modest. It should be easy to test the proliferation of these cells directly and demonstrate there is a difference both functionally and by IHC with Ki67. Also, this experiment is missing the critical control of cell that aren't pre-nodal. Presumably based on the authors hypothesis there should be a difference in PD1 response but this is not reported which is worrisome. In addition, the idea that MDK-driven suppression might suppress ICB seems interesting but also potentially true-true and unrelated. The authors haven't don't the leg work to explain this expectation and backed it up with data. Finally, I would hardly describe a single humanized mouse model with a single tumor engrafted as adequate for the broad statements being made such as "these results implicate MDK-276 signaling as a pathway through which pre-nodal cells evade CD8-mediated immune-editing."

OTHER

1. Intro – what is the basis to described lymphovascular invasion as “pre-metastatic.” Certainly it is a sign of local invasion, but there is no firm association between this feature and tumors that eventually metastasize. For example, plenty of tumors have mets without lymphovascular invasion.
2. The description/terminology of some of the biology is a bit strange – “full blown distant metastasis” is not a real or relevant term I have seen used. Patients either have distant mets or do not have distant mets (M0 or M1).
3. When cells have been mixed 1:1 CD45+:CD45- it is hard to make any comments about cell proportions as these are no longer representative. Deconvolution of bulk is more reliable.
4. What EMT score is being used for analysis? Prior analyses in HNSCC patient samples suggest that traditional/classical EMT signatures are not really found and there are more hybrid states identified. I would assume these are the signatures that should be use in Fig 2B, for example.
5. Discussion is lacking in its description of limitations and also placing the findings into a broader context.
6. Citation of prior work in the field related to these specific areas of focus is lacking.

Reviewer #2:

Remarks to the Author:

In this manuscript, the authors are analysing from 7 untreated patient, the cell infiltrates of primary tumors and nodal sites. The approach is elegant and relevant to understand metastasis evolution. To achieve their scientific objectives, they mainly used scRNASeq as a way to characterize the cellular composition of the two sites analysed. E conclusion regarding the potential discovery of single-cell analysis makes no doubt. However, what did we learn at the end from this study besides that early metastasis is a complex and heterogeneous process? Authors should perform additional analysis to either highlight that from patient to patient, the process is different or to highlight eventually the, maybe few, commonalities; in other words, quantify more the heterogeneity and nuance.

Major comments:

Generally speaking, authors face the classical issues of single cell experiments which is the reproducibility. Although patient heterogeneity can explain the huge variation in terms of gene numbers per sample, it is more likely basic technical limitations from the method used. To strengthen the results and conclusions, authors should provide an evaluation of the consistency between patients/tumor sites of the obtained results, at least focusing on the major ones, to ensure that the conclusions are not driven by one outlier.

Sup Fig 1B/Tumor site should be in the main results as it is showing the imbalance of the cell count obtained depending on the tumor site.

T-cell analysis: Out of >3000 cells, about 1,461 and 1,948 contained a productive TCRA and TCRb sequence respectively. Out of these cells, in fact 1500 unique TCRab were found, suggesting that most of the TCR were represented only once, and therefore not shared between any of the patients. This is what authors showed. Is this expected? Authors suggested that dysfunctional CD8 have more expanded clones considering and clone to be expanded when >2 cells with the same TCR were identified. Several question and analysis should be done to better characterize the samples:

- Is there any TCR shared between cells at different state of differentiation?
- Better representation of the clonality distribution is required. Authors should objectively and statistically define the threshold from which they consider a TCR expanded.
- When a TCR is considered as expanded, was it based on the nucleotide sequence or based on the in silico amino-acid translated sequence?
- Although there is no sharing between patients, authors could consider using similarity network analysis (using levenshtein distance or hamming distance) or motif inference (GLIPH2, DeepTCR, ...) to eventually identify common patterns between patients and tumor sites.

Reviewer #4:

Remarks to the Author:

This is an excellent investigation yielding large amounts of new data, some of it clinically actionable, from a small number of HNSCC patient samples. Using pseudo time-ordering analysis, the authors first identify genes that are associated with a transition from pre-nodal to nodal disease - of which four are potentially actionable (AXL, Auk, TYMS and STAT2). They go on to analyse CD8+ T cells, and map distinct trajectories from pre-dysfunctional to exhausted phenotypes for immigrant and resident T cells. This is a particularly important contribution to the field which the authors take further and show that SOX4 expression correlates with the transition in the first of these lineages. They validate this finding in a humanised mouse PDX model.

I recommend the manuscript for publication on condition that the authors

- a) Contextualise their findings more explicitly with what is already known about the pathways to T cell exhaustion in cancer. In particular, the reversible loss-of-function phenotypes characterised by SLAMF6 (and CD69) expression, as well as the bifurcation of TCF1 expression that appears to be involved in fate determination in newly activated T cells.
- b) Discuss their findings in the context of spatiotemporal phenotype data in HNSCC. For example, the potential involvement of CAF in influencing the immune microenvironment and the transition to metastatic disease.

Dear Reviewers,

Thank you for your detailed review and comments in response to our submitted manuscript entitled “Single cell analysis of early metastasis identifies targetable tumor subpopulation and mechanisms of immune evasion in squamous cell cancers”. We are heartened by the positive comments below and have provided a point-by-point response to specific queries raised, and with the associated changes, have undoubtedly improved the quality of this manuscript. We hope you find these satisfactory and look forward to hearing from you.

**Reviewer #1, expert in single cell sequencing/head and neck cancer/immune cells
(Remarks to the Author):**

I reviewed “Single cell analysis of early metastasis identifies targetable tumor subpopulation and 2 mechanisms of immune evasion in squamous cell cancers” by Quah et al initially with interest. Unfortunately, I found the methodology troubling, the conclusions poorly supported, and generally numerous assumptions being made to fit an explanation to the data rather than using the data to come up with robust observations. I am most troubled by the use of just a few tumors for most analyses, despite capturing data from 14 tumors, as well as the back and forth between which tumors are used (almost entirely distinct) from one section of the paper to another. The biological validations at a surface level seem interesting, but are missing important controls and do not rigorously support the broader conclusions outlined. These limitations severely dampen my enthusiasm. At the end, I was left with no really clear idea of what this paper is convincingly demonstrating or trying to convey.

>Thank you for this comment. This is a dense manuscript where we attempted to analyse a range of different cell populations that influence early nodal metastases, and even after only focusing on tumor and CD8 positive cells, have a range of topics to discuss. The main message of this paper is to highlight the power of single cell analyses, specifically in relation to specific algorithms such as trajectory mapping and interactome analyses, to answer specific questions in a curated sample set. While the number of patients included here may appear to be small, there are few similar analyses in the literature that have paired samples such as these (which has been an issue for us finding good datasets for validation as well). In order to clarify this, we have clarified the specific objectives of this study at the end of the introduction. We have also re-organized the text and figures in a more straightforward manner to go with the flow of the text, and have modified the text at various sections to clarify the thought process for each specific analysis and conclusion. We have also rearranged some of the Supplementary figures to be more self-explanatory, for example Supplementary Figure 2 to go in line with the flow of the text. In this case, we first described a “class” of tumors that exhibited an ordered, progressive, step-wise transition from primary to nodal disease and thus we were able to detect pre-nodal primary cancer cells in our data and also validated this in Puram’s data. These pre-nodal primary cancer cells expressed genes (AXL, AURKB, STAT2, REL, RXRA, etc) and we

posited these genes to be “pro-metastatic”. Second, we described another “class” of tumors that exhibited a haphazard fashion in our data and subsequently validated this in Puram’s data.

MAJOR

1. Median number of genes detected is very very low. Most studies we would expect ~2000 genes per cell. This raises concerns about the quality of the data obtained here.

>Thank you for highlighting this. Is this in reference to totals or unique genes per cell? We will clarify this in our text, as our result reported is the median number of **unique** genes per cell. Actually with most 10x droplet based technology for solid tumors, the median number of unique gene per cell is around 1000. 2000 genes usually applies to work from cell lines, liquid tumors and some of the initial technologies.

Here, the samples were derived from freshly dissociated patient tumor tissues which contained a heterogeneous population of cell types expressing genes at different amounts. If we focus on epithelial and T cells, the median number of genes per cell is 1426 and 540, respectively (Rebuttal Figure 1), and this number is comparable to Kurten et al (Kurten CHL et al 2021 Nat Comms PMID: 34921143). Many researchers now believe that reducing the cutoff to 200-300 genes per cell allows the capture of data from a broader range of cells and not just focus the analysis on transcriptionally active cell types. For example, T-cells that are “less transcriptionally active” which includes dysfunctional/exhausted T cells, cells in senescence), and quiescent memory cell populations. These also include epithelial cells that are less transcriptionally active in quiescence or undergoing EMT, and other cell types such as granulocytes that are often not seen in some of the older analyses and having a high threshold leaves these cell types out of the analyses. The heterogeneity of cells in the tumor with low and high number of genes in Supplementary Figure 1A as a UMAP titled Number of Genes.

Nonetheless, we are confident that our data is of high quality. We applied a series of stringent quality control (QC) criteria to our data. The viability of each sample was more >85% upon the single cell capture. In addition to the genes per cell being used as a criteria as stated by the reviewer, additional QC criteria were also applied to exclude cells with greater than 20% mitochondrial RNA content, greater than 40,000 UMI or 6000 genes. These QC criteria are applied in many papers published in reputable journals including Nature Comms. Nonetheless, after performing these additional QC steps, we were able to annotate the cell types and execute downstream analyses. As shown in Rebuttal Figure 2, cell clusters for all the cells, or only extracted epithelial (shown in Supplementary Figure 2A) or CD8+ T-cells (shown in Supplementary Figure 4E) were not determined by the number of genes, indicating the quality of the data was not “as bad” as highlighted.

cell type	num_UMI	num_Gene
Epithelial	3291	1427
T cells	889	540

	Median gene per cell
Zheng Y et al 2020 Nat Commun	1170
Kurten CHL et al 2021 Nat Commun	1077

	Our data		Kurten CHL	
	median_UMI	median_Gene	median_UMI	median_Gene
All cells	1631	776	3163	1077
Epithelial	3291	1427	7711	2352
CD8+ T-cells	776	529	2574	1009

Rebuttal Figure 1. Table showing the median number of UMI and median number of **unique** genes of epithelial or T cells.

Rebuttal Figure 2. UMAP showing number of UMI counts, number of genes and percentage of mitochondrial genes for all the cells, epithelial cells and CD8+ T-cells.

2. “95% of tumors had aneuploidy” – this is very surprising and unexpected. Most head and neck tumors have gain or loss of an arm, but not an entire chromosome and this finding is inconsistent with head and neck TCGA, again raising concerns about data analysis. I think the authors mean 95% of epithelial cells have CNAs, in which case that makes more sense to me.

>Thank you for pointing this out. Your second statement is the correct one, and we have clarified this in the text: “Inferred copy number variant analyses on the epithelial population showed that aneuploidy was evident in >95 % of cells validating that this population comprised predominantly cancer cells (Figure 1E and Supplementary Figure 1E)”.

3. I do not really appreciate or see the site specific clustering mentioned for HN251 and HN279 – on the UMAP these are hardly distinct clusters and a much more rigorous approach would be needed to show these clusters are robust. For example, do the authors see a distinct cluster by NMF or a similar approach? And does that cluster appear to be primarily from the mets vs pri? What are the top differentially expressed genes if so?

Thank you for pointing this out, as you are right that we did not show this in an obvious manner and have attempted to rectify this. We show this more clearly in the bar plot below (which is now in the supplementary data) and have also highlighted this in the text in a clearer manner. This showed that HN251 primary cells are mainly from cluster 10 and 11, while HN251 met cells are in cluster 11 only. Hence, cluster 11 is the met-specific cluster (Rebuttal Figure 3). Such distinct clusters were also observed when we apply PCA on HN251 cells. Rebuttal Figure 4 shows that cells from cluster 11 and 10 can be separated along PC1 and met cells are within cluster 11. In addition, cells are not grouped by number of genes. Instead, top differentially expressed genes along PC1 (Rebuttal Figure 5) shows mesenchymal signatures (e.g. gene MT2A, YAP1) for cluster 11 and epithelial signatures (e.g. gene S100A8, KRT16) for cluster 10. Same distinct clusters can also be observed when we apply tSNE dimension reduction (Rebuttal Figure 6). We have included all of these figures as Supplementary Figure 2A. Similar findings are apparent for HN279, where there is an increasing number of metastatic epithelial cells in clusters 4,3,9 and 5, where we believe that the primary cells in cluster 5 are closest to the nodal population, with tSNE shown below. While we have highlighted these in the text, they also demonstrate that this may not be ideal methodology to identify pre-nodal primary cells, and hence we opted to use trajectory analyses as our preferred algorithm, and we have now stated this more clearly in the text.

Rebuttal Figure 3. Barplot showing the percentage of site of origin (Metastatic/Primary) for each cluster. The patient ID is the major composition for the corresponding clusters. Cluster numbers on the x-axis correspond to those of Figure 2A.

Rebuttal Figure 4. PCA plots of cells from patient HN251 colored by site of origin, cluster and number of genes (upper panel). Gene expression plots in PCA embedding for gene MT2A, S100A8 and STAT2.

Rebuttal Figure 5. Heatmap of top differentially expressed genes along PC1.

Rebuttal Figure 6. tSNE plots of cells from patient HN251 colored by site of origin, cluster and number of genes (upper panel). Gene expression plots in tSNE embedding for gene MT2A, S100A8 and STAT2.

Rebuttal Figure 7. tSNE plots of cells from patient HN279 colored by site of origin, cluster and number of genes (upper panel). Gene expression plots in tSNE embedding for gene CXCL3, CXCL10 and AXL.

4. *There is a fundamental problem with using trajectories of any sort with the experimental paradigm as presented. Primary tumors do not represent an evolutionarily “earlier” time point because they are continuing to evolve at the same time as the LN metastasis forms (patients have surgery as a single time point where both are collected). Thus the statement “primary tumor presumed to pre-date nodal disease” is fundamentally flawed. That is why time-based trajectories with a single chronologic time point are not really meant to be used in the fashion utilized here. This is emphasized by the unclear separation of primary and metastatic states seen by the authors. If the authors really saw clear differences in mets vs primary with EMT or other signatures, then that’s perhaps interesting but their data suggest these differences are minor at best. The authors use an orthogonal dataset to help support their points, but even here only 3/5 tumors suggest “more EMT” while the other 2/5 showed primary to have more EMT and these differences are rather underwhelming overall. Hardly, the type of robust orthogonal validation one would hope for.*

>Thank you for pointing this out, as it was something we should have clarified in the text. We agree that that NOT all primary tumors represent an evolutionary “earlier” time point as they can continue to evolve at the same time as LN met forms. However, we used the trajectory analyses based on an assumption that the concept of an evolutionary time point could still exist in a fraction of the tumors, while being prepared to reject this hypothesis in others (and potentially even a pattern in between. Indeed, this exactly what we observed: 3 different patterns: one which follows the rule of an evolutionarily early primary tumor which does not appear to evolve further (eg HN251 and HN242 and Puram p26 and p28), a second where there is a clear trajectory from the primary to met, BUT the primary tumor continues to diverge to other pathways (HN272 and HN279; Puram p25). In both of these, pri-met tumors behave in an ordered, progressive, stepwise transition from primary to nodal met (higher EMT in the met),

where the met likely continues to progress after dissemination from the primary had occurred. In this instance, we were able to narrow down and identify pre-nodal metastases subpopulations for further downstream analyses. In the third “class” of pri-met tumors (HN257; p5 and p20), the evolution certainly follows what the reviewer has pointed out, where trajectories are haphazard (higher EMT in the primary), and we postulate that a subpopulation within the primary tumor likely evolved further after nodal dissemination making the identification of pre-nodal clones impossible. We have clarified this in the main text.

5. *In general there is a lot that is being assumed.*

- For example, it is unclear what the basis for the assumption that a primary tumor population with EMT evolved after nodal metastasis. It could also be that the authors' hypothesis is simply wrong, but they don't acknowledge this possibility. In some sense the authors have put themselves in a bind – on the one hand they argue the primary is representative of any early time point in tumor biology yet on the other hand when the data don't fit their model, they say it must be because the tumor evolved. This inconsistency is very tough to evaluate the results and emphasizes the points above about issues with pseudotime (or making comments about tumor evolution) in a static biological specimen.

>As stated above, we have clarified by dividing the trajectories into 3 different patterns seen both in our and Puram's dataset, and have clarified this in the main text as well.

- I would raise the same concerns about the T-cell analysis. There are many poorly executed papers that use this approach and I have yet to see a finding here that is robust and biologically validated in cancer. The same issue of making broad assumptions extends to these analyses where the authors presume there is a “burst” of CD8+ tumor specific T-cells without doing the clonal analysis to support that.

>We thank the reviewer for the comment. Firstly, the analysis is definitely easier to understand in the context of T cells as the populations and trajectory is much more clearly defined by known T cell biology, and the trajectory we generate fits perfectly with the expected gene expression during T cell maturation. However, with TCR sequencing we are able to do exactly what the reviewer suggested, which is clonal analysis. To better represent our data and clarify our thoughts, we generated two types of patient-specific TCR clonal analysis plots - 1) TCR clonal sharing across differentiation states and 2) TCR clonal sharing between primary and met sites. These plots are displayed as Supplementary Figure 5J-K in the manuscript, and these demonstrate how clonality predominates in the dysfunctional subpopulations across the clonotypes examined, with evidence of the same clonal expansion in the subpopulations demonstrating a proliferative burst (which few studies have shown before)

Rebuttable Figure 8. (Top) TCR clonotype distribution across T cell differentiation states from tumors of HN272, HN257 and HN263. (Bottom) TCR clonotype distribution between primary or metastatic in HN272 and HN257. Only the top 5 clonotypes from each sample are shown.

6. *Generating tumor derived cultures is challenging in my experience for HNSCC and I reviewed the methods outlined here which seem rather simple. I would therefore think it is important to confirm that these tumor derived cultures are in fact malignant cells with WES to confirm CNAs are present and match the inferred CNAs by Copykat or something similar. Otherwise, many of the signatures could represent merely fibroblasts being present in one sample vs another, or alternatively genetic drift in the derived cultures.*

>One of our lab's greatest strength is to establish patient-derived cell lines and have a pipeline that has been well-established, with a cell line establishment rate of 30-40%. In fact ALL our functional work is only performed on patient-derived lines which have been characterised and confirmed to be genetically matched to the original tumors. We have included a number of references to support this based on our previous publications.

7. *Finding one or two genes of interest such as AXL and AURK misses the real story and biology. Single genes being identified and studied by scRNA-seq is notoriously poor as an approach – the authors should be looking for more robust signatures. I would expect other members of these pathways to be modulated but I do not see that data shown or presented. As a small aside, I would note that AXL has been shown to play a role in metastasis in HNSCC previously and I am unclear what the added value is of the findings shown here that goes beyond what's been done.*

Thank you for pointing this out. We have clarified that our objective is to find single gene and/or protein that can be modified or targeted in a specific tumor context, and we show this using multiple layers of data presented here: single cell data from our fresh patient-derived tumors and cell cultures, validation of our findings using Puram's data, our functional experiments by blocking AXL and its effect on invasion on bulk and sorted populations.

The identification of a gene like AXL is exactly what would hope to find, as it provides orthogonal validation, that despite all the assumptions, our methodology is able to identify a well-established gene such as AXL. Yet, we also want to state that our data shows that AXL is only active in a fraction of tumors and even then, only a subpopulation of cells that appear to contribute to nodal metastasis, and so targeting AXL needs to take these into context. We have stated this point in the main text well

8. AXL is specifically higher in metastatic cells in HN137 according to the authors yet AXL inhibitors have an effect on inhibition in both pri and met. This goes against the authors conclusions and findings.

>As there is a gradation of AXL expressing cells and not an all or none observation, it is expected that AXL inhibition will have an inhibitory effect on both pri and met. We provided multiple layers of data and compounding evidence to show this is the case. Immunostaining of HN137 primary and met tissues, and PDCs shows presence of AXL expressing cells at both sites, more in the met than the pri (Figure 3D and Supp Fig 3C-3D). In line with this, flow cytometry analysis of HN137 patient-derived cell cultures shows ~30% of cells express high levels of AXL (AXL^{hi}) in the primary (Supplementary Figure 3J-K), and these AXL^{hi} cells are more susceptible to AXL inhibition. To further clarify and strengthen our findings, we then sorted HN137 Primary into populations expressing high, mid or low/neg levels of AXL, prior to the AXL inhibition and invasion assay (Figure 3H).

9. Sox4 finding suffers from many of the same issues as above. A single gene is not proof of a coordinated state or program. There is very little here to suggest that Sox4 is the critical driver of these states. Data from cSCC which is used is not relevant given the distinct biology of these tumors (for example PD1 response in cSCC is 47% while for mucosal it is 14%). Clearly these tumors are different, especially as it relates to the immune system and immune exhaustion. Along these lines, the FACS analysis with Sox4 KD is not all it seems to me. PD1 might be lower in these cells but the differences are not that strong while other markers of exhaustion such as LAG3 are not changed. Is there actually a coordinated difference in exhaustion beyond a single gene? Have the authors confirmed these cells are functionally exhausted?

> Some of these statements are not entirely true, for example PD1 response rates can vary significantly based on various selection criteria across all squamous cell cancers (eg CPS scores, tumor mutation burden, prior treatment, stage of disease etc). Our use of the cSCC data here is only because there is little in the way of post-PD1 treated validation sets that have been

published, and hence our limitation. The fact even in this dataset Sox4 could be seen as associated with reversal of exhaustion is encouraging, but merely supportive at best.

The functional assays here are also somewhat limited, as is the case for most assays of this nature. Our aim here was not to generate exhausted T cells in vitro. Rather it is to induce the upregulation of these markers (ie PD1, LAG3, CD39, CD57) on normal PBMCs using anti-CD3/CD28 activation, and then test whether SOX4 has a role in regulating these genes on CD8+ T cells using siRNA targeting SOX4 mRNA. Therefore it was more a question of modulating genes involved in dysfunction. Our results showed a subtle yet consistent significant reduction of PD1, CD39, CD57 markers on CD8 with SOX4 KD, while LAG3 showed no difference. These suggest 1) SOX4 alone does not have a dominant effect in regulating PD1, CD39 and CD57 but definitely has a role in the signalling cascades of these markers, and 2) SOX4 may not have a role in the LAG3 signalling cascade. Empirically, the reduction of the expression of these markers could delay the exhaustion of CD8 T cells in the tumor settings. We will clarify this in the text.

Nonetheless, we examined multiple layers of orthogonal data - our single cell data, validation of external single cell data, and our functional siSOX4 KD using PBMC. Furthermore, in this dataset, we arrived at Sox4 after having initially identified 3 candidates, but excluding 2 (DUSP4 and RBPJ) as they did not fit into the external datasets nor functional studies. Altogether, these orthogonal data support each other and provide compounding evidence to suggest SOX4 is a driver of these states.

Also on a more of a gut check level, there are thousands of studies on T-cell exhaustion and many on Sox4, yet no other study has seen such a finding. This seems more likely to reflect an issue with the authors' approach than an unexpected finding.

>Lack of orthogonal validation does not negate the discovery here (although ironically there is now orthogonal data supporting the identification of Sox4). One reason Sox4 was not previously identified could be the lack of single cell transcriptomic data and algorithm available previously to perform the pseudo-time analysis of CD8+ T cells from naive to dysfunctional state. The correlation of the expression of SOX4 with the exhaustion markers highlighted in our data is also validated using other reputable published data, and this is shown in our manuscript (Figure 5A and 5B, and Supp Fig 5C and 5E). Just recently, in the midst of manuscript review, two papers were recently published in Dec 2021 by Zheng L et al 2021 Science and Good CR et al 2021 Cell), demonstrating SOX4 has a role in T cell exhaustion. These studies are in line with our findings.

10. "There was a progressive increase in clonality across the dysfunctional gradient, with evidence of single naïve or TRM-derived clones subsequently expanding to give rise to multiple dysfunctional clones that span these trajectories." I see a cartoon but don't see data supporting this statement. These trajectories also add very little in my mind in terms of our understanding of tumor biology more fundamentally as they are all over the place from sample to sample and not much is seen consistently across the dataset (e.g. Fig 5F).

>Thank you for bringing this up. We merely used the cartoon as a representation of our findings, but have now included clearer data (in the form of bar graphs) in the Supplementary data section as mentioned under point 5.

11. Although the authors analyze 14 tumors, they spend the majority of their efforts analyzing just 2-3 tumors for any given portion of their study. They also jump around between different tumors from section to section (e.g. tumor analyzed for EMT in mets vs pri vs T-cells). This raises questions about just how generalizable these findings are and how the observations being made fit into the broader dataset of the authors' own cohort.

>The fact that we observed different phenotypes specific to individual patients, for cancers, demonstrate the complexity of the metastatic phenomenon in HNSCC, and hence phenotypes or pathways described were limited to fewer than the 14 tumors we started out with. What was heartening was that this data could be validated even in a relatively weaker dataset such as by Puram et al, which comprised 10x fewer cells than our dataset. The immune cell data was more generalisable and driven by the what is known on T-cell biology and maturation, and yet we were able to validate these also using Puram's and Yost's datasets. So the generalizability from the 2-3 samples extends across different studies, lending weight to our findings (notwithstanding the identification of pathways that have been discovered in different circumstances- eg Sox4 in CAR-T exhaustion and MDK interactions in melanoma).

12. I am not sure what the pre-nodal subpopulation is defined as and this label seems highly arbitrary/dubious. Also, why only analyze 3 tumors out of 14 and 2 out of 5 from the prior published dataset? The explanation of "minimum RNA-seq" does not make sense given that the prior dataset had thousands of genes detected per cell. I recommend the authors perform the Cellchat analysis on all tumors from both datasets to show their findings are robust. I would also note that here again the authors use tumors that are entirely different than those used for other analyses (both from their dataset as well as the other study).

We did attempt to perform Cellchat analysis on all tumors from both datasets. However, Chatcell requires a minimal number of signal "donor" and "recipient" cells for interrogation to function accurately. In this case, we set a threshold of minimum 20 of signal "donor" and "recipient" cells. Tumors with number of cells below the threshold yielded no results or no meaningful results

It is not true that the tumors used were entirely different. To re-emphasize, Puram's samples and data were also derived from oral cavity and lymph nodes of HNSCC patients. This is stated in Puram's paper, "Human Tumor Specimens" in the "EXPERIMENTAL MODEL AND SUBJECT DETAILS" section, "Fresh biopsies of oral cavity head and neck squamous cell carcinoma (HNSCC) were collected at the time of surgical resection, either from the primary tumor or lymph node (LN) dissection." Our and their tumor samples were also derived from the same location. The only caveat being that we analysed 50,000 cells in our datasets, while Puram only had 5000 cells available.

13. *Although differences in AURKB and TOP2A might be different after pembro (6E), the magnitude of difference is very modest. It should be easy to test the proliferation of these cells directly and demonstrate there is a difference both functionally and by IHC with Ki67. Also, this experiment is missing the critical control of cell that aren't pre-nodal. Presumably based on the authors hypothesis there should be a difference in PD1 response but this is not reported which is worrisome. In addition, the idea that MDK-driven suppression might suppress ICB seems interesting but also potentially true-true and unrelated. The authors haven't don't the leg work to explain this expectation and backed it up with data. Finally, I would hardly describe a single humanized mouse model with a single tumor engrafted as adequate for the broad statements being made such as "these results implicate MDK-276 signaling as a pathway through which pre-nodal cells evade CD8-mediated immune-editing."*

>Thank you for asking about this. We have actually submitted the histological and flow cytometry analyses for this mouse model in a different manuscript and hence did not show it here. However, we do have data showing reduction in human EPCAM+ cells in tumors of mice treated with anti-PD1 antibody (Supplementary Figure 6N), supporting tumor cell targeting

As the reviewer has pointed out, we have also included the PD1 response curve determined by tumor volume (Supplementary Figure 6I). One week after the start of anti-PD1 treatment, the tumor kinetic curve showed a slight reduction of tumor volume in mice treated with anti-PD1 while continuity of tumor volume growth of those that were untreated. Unfortunately, we had to harvest the tumors after a short period of treatment in order to capture the effect of the PD1 blockade on single cell analyses, and hence did not take the drug treatment to 'completion'.

As requested, we performed an in vitro MDK inhibition experiment with patient tumors (Figure 6C and Supplementary Figure 6F-H).

We have also toned down the statement to, "Taken together, these results suggest MDK-signaling as a pathway through which pre-nodal cells may use to evade CD8-mediated immune-editing."

All these changes were made and included in the text and supplementary.

OTHER

1. *Intro – what is the basis to described lymphovascular invasion as "pre-metastatic." Certainly it is a sign of local invasion, but there is no firm association between this feature and tumors that eventually metastasize. For example, plenty of tumors have mets without lymphovascular invasion.*

>Thank you for pointing this out. It is indeed an assumption and we have toned this down in the text.

2. *The description/terminology of some of the biology is a bit strange – “full blown distant metastasis” is not a real or relevant term I have seen used. Patients either have distant mets or do not have distant mets (M0 or M1).*

>We appreciate the reviewer for pointing this common and casually used term (which may be a cultural thing!). We have edited the text for better clarity.

3. *When cells have been mixed 1:1 CD45+:CD45- it is hard to make any comments about cell proportions as these are no longer representative. Deconvolution of bulk is more reliable.*

>We thank the reviewer for picking this up. We modified and separated the current figure into two major populations, immune cells (CD45+) and non-immune cells (CD45-), and normalized each to 100%. As such, comments made about the cell proportions within CD45+ (eg. T cell in primary vs metastatic tumor) and CD45- (eg. fibroblast in primary vs metastatic tumors) become more reliable and representative. We have made changes to the main text and figures (Figure 1D). This should clarify the concern.

4. *What EMT score is being used for analysis? Prior analyses in HNSCC patient samples suggest that traditional/classical EMT signatures are not really found and there are more hybrid states identified. I would assume these are the signatures that should be use in Fig 2B, for example.*

>As described in Methods section, the GSEA MSigDB gene set “HALLMARK_EPITHELIAL_MESENCHYMAL_TRANSITION” was used to define the EMT score.

5. *Discussion is lacking in its description of limitations and also placing the findings into a broader context.*

>We have modified the text to incorporate these discussion points

6. *Citation of prior work in the field related to these specific areas of focus is lacking.*

>We have included more citations that are relevant to our work.

Reviewer #2, expert in TCR sequencing (Remarks to the Author):

In this manuscript, the authors are analysing from 7 untreated patient, the cell infiltrates of primary tumors and nodal sites. The approach is elegant and relevant to understand metastasis evolution. To achieve their scientific objectives, they mainly used scRNASeq as a way to characterize the cellular composition of the two sites analysed. E conclusion regarding the potential discovery of single-cell analysis makes no doubt. However, what did we learn at the end from this study besides that early metastasis is a complex and heterogeneous process? Authors should perform additional analysis to either highlight that from patient to patient, the

process is different or to highlight eventually the, maybe few, commonalities; in other words, quantify more the heterogeneity and nuance.

Major comments:

Generally speaking, authors face the classical issues of single cell experiments which is the reproducibility. Although patient heterogeneity can explain the huge variation in terms of gene numbers per sample, it is more likely basic technical limitations from the method used. To strengthen the results and conclusions, authors should provide an evaluation of the consistency between patients/tumor sites of the obtained results, at least focusing on the major ones, to ensure that the conclusions are not driven by one outlier.

Sup Fig 1B/Tumor site should be in the main results as it is showing the imbalance of the cell count obtained depending on the tumor site.

>We thank the reviewer for the helpful suggestion. The Tumor site figure is now shown in main Figure 1B as suggested. As stated above, we have also clarified how the data has been validated both with published datasets, a priori data on specific genes (eg AXL, AURK) and functional analyses on independent patient-derived cells.

T-cell analysis: Out of >3000 cells, about 1,461 and 1,948 contained a productive TCRa and TCRb sequence respectively. Out of these cells, in fact 1500 unique TCRab were found, suggesting that most of the TCR were represented only once, and therefore not shared between any of the patients. This is what authors showed. Is this expected? Authors suggested that dysfunctional CD8 have more expanded clones considering and clone to be expanded when >2 cells with the same TCR were identified. Several question and analysis should be done to better characterize the samples:

- Is there any TCR shared between cells at different state of differentiation?

>The reviewer is correct. There is TCR sharing between cells at different states of differentiation. To further clarify this with better representation, we generated two types of patient-specific TCR plots – 1) TCR sharing across differentiation states and 2) TCR sharing between primary and met sites. These plots are displayed as Supplementary Figure 5J-K in the manuscript.

Rebuttable Figure 1. (Top) TCR clonotype distribution across T cell differentiation states from tumors of HN272, HN257 and HN263. (Bottom) TCR clonotype distribution between primary or metastatic in HN272 and HN257. Only the top 5 clonotypes from each sample are shown.

- Better representation of the clonality distribution is required. Authors should objectively and statistically define the threshold from which they consider a TCR expanded.

> We have generated plots to better represent the clonality distribution (see the point above). In addition, clonotypes were defined either as expanded (i.e. detected in at least two cells) or unique (i.e. detected in no more than one cell). We have clarified in the main text, as well as included this statement in the Supplementary Methods under “Data processing of single-cell TCR-seq libraries.”

- When a TCR is considered as expanded, was it based on the nucleotide sequence or based on the *in silico* amino-acid translated sequence?

We apologize for not being clear. The sequencing TCR is based on the *in silico* amino-acid translated sequence.

- Although there is no sharing between patients, authors could consider using similarity network analysis (using levenshtein distance or hamming distance) or motif inference (GLIPH2, DeepTCR, ...) to eventually identify common patterns between patients and tumor sites.

>This is an excellent suggestion to add new findings to our manuscript. We performed GLIPH2 analysis and recovered a total of 126 clusters (pattern). As expected, we observed minimal TCR sharing across patients (5 of the 126 clusters) (Rebuttal Table 1). The table has been included as Supplementary Table S7.

index	pattern	TcRb	Sample_cell-state	Number of cells
1	SLEL	CASSLELAGETQYF	HN257M_Transitional	7
1	SLEL	CASSLELGADTQYF	HN272M_Dysfunctional	13
1	SLEL	CASSLELAGTYEQYF	HN263P_Dysfunctional	16
1	SLEL	CASSLELAGTYEQYF	HN263P_Dysfunctional	4
1	SLEL	CASSLELYGGTDTQYF	HN251P_Dysfunctional	12
1	SLEL	CASSLELYGGTDTQYF	HN251P_Dysfunctional	14
1	SLEL	CASSLELAGNEQFF	HN257P_Dysfunctional	3
2	PGLR	CATSEPGLRASTDQYF	HN272P_Pre-dysfunctional	2
2	PGLR	CASSFQPPGLREETQYF	HN257P_Dysfunctional	4
2	PGLR	CASSFQPPGLREETQYF	HN257P_Transitional	3
2	PGLR	CASSFQPPGLREETQYF	HN257M_Transitional	7
2	PGLR	CASSPGLRGFYNEQFF	HN272M_Transitional	5
3	%SGQGTD	CASSSGQGTDQYF	HN251M_Naive-like	5
3	%SGQGTD	CASTSGQGTDQYF	HN251P_Dysfunctional-proliferative	4
4	GRLE	CASSQGRLEQYF	HN272M_Transitional	8
4	GRLE	CATSPGRLEQFF	HN272M_Transitional	13
4	GRLE	CASSPRASGRLETQYF	HN251P_Dysfunctional	8
5	SSTGG%G	CASSSTGGYGYTF	HN263M_Naive-like	5
5	SSTGG%G	CASSSTGGWGYTF	HN272M_Pre-dysfunctional	4

Rebuttal Table 1. GLIPH2 analysis showed 5 clusters of TCR (indicated by pattern) that were shared across patient tumors (indicated by Sample_cell-state). There were a total of 126 clusters generated by GLIPH2. A table with the full results is included in Supplementary Figure S7.

Reviewer #4, expert in T cell exhaustion (Remarks to the Author):

This is an excellent investigation yielding large amounts of new data, some of it clinically actionable, from a small number of HNSCC patient samples. Using pseudo time-ordering analysis, the authors first identify genes that are associated with a transition from pre-nodal to nodal disease - of which four are potentially actionable (AXL, AuK, TYMS and STAT2). They go on to analyse CD8+ T cells, and map distinct trajectories from pre-dysfunctional to exhausted phenotypes for immigrant and resident T cells. This is a particularly important contribution to the field which the authors take further and show that SOX4 expression correlates with the transition in the first of these lineages. They validate this finding in a humanised mouse PDX model.

>We are grateful and heartened by these comments and would like to thank the reviewer for this.

I recommend the manuscript for publication on condition that the authors

a) Contextualise their findings more explicitly with what is already known about the pathways to T cell exhaustion in cancer. In particular, the reversible loss-of-function phenotypes characterised by SLAMF6 (and CD69) expression, as well as the bifurcation of TCF1 expression that appears to be involved in fate determination in newly activated T cells.

>We thank the reviewer's suggestion to improve the manuscript. We have included this in the discussion.

b) Discuss their findings in the context of spatiotemporal phenotype data in HNSCC. For example, the potential involvement of CAF in influencing the immune microenvironment and the transition to metastatic disease.

>We thank the reviewer's suggestion to improve the manuscript. We have also included this in the discussion.

Reviewers' Comments:

Reviewer #1:

Remarks to the Author:

I re-reviewed "Single cell analysis of early metastasis identifies targetable tumor subpopulation and 2 mechanisms of immune evasion in squamous cell cancers" by Quah et al. Unfortunately, I still have concerns about the methods, conclusions remaining poorly supported, and assumptions being made. I do not feel there are robust conclusions here and the authors continue to shift between different subsets of tumors which is worrisome. In addition, I do not find the identification of 5 unique genes (AXL, AURKB, STAT2, REL, RXRA, etc) which have no other coherent biological significance compelling as something unique to drive a "pro-metastatic" state.

MAJOR

1. Median number of genes detected is very very low. Most studies we would expect ~2000 genes per cell. This raises concerns about the quality of the data obtained here.

> Appreciate the efforts here. With that said, I think the Kurten data is also similarly weak due to the low number of expected genes. Agreed that T-cells tend to have lower detected genes but epithelial genes should really have >2000 unique genes. I couldn't disagree more strongly with the statement "many researchers now believe that reducing the cutoff to 200-300 genes per cell allows the capture of data from a broader range of cells." I would challenge the authors to find Nature, Cell, and Science papers with median detected genes in the range they show. 20% mitochondrial cutoff is also quite high. I would typically expect this to be closer to 10%. Regardless, the authors should take a stringent set of cutoffs and show their data to be robust.

2. "95% of tumors had aneuploidy" – this is very surprising and unexpected. Most head and neck tumors have gain or loss of an arm, but not an entire chromosome and this finding is inconsistent with head and neck TCGA, again raising concerns about data analysis. I think the authors mean 95% of epithelial cells have CNAs, in which case that makes more sense to me.

> I would still avoid term aneuploidy being used interchangeably with CNA, just say CNA.

3. I do not really appreciate or see the site specific clustering mentioned for HN251 and HN279 – on the UMAP these are hardly distinct clusters and a much more rigorous approach would be needed to show these clusters are robust. For example, do the authors see a distinct cluster by NMF or a similar approach? And does that cluster appear to be primarily from the mets vs pri? What are the top differentially expressed genes if so?

> Again appreciate the efforts and clarifications. I suggested the authors utilize NMF within a sample to show these clusters are robust, but this was not done. PCA across samples doesn't convince me that there are unique clusters. Finally, the finding of a mesenchymal PC1 and epithelial PC2 as already reported previously in the Puram paper the authors mention.

4. There is a fundamental problem with using trajectories of any sort with the experimental paradigm as presented. Primary tumors do not represent an evolutionarily "earlier" time point because they are continuing to evolve at the same time as the LN metastasis forms (patients have surgery as a single time point where both are collected). Thus the statement "primary tumor presumed to pre-date nodal disease" is fundamentally flawed. That is why time-based trajectories with a single chronologic time point are not really meant to be used in the fashion utilized here. This is emphasized by the unclear separation of primary and metastatic states seen by the authors. If the authors really saw clear differences in mets vs primary with EMT or other signatures, then that's perhaps interesting but their data suggest these differences are minor at best. The authors use an orthogonal dataset to help support their points, but even here only 3/5 tumors suggest "more EMT" while the other 2/5 showed primary to have more EMT and these differences are rather underwhelming overall. Hardly, the type of robust orthogonal validation one would hope for.

> While am sensitive to the issue of samples, it is hard to make any conclusion from what has been shown. The authors suggest there are 3 evolutionary patterns. But maybe these 3 differences are not

biologically significant and just represent the random variation. We cannot know if this is significant or not because there are just too few samples. This makes it very hard to know if the conclusions are biologically interesting and relevant or just simple anecdotes. 5 mice were injected with tumors and showed 3 different patterns of growth, say one with no tumors, one with medium tumor size, and one group with large tumor size, I hardly think anyone would say there must be 3 different "types" of tumor growth. Rather the first thing would be to have more samples, and I think the sample applies here.

> I also do not believe that trajectory analysis can be reliably used in this context due to the evolutionary pressures that have occurred in the mean time. Its just not the right tool – if the system were a well controlled animal model, I would feel differently.

5. In general there is a lot that is being assumed.

- For example, it is unclear what the basis for the assumption that a primary tumor population with EMT evolved after nodal metastasis. It could also be that the authors' hypothesis is simply wrong, but they don't acknowledge this possibility. In some sense the authors have put themselves in a bind – on the one hand they argue the primary is representative of any early time point in tumor biology yet on the other hand when the data don't fit their model, they say it must be because the tumor evolved. This inconsistency is very tough to evaluate the results and emphasizes the points above about issues with pseudotime (or making comments about tumor evolution) in a static biological specimen.

> Again, would emphasize 3 patterns across 5 tumors is hardly robust enough to make conclusions. The Puram paper actually concluded the opposite from my read of things -- that there were no consistent and major differences found between LN and primary malignant cells that were shared across even 3 of the 5 tumors.

- I would raise the same concerns about the T-cell analysis. There are many poorly executed papers that use this approach and I have yet to see a finding here that is robust and biologically validated in cancer. The same issue of making broad assumptions extends to these analyses where the authors presume there is a "burst" of CD8+ tumor specific T-cells without doing the clonal analysis to support that.

> It is strange to me to show 5 clonotypes and then show that one or two are specific to metastatic contexts as strong evidence of this phenomenon. The real question is what is the distribution of the top 20 clonotypes in primary vs LN and how do these change. The authors still haven't done this rigorously. Also the data from HN272 suggests metastatic clonotypes are not dysfunctional yet HN257 shows the opposite. The data are inconsistent and the problem again is the use of very small sample number which makes it very tough to make conclusions.

6. Generating tumor derived cultures is challenging in my experience for HNSCC and I reviewed the methods outlined here which seem rather simple. I would therefore think it is important to confirm that these tumor derived cultures are in fact malignant cells with WES to confirm CNAs are present and match the inferred CNAs by Copykat or something similar. Otherwise, many of the signatures could represent merely fibroblasts being present in one sample vs another, or alternatively genetic drift in the derived cultures.

> I suggested the cultures used here be confirmed to be similar to the original tumors but the authors merely mention their prior experience. I don't see the data showing that the functional work in this paper is on lines that match.

7. Finding one or two genes of interest such as AXL and AURK misses the real story and biology. Single genes being identified and studied by scRNA-seq is notoriously poor as an approach – the authors should be looking for more robust signatures. I would expect other members of these pathways to be modulated but I do not see that data shown or presented. As a small aside, I would note that AXL has been shown to play a role in metastasis in HNSCC previously and I am unclear what the added value is of the findings shown here that goes beyond what's been done.

> I don't agree. Single cell is plagued by dropouts and other challenges. Identifying a few genes with no cohesive biology or program is not rigorous and lacks any sort of new insight. The finding of AXL that the authors found had previously been show to play a role in metastasis, so nothing new there.

8. AXL is specifically higher in metastatic cells in HN137 according to the authors yet AXL inhibitors have an effect on inhibition in both pri and met. This goes against the authors conclusions and findings.

> Agree that its reasonable to expect gradation in AXL. But the effect seen appear to be similar in both primary and LN, suggesting there is no specificity to the observation of AXL being higher in metastatic cells.

9. Sox4 finding suffers from many of the same issues as above. A single gene is not proof of a coordinated state or program. There is very little here to suggest that Sox4 is the critical driver of these states. Data from cSCC which is used is not relevant given the distinct biology of these tumors (for example PD1 response in cSCC is 47% while for mucosal it is 14%). Clearly these tumors are different, especially as it relates to the immune system and immune exhaustion. Along these lines, the FACS analysis with Sox4 KD is not all it seems to me. PD1 might be lower in these cells but the differences are not that strong while other markers of exhaustion such as LAG3 are not changed. Is there actually a coordinated difference in exhaustion beyond a single gene? Have the authors confirmed these cells are functionally exhausted? Also on a more of a gut check level, there are thousands of studies on T-cell exhaustion and many on Sox4, yet no other study has seen such a finding. This seems more likely to reflect an issue with the authors' approach than an unexpected finding.

> While there is variability in PD1 response rates most studies report 14-22% and even with the best integration of CPS, TMB, etc its still very challenging to predict – all of that is true. I would challenge the authors to show me the evidence that shows an ability to predict higher response rates (in a prospective study not retrospective context which is mired with other flaws). Although the authors are limited by data available, I would again emphasize that cSCC is a COMPLETELY different entity than mucosal HNSCC. The divergent outcomes of patients, response to immunotherapy, etc are proof enough of this.

> The authors acknowledge my point that the changes with Sox4 are very very modest and hard to get excited about from my perspective. The marker expression does not reveal much about functional status as has been acknowledged, and that is the fundamental problem.

> In some way this most exciting finding is the most modest and least well supported. Even the KD experiments are being done in PBMCs rather than TILs. While I agree that lack of orthogonal validation does not negate a discovery, it gives pause of whether the observation is robust and reproducible.

10. "There was a progressive increase in clonality across the dysfunctional gradient, with evidence of single naïve or TRM-derived clones subsequently expanding to give rise to multiple dysfunctional clones that span these trajectories." I see a cartoon but don't see data supporting this statement. These trajectories also add very little in my mind in terms of our understanding of tumor biology more fundamentally as they are all over the place from sample to sample and not much is seen consistently across the dataset (e.g. Fig 5F).

> As above

11. Although the authors analyze 14 tumors, they spend the majority of their efforts analyzing just 2-3 tumors for any given portion of their study. They also jump around between different tumors from section to section (e.g. tumor analyzed for EMT in mets vs pri vs T-cells). This raises questions about just how generalizable these findings are and how the observations being made fit into the broader dataset of the authors' own cohort.

> While tumors are of course different the authors miss my point here – what is troubling is that the tumors analyzed vary across sections of the paper. Presumably there are samples that are good quality and these should be used for all analyses, yet the authors only pick 2 or 3 of the 14 tumors to analyze in each section. This weakens their approach, statistics, and credibility. I do not think this is sufficiently rigorous to make any conclusions. Why not analyze all tumors by the trajectory analyses for EMT? If robust, surely it should be shown in more than just two tumors? Similar comment for T-

cells. The authors didn't try what I suggested.

> As a concrete example, they analyze two tumors in Fig 2, but then use 3 completely different tumors for the validation in Fig 3. This does not make any sense! The same tumors in Fig 3 should be analyzed to show if they look similar in Fig 2 and the tumors analyzed deeply in Fig 2 should be used for validation in Fig. 3

> Then surprisingly we move to a whole different set of tumors in Fig 5. Again, why not just analyze all 14 samples? I suspect the reason is because the effects aren't seen in most of the other samples and thus the authors do not do this. Even though there is an argument made about extending other datasets, here too the authors simply pick a few tumors from that dataset to show their results rather than identifying robust patterns across a subset of the cohort that is reasonably sized (i.e. not just 2-3 tumors).

12. I am not sure what the pre-nodal subpopulation is defined as and this label seems highly arbitrary/dubious. Also, why only analyze 3 tumors out of 14 and 2 out of 5 from the prior published dataset? The explanation of "minimum RNA-seq" does not make sense given that the prior dataset had thousands of genes detected per cell. I recommend the authors perform the CellChat analysis on all tumors from both datasets to show their findings are robust. I would also note that here again the authors use tumors that are entirely different than those used for other analyses (both from their dataset as well as the other study).

> Thank you for trying CellChat – agree that minimum cell numbers are needed. If you didn't have that then it's not possible.

> The authors misunderstand my statement – what I meant was that Fig 6 uses an entirely different set of tumors from THEIR dataset than other sections and that the choice of tumors from the Puram paper is also different than other sections of the paper (p17 and p18).

> I'd like to clearly list the tumors used in each section so there is no miscommunication to highlight my concern:

- Fig 1: all tumors (7)

- Fig 2: HN251 and HN279

- Fig. 3: HN137, HN159, HN220

- Fig 4 and 5: HN272, HN258, HN263, HN272, HN257

- Fig 6: HN251, HN272, and HN279

I am completely shocked by the lack of overlap (almost entirely) between these different sections of the paper. It is highly unconventional and takes the authors' limited sample size and makes it even smaller. This raises concerns of 1) why aren't all high quality tumors being included in these analyses to really see what's robust and consistent 2) why are validation studies being done in tumors different from the ones analyzed.

13. Although differences in AURKB and TOP2A might be different after pembrolizumab (6E), the magnitude of difference is very modest. It should be easy to test the proliferation of these cells directly and demonstrate there is a difference both functionally and by IHC with Ki67. Also, this experiment is missing the critical control of cell that aren't pre-nodal. Presumably based on the authors hypothesis there should be a difference in PD1 response but this is not reported which is worrisome. In addition, the idea that MDK-driven suppression might suppress ICB seems interesting but also potentially true-true and unrelated. The authors haven't done the leg work to explain this expectation and backed it up with data. Finally, I would hardly describe a single humanized mouse model with a single tumor engrafted as adequate for the broad statements being made such as "these results implicate MDK-276 signaling as a pathway through which pre-nodal cells evade CD8-mediated immune-editing."

> I appreciate the authors clarifications here

Reviewer #2:

Remarks to the Author:

The authors perfectly addressed the question raised.

The only remaining part would be to dig into the clusters identified and show whether or not given known antigen specificities could be detected. It is likely that no particular specificities would be identified but by interrogating public databases, authors should be able to provide emphasis on this. This would strengthen the final message of dysfunction in the infiltrating cell driven by circulation. I now recommend the manuscript for publication.

Reviewer #4:

Remarks to the Author:

The Authors have addressed the comments I made to my satisfaction and in my opinion have addressed the criticisms of other referees well.

Dear Reviewers,

Thank you for your detailed review and comments in response to our submitted manuscript entitled “Single cell analysis of cancer cells and CD8+ T cells during early metastasis identifies targetable tumor subpopulation and mechanisms of immune evasion in squamous cell cancers”. We are heartened by the positive comments below and have provided a point-by-point response to specific queries raised, and with the associated changes, have undoubtedly improved the quality of this manuscript. We hope you find these satisfactory and look forward to hearing from you.

Reviewer #1 (Remarks to the Author):

I re-reviewed “Single cell analysis of early metastasis identifies targetable tumor subpopulation and 2 mechanisms of immune evasion in squamous cell cancers” by Quah et al. Unfortunately, I still have concerns about the methods, conclusions remaining poorly supported, and assumptions being made. I do not feel there are robust conclusions here and the authors continue to shift between different subsets of tumors which is worrisome. In addition, I do not find the identification of 5 unique genes (AXL, AURKB, STAT2, REL, RXRA, etc) which have no other coherent biological significance compelling as something unique to drive a “pro-metastatic” state.

MAJOR

1. Median number of genes detected is very very low. Most studies we would expect ~2000 genes per cell. This raises concerns about the quality of the data obtained here.

> Appreciate the efforts here. With that said, I think the Kurten data is also similarly weak due to the low number of expected genes. Agreed that T-cells tend to have lower detected genes but epithelial genes should really have >2000 unique genes. I couldn't disagree more strongly with the statement “many researchers now believe that reducing the cutoff to 200-300 genes per cell allows the capture of data from a broader range of cells.” I would challenge the authors to find Nature, Cell, and Science papers with median detected genes in the range they show. 20% mitochondrial cutoff is also quite high. I would typically expect this to be closer to 10%. Regardless, the authors should take a stringent set of cutoffs and show their data to be robust.

We have listed in Appendix 1 below a number of different publications, information from the 10x genomics site (Table 1) and even a comparative table of a number of papers (Table 2) that have been published. Most researchers accept approximately 1000+ genes per cell for 10x based single cell sequencing, and this is especially variable for solid tumors (with as low as 500+ for CRC). The minimum number of genes per cell is as low as 200, and mitochondrial percentages as high as 20%, as acceptable cutoffs in high impact journals. This is by no means exhaustive but I feel illustrates our point on this critical issue. We do not view Kurten's work to be weak because it had gone through the rigorous and professional peer-reviewed process in Nature Communication before it was deemed fit to be accepted for publication.

2. “95% of tumors had aneuploidy” – this is very surprising and unexpected. Most head and neck tumors have gain or loss of an arm, but not an entire chromosome and this finding is inconsistent with head and neck TCGA, again raising concerns about data analysis. I think the authors mean 95% of epithelial cells have CNAs, in which case that makes more sense to me.

> I would still avoid term aneuploidy being used interchangeably with CNA, just say CNA.

This is an excellent point. We truly apologize for this. We had always (wrongly) assumed that aneuploidy was interchangeable with CNA. As requested, we have removed the term “aneuploidy” and used “with copy number alteration (CNA)” throughout the manuscript.

Many thanks for pointing out this error.

3. I do not really appreciate or see the site specific clustering mentioned for HN251 and HN279 – on the UMAP these are hardly distinct clusters and a much more rigorous approach would be needed to show these clusters are robust. For example, do the authors see a distinct cluster by NMF or a similar approach? And does that cluster appear to be primarily from the mets vs pri? What are the top differentially expressed genes if so?
> Again appreciate the efforts and clarifications. I suggested the authors utilize NMF within a sample to show these clusters are robust, but this was not done. PCA across samples doesn't convince me that there are unique clusters. Finally, the finding of a mesenchymal PC1 and epithelial PC2 as already reported previously in the Puram paper the authors mention.

Thank you for the comment. In line with the reviewer's comment, we performed NMF but the analysis did not show as clean a distribution as the UMAP plot did. Hence, we showed PCA and tSNE instead, which we believe would be sufficient. We are happy to revisit the NMF analyses here, but again this is not a critical statement as we did not base our subsequent analysis on this method of separation in identification of pre-metastatic cells or the genes derived there-in.

4. There is a fundamental problem with using trajectories of any sort with the experimental paradigm as presented. Primary tumors do not represent an evolutionarily "earlier" time point because they are continuing to evolve at the same time as the LN metastasis forms (patients have surgery as a single time point where both are collected). Thus the statement "primary tumor presumed to pre-date nodal disease" is fundamentally flawed. That is why time-based trajectories with a single chronologic time point are not really meant to be used in the fashion utilized here. This is emphasized by the unclear separation of primary and metastatic states seen by the authors. If the authors really saw clear differences in mets vs primary with EMT or other signatures, then that's perhaps interesting but their data suggest these differences are minor at best. The authors use an orthogonal dataset to help support their points, but even here only 3/5 tumors suggest "more EMT" while the other 2/5 showed primary to have more EMT and these differences are rather underwhelming overall. Hardly, the type of robust orthogonal validation one would hope for.
> While am sensitive to the issue of samples, it is hard to make any conclusion from what has been shown. The authors suggest there are 3 evolutionary patterns. But maybe these 3 differences are not biologically significant and just represent the random variation. We cannot know if this is significant or not because there are just too few samples. This makes it very hard to know if the conclusions are biologically interesting and relevant or just simple anecdotes. 5 mice were injected with tumors and showed 3 different patterns of growth, say one with no tumors, one with medium tumor size, and one group with large tumor size, I hardly think anyone would say there must be 3 different "types" of tumor growth. Rather the first thing would be to have more samples, and I think the sample applies here.
> I also do not believe that trajectory analysis can be reliably used in this context due to the evolutionary pressures that have occurred in the mean time. Its just not the right tool – if the system were a well controlled animal model, I would feel differently.

We thank the reviewer for the comment. Our attempt here is to look backwards in time using data from a fixed point in time, with the necessary assumptions, and this is not dissimilar to other genomics evolutionary methodologies. Our trajectory analyses actually suggest 2 major types of evolution- one where sense can be made using the trajectory tools (5/7 tumors), and another where these tools do not help as evolution is likely too rapid or has 'moved on' (ie. HN257). Apart from our own dataset, these are replicated on an independent dataset (3/7 from our own vs 2/5 from Puram's dataset). That is our major premise before going into details on pre- metastatic population and genes involved.

5. In general there is a lot that is being assumed.

- For example, it is unclear what the basis for the assumption that a primary tumor population with EMT evolved after nodal metastasis. It could also be that the authors' hypothesis is simply wrong, but they don't acknowledge this possibility. In some sense the authors have put themselves in a bind – on the one hand they argue the primary is representative of any early time point in tumor biology yet on the other hand when the data don't fit their model, they say it must be because the tumor evolved. This inconsistency is very tough to evaluate the results and emphasizes the points above about issues with pseudotime (or making comments about tumor evolution) in a static biological specimen.
> Again, would emphasize 3 patterns across 5 tumors is hardly robust enough to make conclusions. The Puram paper actually concluded the opposite from my read of things -- that there were no consistent and major differences found between LN and primary malignant cells that were shared across even 3 of the 5 tumors.

We agree with the reviewer that this is partially correct, that more numbers are required to find recurrent genes and themes. In line with the reviewer's comment, this is why we are happy to find recurrences among our tumors, Puram's tumors, our cancer cell cultures and existing literature on AXL, AURK etc.

- I would raise the same concerns about the T-cell analysis. There are many poorly executed papers that use this approach and I have yet to see a finding here that is robust and biologically validated in cancer. The same issue of making broad assumptions extends to these analyses where the authors presume there is a "burst" of CD8+ tumor specific T-cells without doing the clonal analysis to support that.

> It is strange to me to show 5 clonotypes and then show that one or two are specific to metastatic contexts as strong evidence of this phenomenon. The real question is what is the distribution of the top 20 clonotypes in primary vs LN and how do these change. The authors still haven't done this rigorously. Also the data from HN272 suggests metastatic clonotypes are not dysfunctional yet HN257 shows the opposite. The data are inconsistent and the problem again is the use of very small sample number which makes it very tough to make conclusions.

We thank the reviewer for this comment. The immune context and development certainly do not seem to mirror tumor cell evolution. This was one of our hypotheses that was busted early (unfortunately, as I had hoped that the immune evolution of T cell could follow the patterns of tumor cell evolution, which has never been shown in any case). Hence, I am not surprised at the 'inconsistencies' between CD8+ T cell development in HN272 vs HN257. We show the top 5 clonotypes instead the top 20 as we feel this is sufficient and will overwhelm the already-dense diagram. In fact, as suggested by reviewer #2, we performed GLIPH2 analysis of ALL clonotypes from HN251, HN263 and HN272, and the table in Supplementary Table S7 shows the extensive result from the analysis. If required, we are more than happy to show the top 20 clonotypes as a Supplementary Table.

6. Generating tumor derived cultures is challenging in my experience for HNSCC and I reviewed the methods outlined here which seem rather simple. I would therefore think it is important to confirm that these tumor derived cultures are in fact malignant cells with WES to confirm CNAs are present and match the inferred CNAs by Copykat or something similar. Otherwise, many of the signatures could represent merely fibroblasts being present in one sample vs another, or alternatively genetic drift in the derived cultures.

> I suggested the cultures used here be confirmed to be similar to the original tumors but the authors merely mention their prior experience. I don't see the data showing that the functional work in this paper is on lines that match.

Thank you for the comment. We have listed a number of our data for the tumor derived cultures (cancer cell cultures) in the reference sections, and a number of these are currently being prepared for a future submission, we would be happy to cite or show this data. Our lab at the moment has between 40-45 patient derived cancer cell cultures with robust matching to primary tumors.

7. Finding one or two genes of interest such as AXL and AURK misses the real story and biology. Single genes being identified and studied by scRNA-seq is notoriously poor as an approach – the authors should be looking for more robust signatures. I would expect other members of these pathways to be modulated but I do not see that data shown or presented. As a small aside, I would note that AXL has been shown to play a role in metastasis in HNSCC previously and I am unclear what the added value is of the findings shown here that goes beyond what's been done.

> I don't agree. Single cell is plagued by dropouts and other challenges. Identifying a few genes with no cohesive biology or program is not rigorous and lacks any sort of new insight. The finding of AXL that the authors found had previously been shown to play a role in metastasis, so nothing new there.

Please see our response to this comment under point 8.

8. AXL is specifically higher in metastatic cells in HN137 according to the authors yet AXL inhibitors have an effect on inhibition in both pri and met. This goes against the authors conclusions and findings.

> Agree that its reasonable to expect gradation in AXL. But the effect seen appear to be similar in both primary and LN, suggesting there is no specificity to the observation of AXL being higher in metastatic cells.

Although we respect the reviewer's view, we feel these points are somewhat contradictory. AXL is well known to be involved in metastasis in the literature, but clearly can only be implicated in a fraction of head and neck cancers, and even so, in a subpopulation of the tumor. That is the point we are bringing across here, similarly with the other genes listed.

9. Sox4 finding suffers from many of the same issues as above. A single gene is not proof of a coordinated state or program. There is very little here to suggest that Sox4 is the critical driver of these states. Data from cSCC which is used is not relevant given the distinct biology of these tumors (for example PD1 response in cSCC is 47% while for mucosal it is 14%). Clearly these tumors are different, especially as it relates to the immune system and immune exhaustion. Along these lines, the FACS analysis with Sox4 KD is not all it seems to me. PD1 might be lower in these cells but the differences are not that strong while other markers of exhaustion such as LAG3 are not changed. Is there actually a coordinated difference in exhaustion beyond a single gene? Have the authors confirmed these cells are functionally exhausted? Also on a more of a gut check level, there are thousands of studies on T-cell exhaustion and many on Sox4, yet no other study has seen such a finding. This seems more likely to reflect an issue with the authors' approach than an unexpected finding.

> While there is variability in PD1 response rates most studies report 14-22% and even with the best integration of CPS, TMB, etc its still very challenging to predict – all of that is true. I would challenge the authors to show me the evidence that shows an ability to predict higher response rates (in a prospective study not retrospective context which is mired with other flaws). Although the authors are limited by data available, I would again emphasize that cSCC is a COMPLETELY different entity than mucosal HNSCC. The divergent outcomes of patients, response to immunotherapy, etc are proof enough of this.

> The authors acknowledge my point that the changes with Sox4 are very very modest and hard to get excited about from my perspective. The marker expression does not reveal much about functional status as has been acknowledged, and that is the fundamental problem.

> In some way this most exciting finding is the most modest and least well supported. Even the KD experiments are being done in PBMCs rather than TILs. While I agree that lack of orthogonal validation does not negate a discovery, it gives pause of whether the observation is robust and reproducible.

Thank you for this comment. Even in an artificial system of PBMC's we see an effect, not to mention its recent identification in a large cohort in the pan-cancer context. Despite the modest knock down effect of SOX4, we should not dismiss the potential compounding effect(s) due to reduction of expression of multiple markers involved in dysfunction. If the reviewer would like to see this repeated in TILs-tumor pairs, we are happy to attempt these experiments (we have the reagents for these at the moment).

10. "There was a progressive increase in clonality across the dysfunctional gradient, with evidence of single naïve or TRM-derived clones subsequently expanding to give rise to multiple dysfunctional clones that span these trajectories." I see a cartoon but don't see data supporting this statement. These trajectories also add very little in my mind in terms of our understanding of tumor biology more fundamentally as they are all over the place from sample to sample and not much is seen consistently across the dataset (e.g. Fig 5F).
> As above

Please see our response above.

11. Although the authors analyze 14 tumors, they spend the majority of their efforts analyzing just 2-3 tumors for any given portion of their study. They also jump around between different tumors from section to section (e.g. tumor analyzed for EMT in mets vs pri vs T-cells). This raises questions about just how generalizable these findings are and how the observations being made fit into the broader dataset of the authors' own cohort.
> While tumors are of course different the authors miss my point here – what is troubling is that the tumors analyzed vary across sections of the paper. Presumably there are samples that are good quality and these should be used for all analyses, yet the authors only pick 2 or 3 of the 14 tumors to analyze in each section. This weakens their approach, statistics, and credibility. I do not think this is sufficiently rigorous to make any conclusions. Why not analyze all tumors by the trajectory analyses for EMT? If robust, surely it should be shown in more than just two tumors? Similar comment for T-cells. The authors didn't try what I suggested.
> As a concrete example, they analyze two tumors in Fig 2, but then use 3 completely different tumors for the validation in Fig 3. This does not make any sense! The same tumors in Fig 3 should be analyzed to show if they look similar in Fig 2 and the tumors analyzed deeply in Fig 2 should be used for validation in Fig. 3
> Then surprisingly we move to a whole different set of tumors in Fig 5. Again, why not just analyze all 14 samples? I suspect the reason is because the effects aren't seen in most of the other samples and thus the authors do not do this. Even though there is an argument made about extending other datasets, here too the authors simply pick a few tumors from that dataset to show their results rather than identifying robust patterns across a subset of the cohort that is reasonably sized (i.e. not just 2-3 tumors).

Please see our response to this comment under point 12.

12. I am not sure what the pre-nodal subpopulation is defined as and this label seems highly arbitrary/dubious. Also, why only analyze 3 tumors out of 14 and 2 out of 5 from the prior published dataset? The explanation of "minimum RNA-seq" does not make sense given that the prior dataset had thousands of genes detected per cell. I recommend the authors perform the Cellchat analysis on all tumors from both datasets to show their findings are robust. I would also note that here again the authors use tumors that are entirely different

than those used for other analyses (both from their dataset as well as the other study).
> Thank you for trying CellChat – agree that minimum cell numbers are needed. If you didn't have that then its not possible.

> The authors misunderstand my statement – what I meant was that Fig 6 uses an entirely different set of tumors from THEIR dataset than other sections and that the choice of tumors from the Puram paper is also different than other sections of the paper (p17 and p18).

> I'd like to clearly list the tumors used in each section so there is no miscommunication to highlight my concern:

- Fig 1: all tumors (7)
- Fig 2: HN251 and HN279
- Fig. 3: HN137, HN159, HN220
- Fig 4 and 5: HN272, HN258, HN263, HN272, HN257
- Fig 6: HN251, HN272, and HN279

I am completely shocked by the lack of overlap (almost entirely) between these different sections of the paper. It is high unconventional and takes the authors' limited sample size and makes it even smaller. This raises concerns of 1) why aren't all high quality tumors being included in these analyses to really see what's robust and consistent 2) why are validation studies being done in tumors different from the ones analyzed.

We appreciate and respect the reviewer's comments. We feel Point 11 and 12 are not entirely accurate. We have listed the samples used for each figure in Appendix 1, Table 3 for better clarity. Apart from Figure 3, which was always about using a separate/independent cancer cell cultures (and hence 7 separate patient-derived cancer cell culture pairs from primary and metastatic tumors), the rest were based on 7 tumor pairs. All seven tumors pairs were used for Figure 1. While some had to be dropped due quality control expectations in each algorithm used for subsequent analyses, Figure 2 (including supplementary henceforth) showed 6/7 tumor pairs; Figure 4 showed all 7 tumor pairs; Figure 5 was limited to 6 pairs due to TCRseq dropout in HN237; Figure 6 suffered the most as we set a minimum cut-off number of primary and pre-met cells with CD8+T cell to run an accurate interactome analyses. This approach to start with a larger number of tumors and then narrow down to a few to make a point, and then validating these functionally is commonly used across many papers that are not merely landscaping papers.

13. Although differences in AURKB and TOP2A might be different after pembro (6E), the magnitude of difference is very modest. It should be easy to test the proliferation of these cells directly and demonstrate there is a difference both functionally and by IHC with Ki67. Also, this experiment is missing the critical control of cell that aren't pre-nodal. Presumably based on the authors hypothesis there should be a difference in PD1 response but this is not reported which is worrisome. In addition, the idea that MDK-driven suppression might suppress ICB seems interesting but also potentially true-true and unrelated. The authors haven't don't the leg work to explain this expectation and backed it up with data. Finally, I would hardly describe a single humanized mouse model with a single tumor engrafted as adequate for the broad statements being made such as "these results implicate MDK-276 signaling as a pathway through which pre-nodal cells evade CD8-mediated immune-editing."

> I appreciate the authors clarifications here

Thank you very much for this.

Reviewer #2 (Remarks to the Author):

The authors perfectly addressed the question raised.

The only remaining part would be to dig into the clusters identified and show whether or not given known antigen specificities could be detected. It is likely that no particular specificities would be identified but by interrogating public databases, authors should be able to provide emphasis on this.

This would strengthen the final message of dysfunction in the infiltrating cell driven by circulation.

I now recommend the manuscript for publication.

We really appreciate the effort and helpful comments/suggestions contributed by this reviewer to improve the manuscript. Thank you very much.

Reviewer #4 (Remarks to the Author):

The Authors have addressed the comments I made to my satisfaction and in my opinion have addressed the criticisms of other referees well.

We really appreciate the effort and helpful comments/suggestions contributed by this reviewer to improve the manuscript. Thank you very much.

Appendix 1

Table 1. Sequencing and performance metrics adapted from the official 10x Genomics website and application note. See row highlighted in red box. Adapted from https://support.10xgenomics.com/single-cell-vdj/software/tutorials/tme_lcb_lvb and https://pages.10xgenomics.com/rs/446-PBO-704/images/10x_AN022_IP_TumorMicroenvironment_digital.pdf

Sample	CRC			NSCLC		
	Gene Expression	TCR-V(D)J	Ig-V(D)J	Gene Expression	TCR-V(D)J	Ig-V(D)J
Number of Reads	668 M	48 M	10 M	376 M	43 M	44 M
Estimated Number of Recovered Cells	8,400	933	706	7,802	1,993	3,077
Fraction of Reads in Cells	95.20%	45.4%	92.5%	96.8%	70.9%	95.0%
Mean Reads per Cell	79,522	51,321	13,873	48,196	21,438	14,172
Fraction of Reads Mapped to Target (exonic regions in 5' gene expression or any TCR/Ig gene in V(D)J data)	57.2%	56.8%	97.9%	76.3%	80.0%	98.1%
Median Genes per Cell	585	NA	NA	1,442	NA	NA
Cells with Productive V-J Spanning Pair	NA	387	371	NA	1,476	1,410

Table 2. A few examples of quality control (QC) criteria used by authors to analyse single cell RNAseq (10x Chromium). These are examples extracted from papers published in reputable peer-reviewed journals. In summary, the QC criteria of ≥ 100 and ≤ 8000 genes, and $>\sim 5\text{-}20\%$ mitochondrial RNA content, were used to filter out low-quality cells.

	Journal title	Information	Author	Journal
1	A Single-Cell Sequencing Guide for Immunologists	This platform is currently able to detect 500–1,500 genes per primary cell (Figure S1) Fig. 3: PBMCs sensitivity. <small>From: Systematic comparison of single-cell and single-nucleus RNA-sequencing methods</small> a. Distribution of the number of UMIs (a) or genes (b) per cell for each method in the two experiments ($n = 1$ biologically independent sample per experiment). Violin and box plot elements are defined as in Fig. 2. Source data Adapted from Figure 3	Peter See et al	Frontiers in immunology

	Journal title	Information	Author	Journal
2	Single-Cell Analyses Inform Mechanisms of Myeloid-Targeted Therapies in Colon Cancer	As expected, the Smart-seq2 platform captured more genes, including cytokines, CD molecules, ligands/receptors and transcription factors, and exhibited weak batch effects compared to the 10x scRNA-seq platform (Figures S1B–S1F).  Adapted from Figure S1B	Zhang L et al	Cell
3	Systematic comparison of single-cell and single-nucleus RNA-sequencing methods	Among the high-throughput methods, 10x Chromium (v3) had the highest median number of UMIs (4,494) and genes (1,482) per cell (Fig. 3), and inDrops (366 and 1,118 UMIs; 256 and 568 genes) and Seq-Well (844 and 577 UMIs; 513 and 372 genes) had the lowest (Fig. 3).  Adapted from Figure 3	Jiarui D et al	Nature Biotechnology

	Journal title	Information	Author	Journal
4	Coupled scRNA-Seq and Intracellular Protein Activity Reveal an Immunosuppressive Role of TREM2 in Cancer	The mean reads per cells varied from 13,480 and 353,472 with median UMI of 561 to 8092 per cell. Low-quality cells were discarded if the number of expressed genes was smaller than 300. Cells were also removed if their mitochondrial gene expression were larger than 10 percent. See Figure 1D  Adapted from Figure 1D	Katzenelenbogen Y et al	Cell
5	Pan-cancer single-cell landscape of tumor-infiltrating T cells	200 genes detected or >10% mitochondrial UMI 25 counts were filtered out; genes detected in > 3 cells were kept	Zheng L et al	Science
6	Cross-tissue organization of the fibroblast lineage	filtered low quality cells with <500 measured genes and a high percentage of mitochondrial contamination (>~5–20%, depending on the dataset)	Buechler MB	Nature
7	CRISPR-engineered T cells in patients with refractory cancer	A minimum of 250 genes detected / cell and maximum of 20% mitochondrial reads were used to exclude low-quality cells	Stadtmauer EA et al	Science
8	Clonal replacement of tumor-specific T cells following PD-1 blockade	On average, we obtained reads from 1,862 genes per cell (median: 1,716) and 6,304 unique transcripts per cell (median: 4,777). Cells with less than 200 genes detected or greater than 10% mitochondrial RNA content were excluded from analysis	Yost KE et al	Nature Medicine

	Journal title	Information	Author	Journal
9	c-Jun overexpression in CAR T cells induces exhaustion resistance	first selected genes with at least one unique molecular identifier (UMI) counts in any given cell. we selected genes expressed in ≥ 50 cells. Single live cells were selected as droplets expressing ≥ 500 genes with $\leq 20,000$ UMI counts and $\leq 10\%$ mitochondrial reads.	Lynn R et al	Nature
10	Differences in Tumor Microenvironment Dictate T Helper Lineage Polarization and Response to Immune Checkpoint Therapy	For each sample, genes that were expressed in less than 3 cells, cells that expressed < 200 genes or > 5000 genes, and cells with mitochondrial genes constituting $> 5\%$ were discarded.	Jiao S et al	Cell
11	Onco-fetal Reprogramming of Endothelial Cells Drives Immunosuppressive Macrophages in Hepatocellular Carcinoma	cells are filtered based on the criteria of expressing a minimum of 200 genes and a gene which is expressed by a minimum of 30 cells. Dying cells with a mitochondrial percentage of more than 5% were excluded.	Sharma A et al	Cell
12	Single-cell landscape of bronchoalveolar immune cells in patients with COVID-19	The following criteria were then applied to each cell of all nine patients and four healthy controls: gene number between 200 and 6,000, UMI count $> 1,000$ and mitochondrial gene percentage < 0.1 . After filtering, a total of 66,452 cells were left for the following analysis.	Liao M et al	Nature Medicine
13	Single-cell analysis reveals new evolutionary complexity in uveal melanoma	Filtering was conducted by retaining cells that had unique molecular identifiers (UMIs) greater than 400, expressed 100 and 8000 genes inclusive, and had mitochondrial content less than 10 percent. No sample batch correction was performed. This resulted in a total of 59,915 cells	Durante MA	Nature Communications

	Journal title	Information	Author	Journal
14	Investigating immune and non-immune cell interactions in head and neck tumors by single-cell RNA sequencing	After quality control, 134,606 cells with 1077 median genes per cell were retained. Based on the QC metrics suggested in the Scanpy tutorial¹⁷, cells with less than 200 genes expressed were filtered out. Cells expressing more than 5000 genes, and more than ten percent mitochondrial genes were also removed to ensure only the high quality of cells used in the downstream analyses. Genes expressed in less than 3 cells were also filtered out of the analysis.	Kurten CHL et al	Nature Communications
15	Immune suppressive landscape in the human esophageal squamous cell carcinoma microenvironment	Low-quality cells (<400 genes/cell and >10% mitochondrial genes) were excluded. As a result, 80,787 cells with a median of 1170 detected genes per cell were included in downstream analyses.	Zheng Y et al	Nature Communications

Table 3. Samples used for analysis in each main and supplementary figure.

	What reviewer #1 claimed		What our manuscript showed		Remarks
	Tumors	Cancer cell culture	Tumors	Cancer cell culture	
Main and Supplementary Figure 1	All tumors (7 pairs of tumors from 7 patients)	Not applicable	All tumors (7 pairs of tumors from 7 patients)	Not applicable	Not applicable
Main and Supplementary Figure 2	HN251 HN279	Not applicable	HN251 HN279 HN257 HN242 HN257 HN272 Puram's p26 Puram's p28 Puram's p25 Puram's p5 Puram's p20	Not applicable	Apart from HN237 which was not analysed due to low cell number, the rest of 6 tumor pairs were used for the analysis. All of Puram's 5 pairs of tumors (pri and met) were also used for the analysis.
Main and Supplementary Figure 3	Not applicable	HN137 HN159 HN220	Not applicable	HN137 HN159 HN220 HN120 HN148 HN160 HN217	All 7 pairs of cancer cell cultures were used in the analysis to derive the UMAP and heatmap

	What reviewer #1 claimed		What our manuscript showed		Remarks
	Tumors	Cancer cell culture	Tumors	Cancer cell culture	
Main and Supplementary Figure 4 & 5	HN272 HN251 HN263 HN272 HN257	Not applicable	HN272 HN251 HN263 HN272 HN257 HN237 HN242 Yost KE et al's dataset Puram's CD8 dataset	Not applicable	All 7 tumors were used in the analysis to derive Figure 4. For Figure 5, 6 out of 7 tumors were used because HN237 had very few cells and thus did not contain enough TCRseq information to perform the analysis
Main and Supplementary Figure 6	HN251 HN272 HN279	Not applicable	HN251 HN272 HN279 HN237 HN242 HN263 HN257 Puram's p17 Puram's p18	Not applicable	All 7 tumors primaries were tested but only 3 had sufficient cell numbers to proceed with the CellChat analysis. All Puram's tumor primaries were tested but only 2 had sufficient cell numbers to proceed with the CellChat analysis.

Reviewers' Comments:

Reviewer #5:

Remarks to the Author:

I reviewed "Single cell analysis of early metastasis identifies targetable tumor subpopulation and 2 mechanisms of immune evasion in squamous cell cancers" by Quah et al., in light of comments from Reviewer #1 and the authors' responses to these comments. There are a number of points brought up by Reviewer #1 that are not adequately addressed by the authors. Overall, the big picture concerns raised by Reviewer #1 are not fundamentally addressed, either by new analyses or a revised discussion of conclusions and/or limitations of the study.

(1) I appreciate the authors' validation of cell classification by overlap with clustering of classified cells from Puram et al in the UMAP. I also appreciate the attempt of the authors to demonstrate that there are papers that accept lower numbers of genes per cell as adequate, as demonstrated in Appendix 1. However, the authors still do not describe their cutoffs and quality control metrics in selecting the final dataset of cells to be analyzed. In fact, they state in lines 78-79, "details on quality controls steps in Methods..." but I do not see these steps detailed in the Methods section. The authors should clearly describe their cutoffs for eliminating poor quality cells from the final dataset.

(2) Thank you for this clarification.

(3) I appreciate the authors efforts to perform NMF and demonstrate clustering in this way. However, these results still do not support their statement in lines 106-107: "...while patients HN251 (cluster 10 vs 11 nodal), HN279 (clusters 4,3,9 vs 5 nodal), and HN272 (clusters 0 vs 6 nodal) show distinct sub-clusters where nodal tumor cells predominate..." It appears that this statement really only applies to HN272, where cluster 0 is primarily cells from the primary tumor, while cluster 6 is primarily lymph node cells. For HN251, even cluster 11 is primarily cells from the primary tumor, and the limited number of cells overall from the lymph node makes it difficult to draw any conclusions about site specific clustering. In HN279, all four clusters have varying proportions of cells from the primary tumor vs. lymph node, and only one cluster (cluster 4) has >80% of cells from a single subsite, again calling into question the idea of site-specific clustering. The authors should revise they way in which they discuss these findings, particularly in light of the fact that coherent differences in malignant cell expression between primary tumors and lymph nodes have not previously been demonstrated in HNSCC.

(4) The authors reply that the "trajectory analyses actually suggest 2 major types of evolution" – however, it is clearly stated in the text, line 120, that "This approach identified three different patterns." These patterns are then discussed in detail in lines 121-144. The authors do not adequately assess the limitations of sample size here in drawing the conclusion that three such patterns exist. Given the limited numbers, it is alternatively possible that no coherent patterns exist and that lymph node metastasis is a passive drainage process that does not mark a biologically (and thus temporally) distinct phase of disease, but instead simply a marker of an aggressive primary tumor. The authors should consider this possibility in discussing these results. They should also adequately acknowledge the reviewer's concern about using trajectories with the experimental paradigm as presented – this should be emphasized as a limitation of the analysis presented.

(5) The authors do not adequately address the reviewer's concerns about the temporal relationship between primary tumors, nodal metastasis, and EMT. At minimum, this should be discussed as a potential negative result in discussion of limited sample size. I appreciate the new analysis of T cell clonotypes. However, the authors propose that clonal overlap between primary tumor and lymph node T cells indicate "the presence of similar antigens at both tumor sites" (line 245). The authors should be cautious about drawing such conclusions, as it is unclear whether overlapping clonotypes represent a passive movement/collective migration of T cells along with tumor cells from one site to the other or the active de novo generation of similar clonotypes at both sites, as proposed. The current

methodologies do not allow for the distinction of one mechanism from the other. Moreover, the most clonotypes do not overlap across sites, suggesting, in fact, that there may be differences in tumor antigens across sites (although this is not necessarily supported by other analyses).

(6) Thank you for this clarification.

(7) and (8) The authors continue to focus on single genes that are differentially expressed in individual tumors and have not discussed the relevance of AXL being a targetable marker of "pre-nodal" cells. In general, the authors have not clearly defined what is meant by "pre-nodal" cells or what the significance of these cells may be. Are the same actionable markers within pre-nodal cells (AXL, AURKB) that were found in patient derived cultures also present within fresh tumor samples? Are pre-nodal cells also thought to be present in tumors that have not yet displayed lymph node metastases?

(9) Thank you for the clarification and for the additional data. Still, the authors should better acknowledge that cSCC is a completely different entity than mucosal HNSCC, both clinically and biologically, and should not be used as a validation dataset for findings in HNSCC. cSCC findings may merely be used as supporting evidence for a general concept.

(10) Thank you for the clarification.

(11) and (12) Thank you for the clarification. I understand the limitations of data and need to pare down the dataset as analyses proceed. However, Figure 6 continues to use an entirely different set of tumors, particularly with regard to MDK inhibitor treatment and it is unclear why this is the case. Were these new tumors chosen for a particular reason?

Dear Reviewer and Editorial team,

Thank you for the detailed review and comments in response to our submitted manuscript entitled “Single cell analysis of cancer cells and CD8+ T cells during early metastasis identifies targetable tumor subpopulation and mechanisms of immune evasion in squamous cell cancers”. We are heartened by the constructive comments by reviewer #5 and have provided a point-by-point response to specific queries raised. We feel that we have answered and made the associated changes to the queries to improve the quality of this manuscript. We hope you find these adequate and look forward to hearing from you.

REVIEWER COMMENTS

Reviewer #5, expertise in single cell sequencing/head and neck cancer/immune cells to comment on your responses to Reviewer's #1 previous concerns (Remarks to the Author):

I reviewed “Single cell analysis of early metastasis identifies targetable tumor subpopulation and 2 mechanisms of immune evasion in squamous cell cancers” by Quah et al., in light of comments from Reviewer #1 and the authors’ responses to these comments. There are a number of points brought up by Reviewer #1 that are not adequately addressed by the authors. Overall, the big picture concerns raised by Reviewer #1 are not fundamentally addressed, either by new analyses or a revised discussion of conclusions and/or limitations of the study.

(1) I appreciate the authors’ validation of cell classification by overlap with clustering of classified cells from Puram et al in the UMAP. I also appreciate the attempt of the authors to demonstrate that there are papers that accept lower numbers of genes per cell as adequate, as demonstrated in Appendix 1. However, the authors still do not describe their cutoffs and quality control metrics in selecting the final dataset of cells to be analyzed. In fact, they state in lines 78-79, “details on quality controls steps in Methods...” but I do not see these steps detailed in the Methods section. The authors should clearly describe their cutoffs for eliminating poor quality cells from the final dataset.

>We truly apologize for not including the details upfront in the Methods section of the main text due to the word count limit, and therefore initially only included this in the Supplementary Methods, and hence the reference to this was not accurate. We have now amended the Methods section of the main text to include these details. In addition, we have added two figures showing the quality control parameters applied in this manuscript. These figures are now included as Supplementary Figure 1A and 1B, and herein as Rebuttal Figure 1.

Rebuttal Figure 1. (A) Distribution of each cell by number of genes and UMI as shown by scatterplots. The blue lines indicate the threshold of cutoffs being applied. (B) Violin plots showing the number of genes, number of UMI, and mitochondrial gene percentage in each cell, before (left) and after quality control (right).

(2) Thank you for this clarification.

>We thank you very much.

(3) I appreciate the authors efforts to perform NMF and demonstrate clustering in this way. However, these results still do not support their statement in lines 106-107: "...while patients HN251 (cluster 10 vs 11 nodal), HN279 (clusters 4,3,9 vs 5 nodal), and HN272 (clusters 0 vs 6 nodal) show distinct sub-clusters where nodal tumor cells predominate..." It appears that this statement really only applies to HN272, where cluster 0 is primarily cells from the primary tumor, while cluster 6 is primarily lymph node cells. For HN251, even cluster 11 is primarily cells from the primary tumor, and the limited number of cells overall from the lymph node makes it difficult to draw any conclusions about site specific clustering. In HN279, all four clusters have varying proportions of cells from the primary tumor vs. lymph node, and only one cluster (cluster 4) has >80% of cells from a single subsite, again calling into question the idea of site-specific clustering. The authors should revise they way in which they discuss these findings, particularly in light of the fact that coherent differences in malignant cell expression between primary tumors and lymph nodes have not previously been demonstrated in HNSCC.

>This is an excellent point raised by the reviewer and we realized after reading this point that we have gone about describing our data in the wrong way, even though our rational was completely in line with the reviewer is saying. In fact, in complete agreement with the reviewer, we were never convinced about the ability to identifying pre-nodal cells using these clustering methodologies, which is why we proceeded with a more rational approach involving trajectory analyses, EMT and Cytotrace (which is a readout of differentiation) to identify this subpopulation. In fact none of these clustering data analyses (UMAP, NMF or TSNE) were subsequently utilized. As such, we have revised how we described this data in the results section to reflect this as follows (Results section):

“Although patients HN251 (cluster 10 vs 11 nodal), HN279 (clusters 4,3,9 vs 5 nodal) and HN272 (clusters 0 vs 6 nodal) show sub-clusters where nodal tumor cells appear to predominate, these were not sufficiently robust to support the identification a distinct pre-nodal subpopulation. Similar findings can be seen using other algorithms such as NMF and TSNE”

(4) The authors reply that the “trajectory analyses actually suggest 2 major types of evolution” – however, it is clearly stated in the text, line 120, that “This approach identified three different patterns.” These patterns are then discussed in detail in lines 121-144. The authors do not adequately assess the limitations of sample size here in drawing the conclusion that three such patterns exist. Given the limited numbers, it is alternatively possible that no coherent patterns exist and that lymph node metastasis is a passive drainage process that does not mark a biologically (and thus temporally) distinct phase of disease, but instead simply a marker of an aggressive primary tumor. The authors should consider this possibility in discussing these results. They should also adequately acknowledge the reviewer’s concern about using trajectories with the experimental paradigm as presented – this should be emphasized as a limitation of the analysis presented.

>Thank you for raising this point and we do agree to these limitations, and as suggested by the reviewer have included the following statements to highlight these limitations (paraphrasing the reviewer’s own words) (Discussion section):

“Conversely, the complexity of these analyses highlights one of the limitations of these conclusions, which is examining a small number of tumors. An alternative hypothesis is that there is no coherent pattern and lymph node metastasis is merely a passive drainage process that does not mark a biologically (and thus temporally) distinct phase of disease, but instead a marker of an aggressive primary tumor. Differentiating these requires a well-controlled system including animal models that could capture this evolutionary trait dynamically, and this would be an important extension to validate our findings.”

(5) The authors do not adequately address the reviewer’s concerns about the temporal relationship between primary tumors, nodal metastasis, and EMT. At minimum, this should be discussed as a potential negative result in discussion of limited sample size. I appreciate the new analysis of T cell clonotypes. However, the authors propose that clonal overlap between primary tumor and lymph node T cells indicate “the presence of similar antigens at both tumor sites” (line 245). The authors should be cautious about drawing such conclusions, as it is unclear whether overlapping clonotypes represent a passive movement/collective migration of T cells along with tumor cells from one site to the other or the active de novo generation of similar clonotypes at both sites, as proposed. The current methodologies do not allow for the distinction of one mechanism from the other. Moreover, the most clonotypes do not overlap across sites, suggesting, in fact, that there may be differences in tumor antigens across sites (although this is not necessarily supported by other analyses).

>We apologize for overstatement and thank the reviewer for the explanation. We have thus toned down this statement to avoid drawing such a strong conclusion. This was done by:

- removing the statement referring to “similar antigens at both tumor sites”
- including a statement in the Discussion section as follows: “...although one of the limitations in this study is that these data neither precludes passive drainage of T-cells across lymphatic channels nor collective migration of T cells along with tumor cells from one site to the other.”

However, this line of enquiry prompted us to run further analyses on our shared clonotype data. To do this, we extracted only shared clonotypes that were detected in both the primary tumors and nodal mets for each patient. We then proceeded to analyze CD8+ cells with shared clonotypes in greater detail, specifically to identify those that are “antigen-encountered” as defined as those specifically expressing genes of T cell activation (GZMB, GZMA, PRF1, IFNG, TNFA, CD69 and/or TNFRSF9). This data suggests the presence of subpopulations of “antigen-encountered” CD8+ cells with shared clonotypes at both sites. Generally, the number of “activation genes” is higher in the primary tumors than nodal mets in six out of seven patients, reflecting higher tumor burden and an “earlier encounter” as expected. We have included this data in the Results section, Supplementary Figure 5V and herein as Rebuttal Figure 2.

V

Rebuttal Figure 2. (V) T cell clones expressing *GZMB*, *GZMA*, *PRF1*, *IFNG*, *TNFA*, *CD69* and/or *TNFRSF9*, and have clonal sharing between primary and lymph nodes (metastatic) tumors of patients. Combined clonotypes (left) and individual clonotype (right) at each site. Each dot point represents one cell.

(6) Thank you for this clarification.

>Thank you very much.

(7) and (8) The authors continue to focus on single genes that are differentially expressed in individual tumors and have not discussed the relevance of AXL being a targetable marker of “pre-nodal” cells. In general, the authors have not clearly defined what is meant by “pre-nodal” cells or what the significance of these cells may be. Are the same actionable markers within pre-nodal cells (AXL, AURKB) that were found in patient derived cultures also present within fresh tumor samples? Are pre-nodal cells also thought to be present in tumors that have not yet displayed lymph node metastases?

>We thank the reviewer for the comments as it prompted a number of necessary clarifications and to run further analyses using TCGA data.

- Firstly, we defined the putative “pre-nodal” subpopulation up-front as suggested, and as follows (in the Results section): “One of the major objectives here was to identify pre-nodal cells, which are cancer cells within that primary tumor that have the capacity to metastasize to the lymph nodes, and hence we hypothesize should have similar gene signatures to cancer cells within the lymph node”.
- Secondly, we did find the same actionable markers in the fresh tumor samples as the cell culture models, as shown in Figure 2F, and Supplementary Figure 2D-F and O, and these include AXL, AURKB, AURKA and RXRA.
- The question about pre-nodal cells in primary tumor prior to the event of lymph node metastases prompted us to interrogate TCGA data, and ask about the relationship between AXL or AURKB with EMT score in HNSCC. Data from these tumors were then separated into two groups of patients: 1) regardless of nodal status or 2) patients with no nodal metastasis only (ie N0 disease). It is important to note that these were all primary tumors with differing tumor content, and not an ideal system to directly answer our question as the data available was bulk RNAseq data and the correlation was only with one parameter (ie EMT). Despite these numerous caveats, the results demonstrated that AXL expression is significantly, positively-correlated with increasing EMT scores, even if this analysis was only limited to patients with no nodal metastasis. Conversely, we can also demonstrate that AURKB has an opposite trend, even in N0 tumors, although the association is less robust. Therefore, in line with the reviewer’s questions, these results support the notion that pre-nodal cells can be present in primary tumors that have not yet displayed lymph node metastases. We have included this data in the results section, figures as Supplementary Figure 3L-M and herein as Rebuttal Figure 3.

L

All nodal status

M

Nodal status 0

Rebuttal Figure 3. (L-M) Scatter plots showing the relationship between AXL or AURKB and EMT score of HNSCC primary tumors from patients (L) regardless of nodal status ($n=500$) or (M) with no nodal metastasis only ($n=171$). (L-M) Data derived from TCGA. ρ = Pearson correlation; $P \leq 0.05$ indicates statistical significance.

(9) Thank you for the clarification and for the additional data. Still, the authors should better acknowledge that cSCC is a completely different entity than mucosal HNSCC, both clinically and biologically, and should not be used as a validation dataset for findings in HNSCC. cSCC findings may merely be used as supporting evidence for a general concept.

>Thank you very much for this comment, and we agree that this is merely supporting and not validating data. As such, we have significantly toned down our statement in the Results section as follows: "Although cSCC is a completely different entity than mucosal HNSCC, the cSCC scRNAseq data lends support to a more general concept of these specific genes in tumor-targeting CD8+ cells, and the effect of immune checkpoint blockade on the expression of these transcription factors."

We used this dataset as it was a well-controlled, accessible dataset with scRNAseq data from sufficient CD8+ TILs derived from patients before and after pembrolizumab treatment. Such datasets are still limited in the public domain.

(10) Thank you for the clarification.

>Thank you very much.

(11) and (12) Thank you for the clarification. I understand the limitations of data and need to pare down the dataset as analyses proceed. However, Figure 6 continues to use an entirely different set of tumors, particularly with regard to MDK inhibitor treatment and it is unclear why this is the case. Were these new tumors chosen for a particular reason?

>We appreciate the reviewer's understanding. We used fresh rather than cryopreserved tumor cell suspensions as the latter was not sufficiently viable nor representative of the original tumor for such co-culture type experiments (at least in our hands). We have generally found that it is more feasible and reliable to use freshly-dissociated tumors to run such experiments, which fortunately we have access to, and hence conducted these on a separate cohort of tumors. These tumors were not chosen for any particular reason, but we felt that being able to replicate the expected phenotype using MDK-inhibitors provided better orthogonal validation of our hypothesis than using the same tumors (although we agree that ideally we would have loved to have seen the phenotype in the same tumors). For the the MDK inhibitor treatment experiment we did not have any viable cryopreserved tumor cell suspension from the same patients subjected to the single RNAseq analysis, and hence we could only perform this experiment with tumors from new patients.

We hope you find these responses satisfactory, as we believe that the clarifications and further analyses performed consequent to the reviewer comments have certainly improved our manuscript. We look forward to a favorable response.

Yours sincerely,

N Gopalakrishna Iyer, MBBS (Hons), PhD (Cantab), FRCSEd
Head, Department of Head and Neck Surgery, National Cancer Centre Singapore
Principal Investigator, Cancer Therapeutics Research Laboratory
Professor, Duke-NUS Medical School

Reviewers' Comments:

Reviewer #5:

Remarks to the Author:

The authors have satisfactorily addressed the comments and concerns with changes to the language, added discussion of limitations, and new analyses. There are no new concerns and thus I believe the work is acceptable for publication.

Dear Reviewer and Editorial team,

Thank you very much for the detailed review and comments in response to our submitted manuscript. We really appreciate the time and effort of everyone involved.

REVIEWERS' COMMENTS

Reviewer #5 (Remarks to the Author):

The authors have satisfactorily addressed the comments and concerns with changes to the language, added discussion of limitations, and new analyses. There are no new concerns and thus I believe the work is acceptable for publication.

>Thank you very much for hour help, support and constructive comments.